# Neural correlates and reinstatement of recent and remote memory in children and young adults

Iryna Schommartz[1,2]*, Philip F Lembcke[3,4], Javier Ortiz-Tudela[1], Martin Bauer[3], Angela M Kaindl[4,5,6,7], Claudia Buss[3,8]†, Yee Lee Shing[1,2]*†

[1]Department of Psychology, Goethe University Frankfurt, Frankfurt, Germany; [2]Center for Individual Development and Adaptive Education of Children at Risk (IDeA), Frankfurt, Germany; [3]Charité – Universitätsmedizin Berlin, Department of Medical Psychology, Berlin, Germany; [4]Charité – Universitätsmedizin Berlin, Department of Pediatric Neurology, Berlin, Germany; [5]Charité – Universitätsmedizin Berlin, Center for Chronically Sick Children, Berlin, Germany; [6]Charité – Universitätsmedizin Berlin, Institute of Cell- and Neurobiology, Berlin, Germany; [7]Charité – Universitätsmedizin Berlin, Department of Pediatric Surgery, Berlin, Germany; [8]Development, Health and Disease Research Program, Department of Pediatrics, University of California, Irvine, Irvine, United States

*For correspondence:
schommartz@psych.uni-frankfurt.de (IS);
shing@psych.uni-frankfurt.de (YLS)

†These authors contributed equally to this work

Competing interest: The authors declare that no competing interests exist.

## eLife Assessment

This paper provides potentially **valuable** insight into why memory consolidation may differ between children (5-7 years of age) and adults. The work hints at developmental differences in neural engagement during the retrieval of recent and remote memories. However, there are several major concerns with the analyses not alleviated by included controls, and as such the evidence supporting the authors' main claims remains **incomplete**.

**Abstract** Memory consolidation tends to be less robust in childhood than adulthood. However, little is known about the corresponding functional differences in the developing brain that may underlie age-related differences in retention of memories over time. This study examined system-level memory consolidation of object-scene associations after learning (immediate delay), one night of sleep (short delay), as well as 2 weeks (long delay) in 5- to 7-year-old children (n=49) and in young adults (n=39), as a reference group with mature consolidation systems. Particularly, we characterized how functional neural activation and reinstatement of neural patterns change over time, assessed by functional magnetic resonance imaging combined with representational similarity analysis (RSA). Our results showed that memory consolidation in children was less robust and strong (i.e. more forgetting) compared to young adults. Contrasting correctly retained remote vs. recent memories across time delay, children showed less upregulation in posterior parahippocampal gyrus, lateral occipital cortex, and cerebellum than adults. In addition, both children and adults showed a decrease in scene-specific neural reinstatement over time, indicating time-related decay of detailed differentiated memories. At the same time, we observed the emergence of generic gist-like neural representations in prefrontal brain regions uniquely in children, indicating qualitative difference in memory trace in children. Taken together, 5- to 7-year-old children, compared to young adults, show less robust memory consolidation, possibly due to difficulties in engaging in differentiated neural representations in neocortical mnemonic regions during retrieval of remote memories, coupled with relying more on gist-like generic neural representations.

## Introduction

Every day we form new memories that may become long-lasting through memory consolidation, a complex process in flux between encoding and retrieval (*Dudai, 2012*; *Josselyn et al., 2015*; *Moscovitch and Gilboa, 2022*; *Semon, 1921*). During systems-level consolidation, memory representations and traces are reorganized across medial temporal lobe and neocortical brain networks (*Ranganath and Ritchey, 2012*; *Ritchey and Cooper, 2020*). These networks include brain regions that are involved both in initial encoding and in integration of new memories as time passes (*Axmacher and Rasch, 2017*; *Dudai, 2012*; *Moscovitch and Gilboa, 2022*; *Squire et al., 2015*). While decades of work have shed light on general neural mechanisms of memory consolidation in adults (*Moscovitch and Gilboa, 2022*; *Sekeres et al., 2017*; *Winocur and Moscovitch, 2011*), much less is known about neural mechanisms that support memory consolidation in children – a knowledge gap that we aimed to address with the current study.

### Neural correlates of memory consolidation

Learning through repeated activation and reinstatement is one way to rapidly stabilize memory traces and make them accessible upon retrieval (*Dudai, 2004*; *Nader and Hardt, 2009*; *Teyler and Rudy, 2007*). For instance, in young adults, repeated exposure to word-image pairs during encoding, compared to single exposure, was shown to accelerate memory consolidation. This is achieved through enhanced replay of repeated events in the retrosplenial cortex (RSC) and the medial prefrontal cortex (PFC), as well as via increased hippocampal (HC)-cortical replay that promotes the associative word-object memories (*Yu et al., 2022*). In another study by *Brodt et al., 2016*, it was found that during repeated spatial navigation in a virtual environment, activation in the posterior parietal cortex (PPC), especially the precuneus, increased and remained elevated after 24 hr, while HC activity and HC-PPC connectivity declined with repeated encoding rounds (*Brodt et al., 2016*). In addition, neocortical plasticity measured by diffusion-weighted magnetic resonance imaging (MRI) in the PPC (*Brodt et al., 2018*) and the cerebellum (*Stroukov et al., 2022*) supported rapid cortical storage of memory traces for object-location associations after repeated exposure in young adults 1 hr and 12 hr post-learning. Taken together, these findings indicate that repeated learning in young adults promotes fast creation of neural memory representations, which can remain stable for at least 24 hr and predict behavioral mnemonic performance.

Memory consolidation of well-learnt information does not end with the last learning cycle, but undergoes further neural reorganizing and modification over time (*Roüast and Schönauer, 2023*; *Sekeres et al., 2017*). For example, during cued recall of face-location associations, young adults who were tested 24 hr after learning, compared to 15 min, showed increased activation in the precuneus, inferior frontal gyrus, and fusiform gyrus, whereas the hippocampus showed a decrease in activation (*Takashima et al., 2009*). Similarly, increased activation in the anterior temporal cortex during the retrieval of studied figure pairs 8 weeks prior was observed, while increased activation in the HC was shown for pairs learned immediately before retrieval (*Yamashita et al., 2009*). Furthermore, delayed retrieval of naturalistic video clips after a delay of 7 days in young adults was associated with increased activation in the lateral and medial PFC and decreased activation in the HC and parahippocampal gyrus (PHG) activations over time (*Sekeres et al., 2021*). This is convergent with the notion that the role of the PFC increases during recollection as consolidation progresses over time (*Milton et al., 2011*). Moreover, subsequently recollected memories showed higher post-rest HC-lateral occipital cortex (LOC) connectivity specifically related to scene-related mnemonic content, indicating the role of LOC in associative memory consolidation (*Tambini et al., 2010*). On the other hand, HC activation has been reported to remain stable after 7 days (*Sekeres et al., 2018b*), 3 months (*Harand et al., 2012*), or even years (*Söderlund et al., 2012*) for consistent episodic memories that retained contextual details.

To summarize, in alignment with the Multiple Trace Theory (*Nadel et al., 2000*; *Nadel and Moscovitch, 1997*), studies have shown that memories of well-learned information increasingly engage cortical regions over time. The regions include the prefrontal, parietal, occipital, and anterior temporal brain areas, supporting the retrieval of general and schematic memories, as well as complex associative information. In line with the Standard Consolidation Theory, some studies have demonstrated a decrease in the recruitment of the HC over time (*Squire and Alvarez, 1995*). Conversely, and converging with the Contextual Binding Theory (*Yonelinas et al., 2019*) and the Multiple Trace

Theory, some studies have shown that HC involvement lingers over time, particularly for detailed and contextual memories. However, most research has focused on only a selected delay window and solely on young adults.

## Mnemonic transformation and reinstatement across consolidation

In addition to changes in neural activation during mnemonic retrieval over time, it is important to characterize the transformations and reinstatement of neural representations – i.e., distinctive pattern of neural activity generated by a specific memory (*Averbeck et al., 2006*; *Kriegeskorte et al., 2008*; *Kriegeskorte and Kievit, 2013*) – as these multivariate patterns of neural activity may change over time. For example, memory for perceptual details often declines over time, while memory for gist may tend to remain more stable, suggesting differential temporal trajectories of transformation (*Sekeres et al., 2016*). According to Fuzzy Trace Theory (*Reyna and Brainerd, 1995*; *Reyna and Brainerd, 1998*) and Trace Transformation Theory (*Moscovitch and Gilboa, 2022*), detailed and gist-like memories may be uniquely present or coexist, depending on the strength of formed memories. For instance, *Diamond et al., 2020*, showed that the specific accurate nature may be preserved for correctly recalled memories. In other instances, initially weak detailed memories may be reorganized over time, with lingering specific memories and parallel creation of gist-like generic memories. Further research supports the idea that memory traces undergo transformation and abstraction beyond simple perceptual reinstatement (*Chen et al., 2017*; *St-Laurent and Buchsbaum, 2019*; *Ye et al., 2020*), pointing to the development of schematic, generic representations. However, relatively little is known about how the neural representation of well-learned memories change over the consolidation period – particularly how similar patterns of neural activity are reactivated upon retrieval again (*Clarke et al., 2022*; *Deng et al., 2021*).

Using representational similarity analysis (RSA; *Kriegeskorte et al., 2008*), *Tompary and Davachi, 2017*, showed that a 1-week delay led to differential memory reorganization in HC and mPFC for memories with and without overlapping features. Specifically, after a 1-week mnemonic representation became more similar for memories with overlapping features, indicating consolidation-related gist-like neural reorganization. Moreover, the authors showed memory-specific reinstatement of neural patterns for specific memories in the right HC, indicated by significant encoding-retrieval similarity for remote but not recent memories. Comparing neural reinstatement of visual clips during encoding, immediate, and delayed recall (after 1-week period), *Oedekoven et al., 2017*, showed reliable reinstatement in core retrieval networks, including the precuneus, medial temporal gyrus, occipital gyrus, HC, and PHG among others. In contrast to *Tompary and Davachi, 2017*, this study found no time-related differences in reinstatement effects. Therefore, the findings on memory reinstatement are mixed, and, to date, no study has directly tracked the neural representations of memory traces for perceptual together with more abstract, gist-like features (e.g. semantic categories).

## Neural correlates of memory consolidation and mnemonic transformation and reinstatement in middle childhood

Brain regions involved in memory consolidation show protracted developmental trajectories from early to late childhood (*Badre and Wagner, 2007*; *Gogtay et al., 2004*; *Keresztes et al., 2022*; *Lenroot and Giedd, 2006*; *Mills et al., 2016*; *Ofen et al., 2007*; *Shing et al., 2008*), which could lead to differences in neural activity and/or patterns and subsequently mnemonic reinstatement between children and adults. For instance, univariate selectivity was reduced in children, while fine-grained neural representational similarity along the ventral visual stream was similar in 5- to 11-year-old children and adults (*Cohen et al., 2019*; *Golarai et al., 2015*). *Fandakova et al., 2019*, also showed that the neural representational distinctiveness of information during encoding was similar in 8- to 15-year-old children and adults in the RSC, LOC, and PHG. The fidelity of neural representations was also associated with subsequent memory in a similar way between children and adults. Overall, although these findings did not address the question of neural reinstatement directly in children, they suggest that mnemonic reinstatement may develop prior to univariate selectivity. However, it is yet to be investigated. Moreover, it is unclear whether the age-related differences in neural activation and reinstatement mentioned above are similar for memory consolidation. Specifically, to what extent does consolidation-related transformation of neural representations occur, and how does it impact neural reinstatement of mnemonic content in the developing brain?

In middle childhood, the trade-off between retaining vivid, detail-rich memories and their transformation into vague, gist-like memories due to delay may be more pronounced. *Brainerd et al., 2002b*, demonstrated that, during development, specific memory and gist memory for events emerge together. However, as children mature, they exhibit more false memories based on gist in the absence of exact memories for the events. On the other hand, *Keresztes et al., 2018*, postulated that younger children tend to rely more on generalization when forming new memories, while older children and adults use more specific detail-rich information, suggesting a shift from generalization to specificity as children mature. Hence, there are some inconsistencies in the theoretical postulations and findings regarding item-specific and gist-based memories that may impact memory consolidation in middle childhood. Investigation on the neural reinstatement patterns of item-specific and gist-like memories across time may add to the understanding of these inconsistencies in children.

### Aim of the current study

In this study, we examined the univariate neural activation and multivariate neural reinstatement patterns of memories for object-location associations across a short delay (after one night of sleep) and a long delay (after a 2-week period), relative to recently consolidated memories (after 30 min). Children (5 to 7 years of age) were compared to young adults serving as a reference group with a mature memory consolidation system. We selected 5–7 years as the age range of interest because previous studies showed a large improvement in associative memory around this age (*Sluzenski et al., 2006*). Practically, this is also the youngest age range in which MRI scanning coupled with active task execution could be applied relatively successfully.

We hypothesized (i) according to the Multiple Trace Theory, an increasing involvement of prefrontal, parietal, cerebellar, occipital, and PHG brain regions over time in adults in comparison to children, as these regions are still maturing in preschool and early school-aged children (*Ghetti and Bunge, 2012*; *Keresztes et al., 2022*; *Lebel et al., 2012*; *Shing et al., 2008*; *Shing et al., 2010*); (ii) according to the Contextual Binding Theory, the Multiple Trace Theory, and supported by the evidence from *Sekeres et al., 2018a*, a stable involvement of HC over time in adults and children due to relative maturity of the HC in middle childhood and detailed contextual nature of the repeatedly learned information, as our task emphasizes spatial-contextual binding of objects within scenes (*Keresztes et al., 2017*; *Nadel et al., 2000*; *Sekeres et al., 2018b*; *Shing et al., 2008*; *Sluzenski et al., 2006*; *Yonelinas et al., 2019*); (iii) a decreasing neural reinstatement in all regions of interest (ROIs) over time, with this decrease being more pronounced in children compared to young adults (*Cohen et al., 2019*; *Golarai et al., 2015*); (iv) qualitative differences in representational format between age groups. Specifically, we expected more generic category-level gist-like memory representations in children, whereas adults would retain more detailed item-specific reinstatement patterns over time due to differences in the strength of formed memories and differences in underlying associative and strategic components of memories (*Reyna and Brainerd, 1995*; *Shing et al., 2008*; *Shing et al., 2010*). This assumption aligns with the Fuzzy Trace Theory (*Brainerd and Reyna, 2002a*), which posits that verbatim and gist representations are encoded in parallel and that verbatim memories can be created without the extraction of gist. Our task design – involving repeated strategic learning – may foster the emergence of consolidation-driven, categorical gist-like neural representation in children. These are to be distinguished from mature semantic gist as defined by the FTT for verbal material. Due to the ongoing maturation of associative and strategic memory components and their underlying neural substrates, children may be more inclined to extract generic category-based gist information at the expense of detailed information.

## Results

### Behavioral results

#### Final learning performance

Unique sets of object-location association pairs were learned on day 0, day 1, and day 14. During each initial encoding trial, participants were presented with an object within a congruent scene (e.g. a fox in a spring pine tree forest), and were asked to memorize the exact location of the object within the scene by creating a story and making 'mental' pictures of the scene. The choices for locations varied across scenes while they remained constant across time within individuals. There were 18 unique key

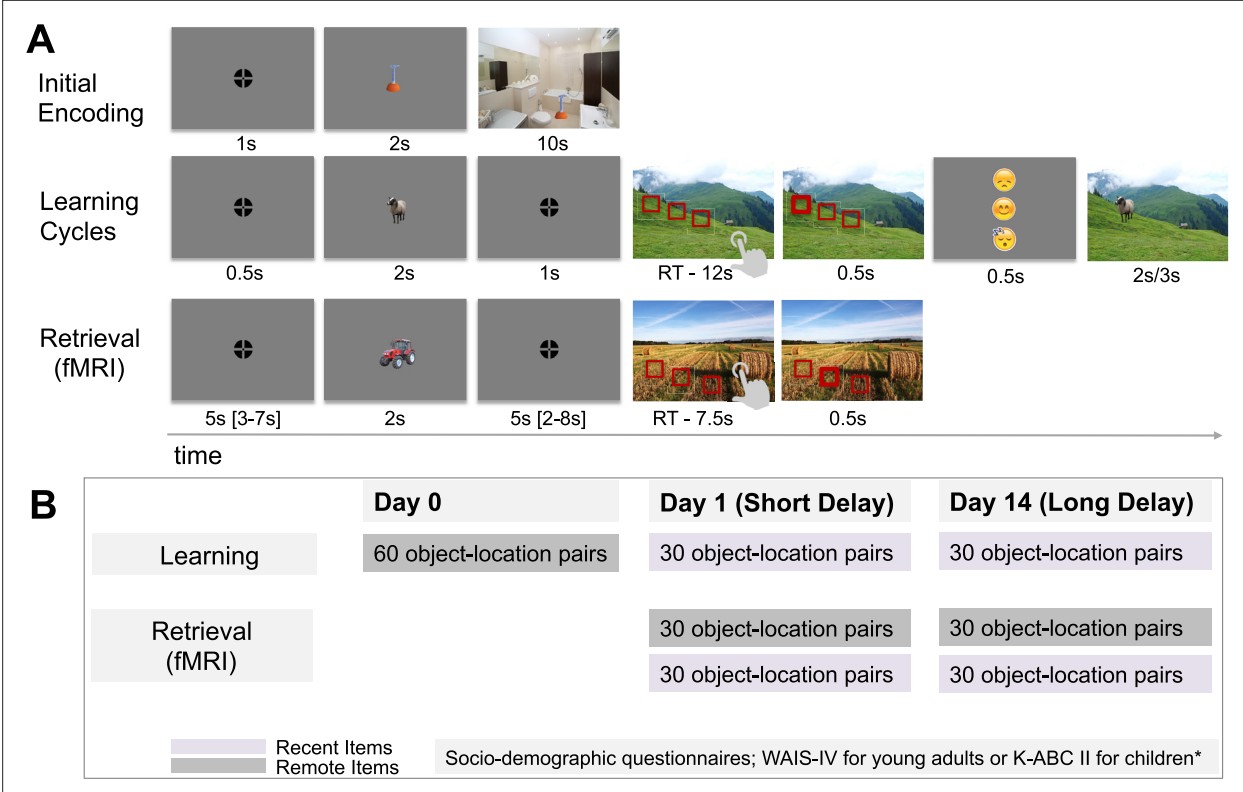

**Figure 1.** Experimental Task and Procedure. (**A**) Trial structures in the experimental task. (i) *Initial encoding*: Participants memorized object-location pairs by creating a story or forming a 'mental photo' of each scene, focusing on the exact location of the object within the scene. (ii) *Learning phase*: Participants selected one of three possible object locations and received feedback: a happy face for correct responses, a sad face for incorrect ones, and a sleeping face for missed responses. The correct object-location pairing was then displayed again. (iii) *Retrieval phase:* Conducted inside the MR scanner, participants chose the object's location in the scene from three options without receiving feedback. (**B**) Experimental procedure. Testing took place across 3 days. On day 0, participants learned 60 object-location associations (*remote items*). On day 1 (*short delay*), they learned 30 new object-location associations (*recent items*) and retrieved 30 remote and 30 recent items. On day 14 (*long delay*), participants learned another 30 new associations and retrieved 30 remote and 30 recent items. Throughout all sessions, participants also completed socio-demographic and psychometric questionnaires, which were distributed across sessions. *Note*: RT – reaction time; s – second, fMRI – functional magnetic resonance imaging.

locations among which objects could be distributed, resulting in a heterogeneous set of locations for objects. We employed an adaptive, repetitive learning-to-criteria procedure to ensure initially strong memories (see *Figure 1A* for the task overview and *Figure 1B* for experimental procedure overview).

Before the learning phase began, participants were instructed to create stories to help them memorize the locations of objects within scenes. To familiarize themselves with this strategy, they first practiced this strategy on two unique sets of five object-location associations. Learning then commenced with the initial encoding block, followed by adaptively repeated retrieval-encoding cycles to strengthen memory for the object locations. During these learning cycles, participants were presented with the same scenes again, now with three rectangles indicating possible locations for each previously learned object. The task followed a three-alternative forced-choice (3AFC) task format, with the correct location randomly appearing on the left, middle, or right. The rectangles were presented in close proximity within each scene, requiring participants to recall location details with high precision. Participants were asked to choose one rectangle that corresponded to the correct location of the object within the scene (*Figure 1A*, 'Learning Cycles'). After each response – regardless of accuracy – the object was shown in its correct location to reinforce learning. The learning cycles were repeated for a minimum of two times and a maximum of four times, or until participants achieved at least 83% accuracy in one cycle. This 83% threshold, established through pilot testing, served as a guideline for starting the next learning cycle rather than as a strict learning criterion to exclude participants. Participants who did not reach this threshold after four cycles were still included in the analysis if their

**Table 1.** Sample characteristics by age group.

| Demographic measures | Children (CH; N=49) | | Young adults (YA; N=39) | | Group effect (CH vs. YA) | |
| --- | --- | --- | --- | --- | --- | --- |
| | M | SD | M | SD | p-Value | $\omega^2$ |
| Age | 6.34 | 0.43 | 25.60 | 2.79 | *** | 0.96 |
| Sex (M/F) | 27/22 | – | 20/19 | – | – | – |
| IQ score | 117.90 | 12.92 | 107.64 | 12.49 | *** | 0.13 |
| Socioeconomic status | | | | | | |
| ISCED – father | 6.22 | 1.43 | 4.39 | 1.75 | *** | 0.29 |
| ISCED – mother | 6.17 | 1.34 | 4.08 | 1.85 | *** | 0.24 |

*Notes*: Income is based on a 1–7 scale (1=less than 15,000 €, 7=more than 100,000 €); ISCED = **Institute for Statistics (UIS), 2011** (**Institute for Statistics (UIS), 2011**); IQ = intelligence quotient based on K-ABC (**Kaufman and Kaufman, 2015**) for children and WAIS-IV (**Wechsler, 2015**) for young adults; M=mean; SD = standard deviation; $\omega^2$=omega squared; *p<0.05; **<0.01, ***<0.001 (significant difference).

performance exceeded chance level (33%). All participants demonstrated at least average cognitive abilities, as determined by a standardized intelligence test (see *Table 1*).

Concerning number of learning cycles, the linear mixed effects (LME) model revealed a significant *Group* effect, $F_{(1,563)}$ = 7.09, p=0.008, $\omega^2$=0.01, with children needing more learning cycles to reach the learning criteria in comparison to adults, b=–0.43, $t_{(563)}$ = –2.66, p=0.008, within the defined minimum and maximum of learning cycles (*Figure 2A*). Five child participants did not reach the learning criteria after the fourth learning cycle, and their final performance ranged between 70% and 80%. The number of learning cycles did not differ between sessions as revealed by nonsignificant *Session* effect and *Group × Session* interaction (all p>0.40).

Final learning accuracy, operationalized as the percentage of correctly identified object locations, was significantly higher in young adults than in children, $F_{(1,79)}$ = 94.31, p<0.001, $\omega^2$=.53, $t_{(185)}$ = 7.55, p<0.001 (*Figure 2B*), as revealed by the LME model. There was no significant effect of *Session* (p=0.79) and no *Session × Group* interaction (p=0.96), indicating that the learning accuracy was stable across sessions with different stimuli sets. Although the learning procedure was adaptive, children showed consistently lower learning performance compared to young adults.

## Memory retention across time

Changes in memory retention were assessed during the retrieval part of the memory task (*Figure 1A*, 'Retrieval (fMRI)'). Participants were cued with the object and were instructed to recall as vividly as possible the associated scene and the location of the object within the scene during the fixation window, when no visual input was presented on the screen. The associated scene was then presented with three choices, and participants had to select a rectangle indicating the correct location of the object in the scene (see Materials and methods for more details).

First, we investigated whether retention rates for recently learned items (initially correctly encoded on day 1 and day 14) differed between sessions in children and adults. The *Session × Group* interaction was not significant, $F_{(1,75)}$ = 1.77, p=0.187, $\omega^2$=0.001, indicating that retention rate differences across sessions did not vary significantly between groups. Based on that, we averaged recent retention rates across sessions within each group for subsequent analysis.

Second, we examined changes in memory retention rates for items that were initially correctly learned (i.e. strong initial memories), focusing on group differences in recent and remote (short- and long-delay) memory retention relative to a 100% baseline (see *Figure 3*). The LME model predicting retrieval accuracy for learned object-location pairs explained a substantial portion of variance, $R^2$=0.77, 95% CI [0.73 –0.81]. We observed a significant main effect of *Item Type*, $F_{(3,250)}$ = 229.18, p<0.001, $\omega^2$ = 0.73. Post hoc comparisons revealed no significant difference between recent memory retention and short-delay remote memory retention, b = 1.49, $t_{(259)}$ = 1.26, p=0.754. However, recent memory retention was significantly higher than long-delay remote retention, b = 21.36, $t_{(259)}$ = 17.59, p<0.001, and short-delay remote retention was significantly higher than long-delay remote memory retention, b = 19.88, $t_{(260)}$ = 16.16, p<0.00. Further, we observed a significant main effect of *Group*, $F_{(1,85)}$ = 55.00,

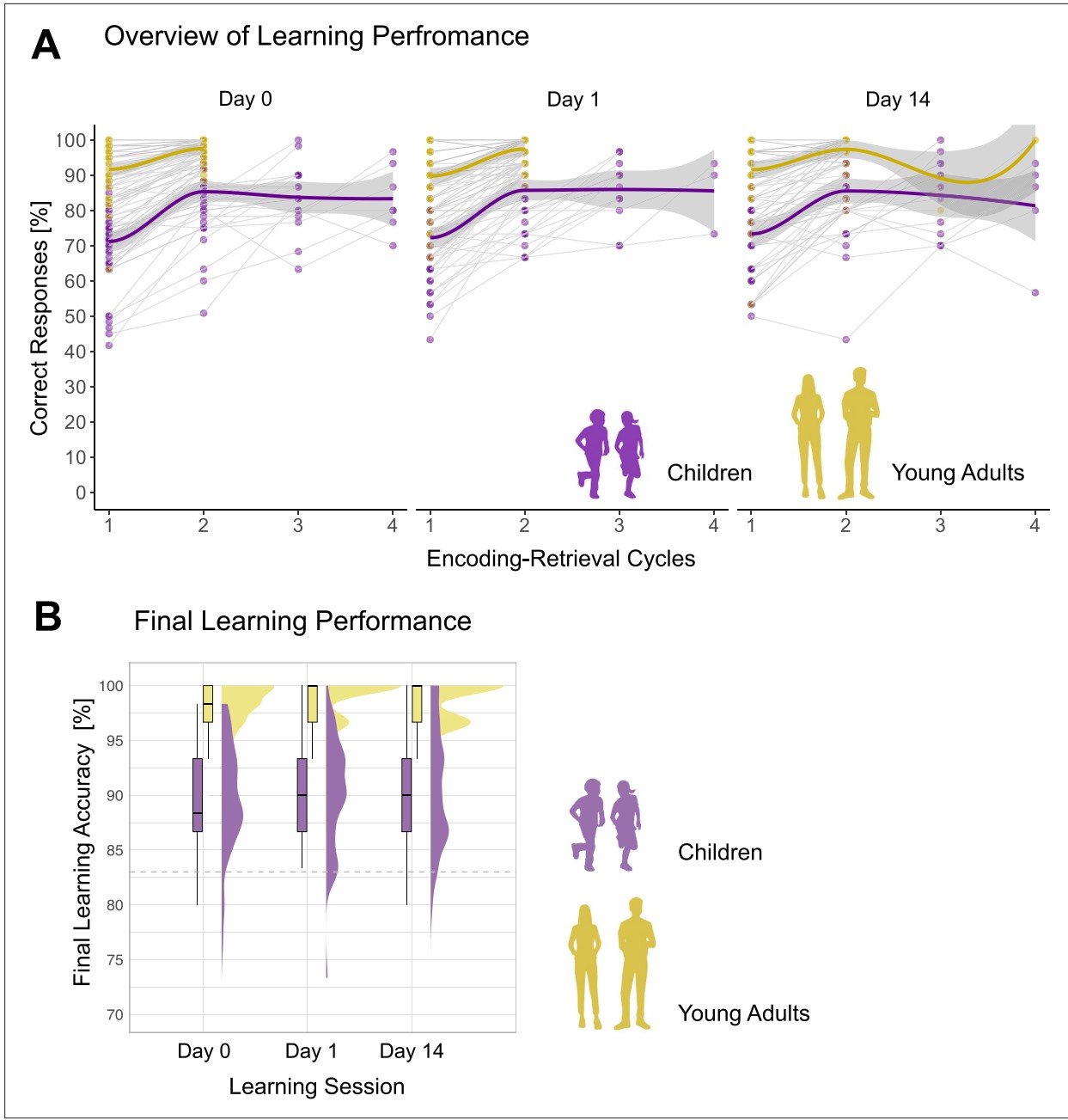

**Figure 2.** Learning Performance. (**A**) Overview of learning performance. Individual learning trajectories across up to four encoding-retrieval cycles for children and young adults on day 0, day 1, and day 14. Each colored dot represents a participant's accuracy (percentage of correct responses) at a given cycle. Transparent connecting lines illustrate within-person changes in accuracy across cycles. Across all sessions, children needed on average between two to four learning-retrieval cycles to reach the criterion of 83% correct responses, while young adults typically reached it within two cycles. (**B**) Final learning performance. Final learning accuracy is calculated as the percentage of correct responses during the last learning cycle for both children and young adults. For each group and session, distributions are visualized using half-eye plots (smoothed density estimates), overlaid with boxplots indicating the median and interquartile range. The shape and spread of the density plot reflect individual data variability. Gray dashed line indicates the criteria of 83% correctly learned items.

p<0.001, $\omega^2$ = 0.38. Post hoc comparisons revealed an overall lower memory retention in children compared to young adults, b = −11.1, $t_{(91)}$ = −7.20, p<0.001. Additionally, we observed a significant *Item Type × Group* interaction, $F_{(3,250)}$ = 17.35, p<0.001, $\omega^2$ = 0.16. Model-based Sidak post hoc comparisons showed that the slope of memory retention decline was significantly steeper in children compared to adults for recent items, b = 15.26, $t_{(254)}$ = 6.56, p<0.001, for short-delay remote items,

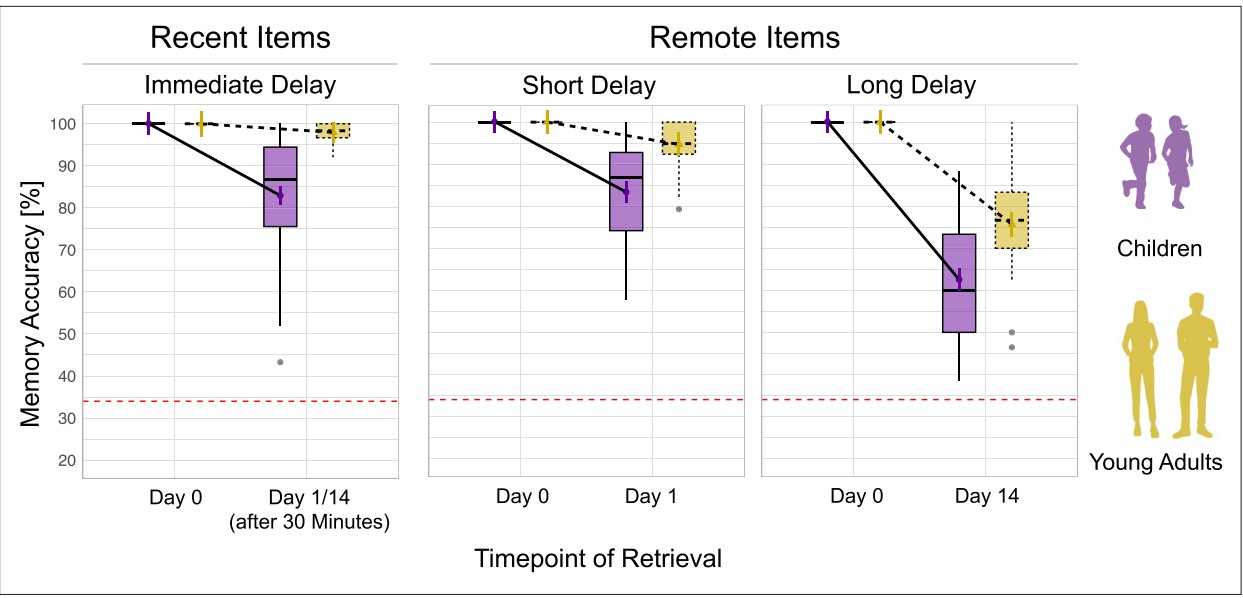

**Figure 3.** Retention rates for initially correctly learned items. Memory accuracy is operationalized as the percentage of correct responses in the retrieval task conducted during the magnetic resonance imaging (MRI) scanning sessions for items that were initially correctly learned, indicating strong initial memories. Memory accuracy for recently consolidated items did not differ between sessions in young adults and children and was collapsed across sessions. Overall, young adults show higher and more stable memory accuracy than children, with memory declining over time for both groups, particularly for long delay. All tests used Sidak correction for multiple comparisons. *p<0.05; **p<0.01; ***p<0.001(significant difference); nonsignificant differences were not specifically highlighted. The boxplot summarizes the distribution of accuracy scores across sessions and delay conditions. In each boxplot, the central line indicates the median, the box represents the interquartile range (25th to 75th percentile), and the whiskers extend to the range of values within 1.5 times the variability. The red dashed line at 34% indicates the threshold for chance-level performance.

The online version of this article includes the following figure supplement(s) for figure 3:

**Figure supplement 1.** Retention rates for initially correctly learned items (for participants who needed only two learning cycles).

b = 11.41, $t_{(255)}$ = 4.84, p<0.001, and for long-delay remote items, b = 13.08, $t_{(258)}$ = 5.38, p<0.001. In addition, memory retention rates significantly increased (corrected for multiple comparisons with false discovery rate [FDR]) with age in the child group for recent items, b = 0.89, t = 2.62, p=0.016, for short-delay remote items, b = 0.91, t = 2.67, p=0.016, but not for long-delay remote items, b = 0.15, t = 0.326, p=0.747.

Of note, we conducted an additional analysis on a subsample that included only participants who needed two learning cycles to reach the learning criterion (see *Supplementary file 1*, *Figure 3—figure supplement 1*, *Supplementary file 2* for details). Twenty-one child participants were excluded, resulting in the final subsample of n=28 children. The results from this subsample fully replicated the findings from the full sample, indicating that the amount of re-exposure to stimuli during encoding did not affect consolidation-related changes in memory retrieval at the behavioral level.

Taken together, both age groups showed a decline in memory performance over time. However, compared to young adults, children showed a steeper slope of memory decline for both immediate recent and remote short- and long-delay memories. In sum, the results showed that children had overall worse memory retention rates compared to young adults, indicating less robust memory consolidation in children.

## fMRI results

### Mean activation for *remote > recent* memory in ROIs

To investigate how neural activation for correctly recalled memories varied across different time delays, we examined the contrast of **remote >recent** correct trials during object presentation at retrieval (*Figure 4*, 'Retrieval fMRI').

We first tested whether the **remote > recent** contrast significantly differed from zero in each age group and session (day 1 and day 14), as an indicator of differential engagement during memory retrieval. FDR-adjusted results showed no significant results in the anterior and posterior HC (*Figure 4A*),

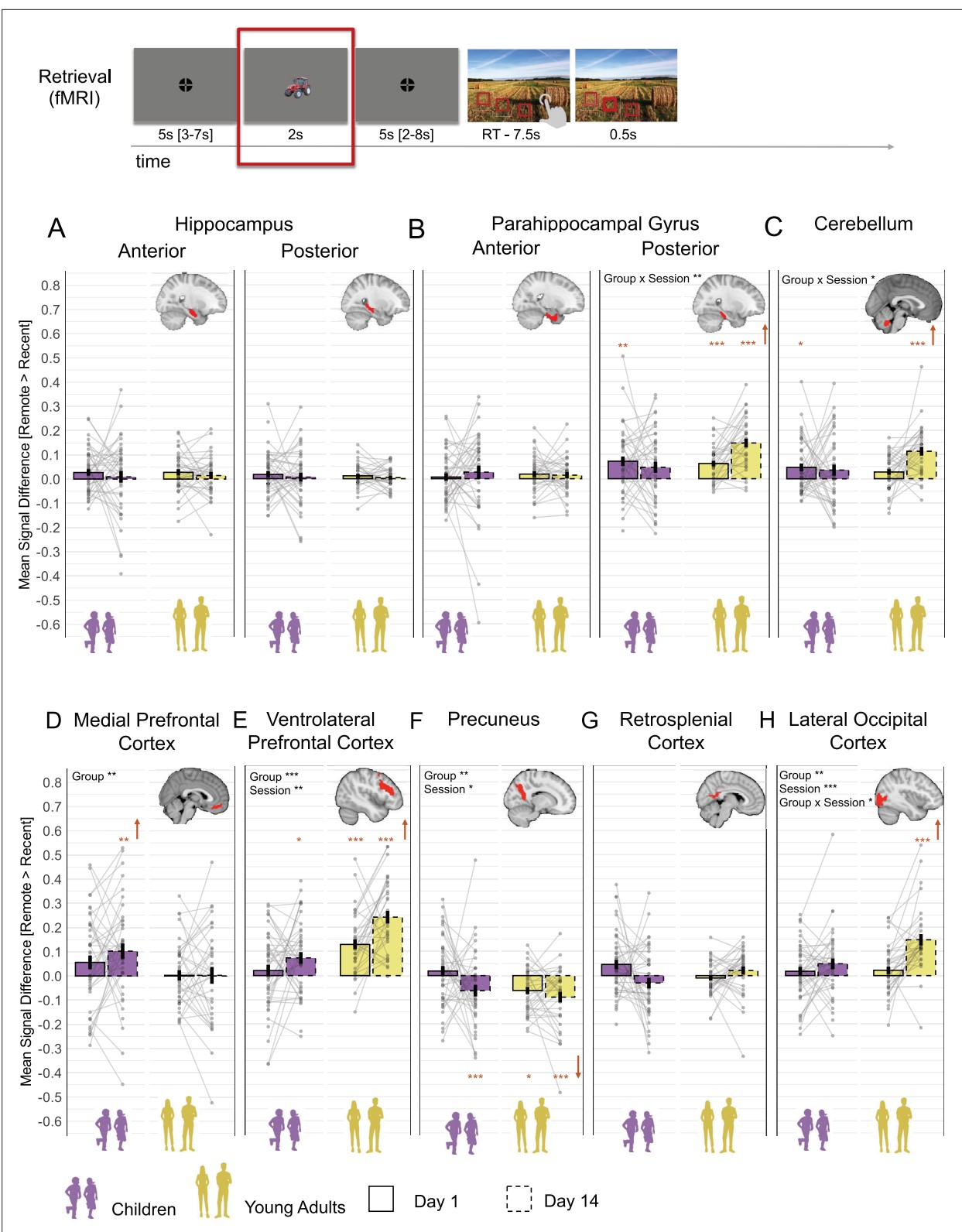

**Figure 4.** Mean signal differences between correct remote and recent memories. The figure presents mean signal difference for remote > recent contrast across sessions and groups during the object presentation time window in (**A**) anterior and posterior hippocampus; (**B**) anterior and posterior parahippocampal gyrus; (**C**) cerebellum; (**D**) medial prefrontal cortex; (**E**) ventrolateral prefrontal cortex; (**F**) precuneus; (**G**) retrosplenial cortex; (**H**) lateral occipital cortex. *Note:* Bars indicate the group mean for each session (solid lines for day 1, dashed lines for day 14), plotted separately for children and young adults. Error bars represent ± 1 standard error of the mean. The color indicated the age groups: purple for children and khaki yellow for

*Figure 4 continued on next page*

*Figure 4 continued*

young adults. Across all panels, the mean of individual subject data is shown with transparent points. The connecting faint lines reflect within-subject differences across sessions. Orange asterisks denote significant difference of **remote > recent** contrast from zero. An upward orange arrow indicates that this difference is greater than zero, while a downward arrow indicates that this is less than zero. *p<0.05; **p<0.01; ***p<0.001 (significant difference); nonsignificant differences were not specifically highlighted. Significant main and interaction effects are highlighted by the corresponding asterisks. All main and interaction p-values were false discovery rate (FDR)-adjusted for multiple comparisons.

The online version of this article includes the following figure supplement(s) for figure 4:

**Figure supplement 1.** Mean blood oxygen level-dependent (BOLD) signal intensity.

**Figure supplement 2.** Mean neural activation for correctly recalled memories during scene presentation time window.

anterior PHG (*Figure 4B*), and RSC (*Figure 4G*) across sessions and age groups (all p>0.054; see *Supplementary file 3* for details). To rule out the possibility that these nonsignificant differences reflect an overall absence of retrieval-related activation, we tested whether mean activation for recent and remote items – each relative to the implicit baseline – was significantly above zero. FDR-adjusted results revealed that activation in these ROIs was significantly greater than zero (all p<0.031), except in the recent day 1 condition in children for the posterior HC (p>0.141) and the precuneus (p>0.056, see *Supplementary file 4* and *Figure 4—figure supplement 1* for details). These findings indicate that the anterior and posterior HC, anterior PHG, and RSC are similarly engaged during successful retrieval of both recent and remote memories, regardless of delay or age group. (As a control analysis, we tested whether the anterior and posterior HC, anterior PHG, and RSC were similarly engaged during retrieval of recent and remote items over time using the LME models. These models included mean activation relative to the implicit baseline, a Session × Delay × Group interaction, and Subject as a random intercept. The results were consistent with the earlier findings, showing no significant main effect of Delay [all p>0.106], Group [all p>0.060], or Session × Delay interaction [all p>0.340], indicating comparable engagement of these ROIs across delays and age groups [see *Supplementary file 6* for full statistical details].) Other ROIs showed more differentiated patterns, which are discussed below. (In contrast, the vlPFC, CE, posterior PHG and LOC, precuneus, and mPFC showed a significant main effect of Delay [all p<0.009, see *Supplementary file 5* for details], indicating time-related changes in the remote > recent contrast. These effects are examined in more detail below. Notably, these findings are consistent with results from the whole-brain analyses; *Supplementary file 7*.)

To further explore the more differentiated patterns observed in other ROIs, we examined changes in the **remote >recent** contrast across age groups and sessions (day 1 and day 14) using LME models, controlling for sex, handedness, general intelligence, and mean reaction time. All main and interaction effects were FDR-adjusted, and all post hoc tests were Sidak-corrected (see *Supplementary file 5* for details).

For the **posterior PHG** (*Figure 4B*), a significant *Session × Group* interaction, $F_{(1,83)} = 9.54$, p=0.020, $\omega^2$=0.09, indicated a more pronounced increase in **remote >recent** mean signal difference over time in young adults compared to children, b=0.11, $t_{(83)} = 3.09$, p=0.003.

Similarly, also for **the cerebellum** (*Figure 4C*), a significant *Session × Group* interaction, $F_{(1,161)} = 7.68$, p=0.020, $\omega^2$=0.04, indicated a stronger increase in **remote > recent** mean signal difference over time in young adults compared to children, b=0.09, $t_{(160)} = 2.77$, p=0.006.

For **the mPFC** (*Figure 4D*), a significant main effect of *Group*, $F_{(1,86)} = 7.61$, p=0.023, $\omega^2$=0.07, denoted that the overall **remote > recent** mean signal difference in children was higher than in young adults, b=−0.10, $t_{(86)} = −2.76$, p=0.007.

For **the vlPFC** (*Figure 4E*), a significant main effect of *Group*, $F_{(1,82)} = 31.35$, p=<0.001, $\omega^2$=0.13, indicated an overall lower **remote > recent** mean signal difference in children compared to young adults, b=−0.125, $t_{(108)} = −3.91$, p<0.001. In addition, a significant main effect of *Session*, $F_{(1,99)}$=10.68, p=0.005, $\omega^2$=0.09, pointed out overall higher **remote > recent** mean signal difference on day 14 compared to day 1, b=0.08, $t_{(99)} = 3.27$, p=0.001.

For **the precuneus** (*Figure 4F*), a significant main effect of *Group*, $F_{(1,161)} = 5.09$, p=0.027, $\omega^2$=0.02, indicated an overall lower **remote > recent** mean signal difference in adults compared to children, b=−0.05, $t_{(160)} = −2.26$, p=0.037. In addition, a significant main effect of *Session*, $F_{(1,161)} = 6.50$, p=0.036, $\omega^2$=0.03, denoted an overall lower **remote > recent** contrast for day 14 compared to day 1, b=−0.05, $t_{(160)} = −2.55$, p=0.012. Although the **remote > recent** contrasts were mostly negative, the mean

activation for recent and remote items – each relative to the implicit baseline – was significantly greater than zero for all delays and group (all p<0.023), except for children's recent items on day 1 (p=0.056).

For the LOC (*Figure 4H*), a significant main effect of *Group*, $F_{(1,82)}$ = 9.12, p=0.015, $\omega^2$=0.09, indicated a higher **remote > recent** mean signal difference in young adults compared to children, b=0.07, $t_{(82)}$ = 3.02, p=0.003. Additionally, a significant main effect of *Session*, $F_{(1,97)}$ = 16.76, p=<0.001, $\omega^2$=0.14, showed an overall increase in **remote > recent** mean signal difference on day 14 compared to day 1, b=0.07, $t_{(97)}$ = 4.10, p=<0.001. Furthermore, a significant *Session × Group* interaction, $F_{(1,81)}$ = 6.42, p=0.032, $\omega^2$=0.06, demonstrated higher increase in **remote > recent** mean signal difference over time in adults compared to children, b=0.09, t(81) = 2.53, p=0.013.

Of note, we conducted an additional univariate analysis using a subsample that included only participants who needed two learning cycles to reach the learning criteria (see *Supplementary file 8* for details). The subsampled results fully replicated the findings from the full sample and demonstrated that the amount of re-exposure to stimuli during encoding did not affect consolidation-related changes in memory retrieval at the neural level.

In summary, our findings revealed distinct consolidation-related neural upregulation for remote memory between children and adults. From day 1 to day 14, adults showed a higher increase in **remote > recent** signal difference for remembered items in the posterior PHG, LOC, and cerebellum than children. Adults showed overall higher remote > recent difference in the vlPFC than children, while children showed overall higher remote > recent difference in the mPFC than adults. Furthermore, we observed a constant activation of anterior and posterior HC, anterior PHG, and RSC in memory retrieval across age groups irrespective of memory type or delay.

## Neural-behavioral correlation

We further investigated whether neural upregulation (i.e. remote >recent univariate signal difference) is related to memory performance. Specifically, considering all ROIs simultaneously and differential directionality of remote >recent signal differences, we investigated whether any specific profile of ROI constellation of neural upregulation is related to variations in memory performance. For this purpose, we employed the partial least squares correlation analysis (PLSC; *Abdi, 2010*; *Abdi and Williams, 2013*). With regard to the interconnectedness of the predefined ROIs, the PLSC is a well-suited method to address multivariate associations between neural measures and memory measures. Consequently, latent variables (LVs) that represent differential profiles of ROI's neural upregulations with robust relation with either short- or long-delay variations in memory performance were extracted (for more detailed description of the PLSC method, refer to Materials and methods section). In addition, we derived for each subject a value that denotes a within-person robust expression of either short- or long-delay brain profile.

For each delay, the permutation test of significance resulted in a single LV that reliably and optimally represents across age groups: (i) the associations of short-delay ROI neural upregulations with variations in short-delay memory accuracy (*Figure 5A*; r=0.536, p=0.0026); and (ii) the associations of long-delay ROI neural upregulations with variations in long-delay memory accuracy (*Figure 5B*; r=0.542, p=0.0024). With further bootstrapping, we identified Z-score estimates of robustness (larger/smaller than ±1.96 (a<0.05)) of the components within the multivariate brain profiles across all participants. Thus, for short delay, we observed that higher memory accuracy was robustly associated with greater neural upregulations in the anterior PHG (Z-score=2.161, r=0.347) and vlPFC (Z-score=3.457, r=0.640), as well as with lesser neural upregulation in precuneus (Z-score=–2.133, r=–0.323) and cerebellum (Z-score=–2.166, r=–0.371) across age groups. In contrast, for long delay, we observed that higher memory accuracy was robustly associated with greater neural upregulation in the vlPFC (Z-score=3.702, r=0.492), RSC (Z-score=4.048, r=0.524), and LOC (Z-score=3.568, r=0.455), and with lesser neural upregulation in mPFC (Z-score=–2.958, r=–0.394) across age groups. The identified LVs indicate that a substantial amount of variance (short delay: r=0.536 and long delay: r=0.542) in either short- or long-delay memory performance was accounted for by the identified differential functional profiles of brain regions.

Identified brain profiles across groups suggest shared patterns between neural mean signal differences in differential sets of ROIs and memory accuracy are consistent across children and adults. As this approach optimizes for consistent covariance pattern across the full sample, it does not test for group-specific profiles. When conducting the same PLS models within each group, no stable latent

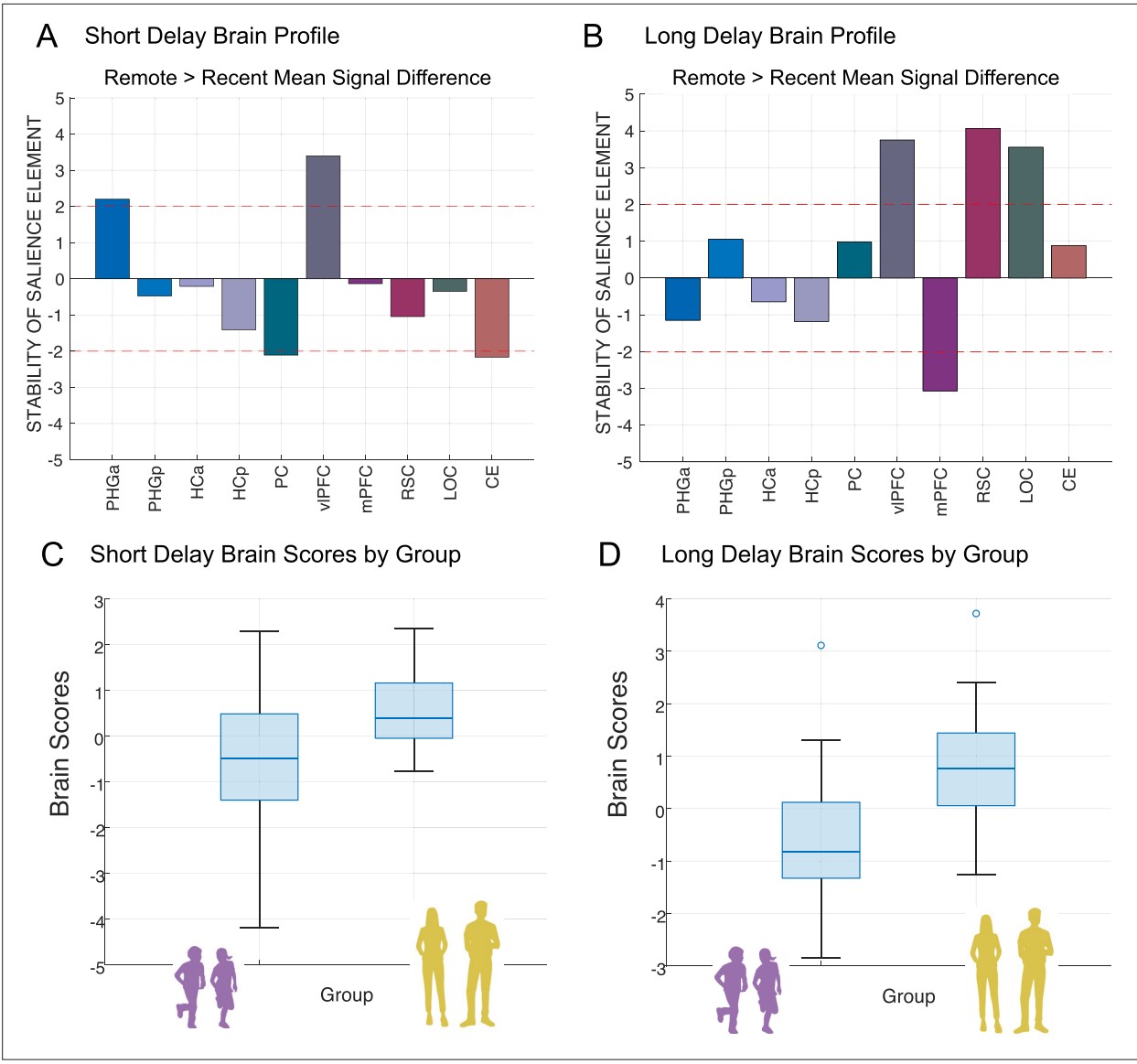

**Figure 5.** Multivariate short- and long-delay brain profiles of neural upregulation (remote vs. recent neural activation differences) are associated with variations in memory accuracy. (**A**) *Short-delay brain profile.* Latent variable weights or saliences for each ROI build up one latent variable that expresses a composite short-delay brain profile across both age groups. (**B**) *Long-delay brain profile.* Latent variable weights or saliences for each ROI build up one latent variable that expresses a composite long-delay brain profile across both age groups. The bar plot shows the bootstrap ration (BSR) values for the latent variable, reflecting the stability of the relationship between brain activation and memory performance. Stability of salience elements is defined by Z-scores depicted as red lines: a value larger/smaller than ±1.96 is treated as reliably robust at (a<0.05). (**C**) Short-delay brain scores by group. (**D**) Long-delay brain scores by group. Each box represents the distribution of brain scores within a group, with central lines indicating the median and boxes showing the interquartile range. Whiskers represent the full range of non-outlier values. *Note:* PHGa – anterior parahippocampal gyrus; PHGp – posterior parahippocampal gyrus; HCa – anterior hippocampus; HCp – posterior hippocampus; PC – precuneus; vlPFC – ventrolateral prefrontal cortex; mPFC – medial prefrontal cortex; RSC – retrosplenial cortex; LOC – lateral occipital cortex; CE – cerebellum; r – Spearman's rank order correlation index.

profile emerged (all p>0.069). The reduced within-group sample may have affected the bootstrap-based stability. To address this, we explored whether groups differ in their expression of the common LV (i.e. brain scores). This analysis revealed that children showed significantly lower brain scores than adults both in short delay, $t_{(83)}$ = –4.227, p=0.0001 (*Figure 5C*), and long delay, $t_{(74)}$ = –5.653, p<0.001 (*Figure 5D*), suggesting that while the brain-behavior profile was shared, its expression varied by group.

Taken together, differential short- and long-delay brain profiles of neural upregulation were related to variations in memory accuracy. Despite age-related differences in the derived brain scores, higher

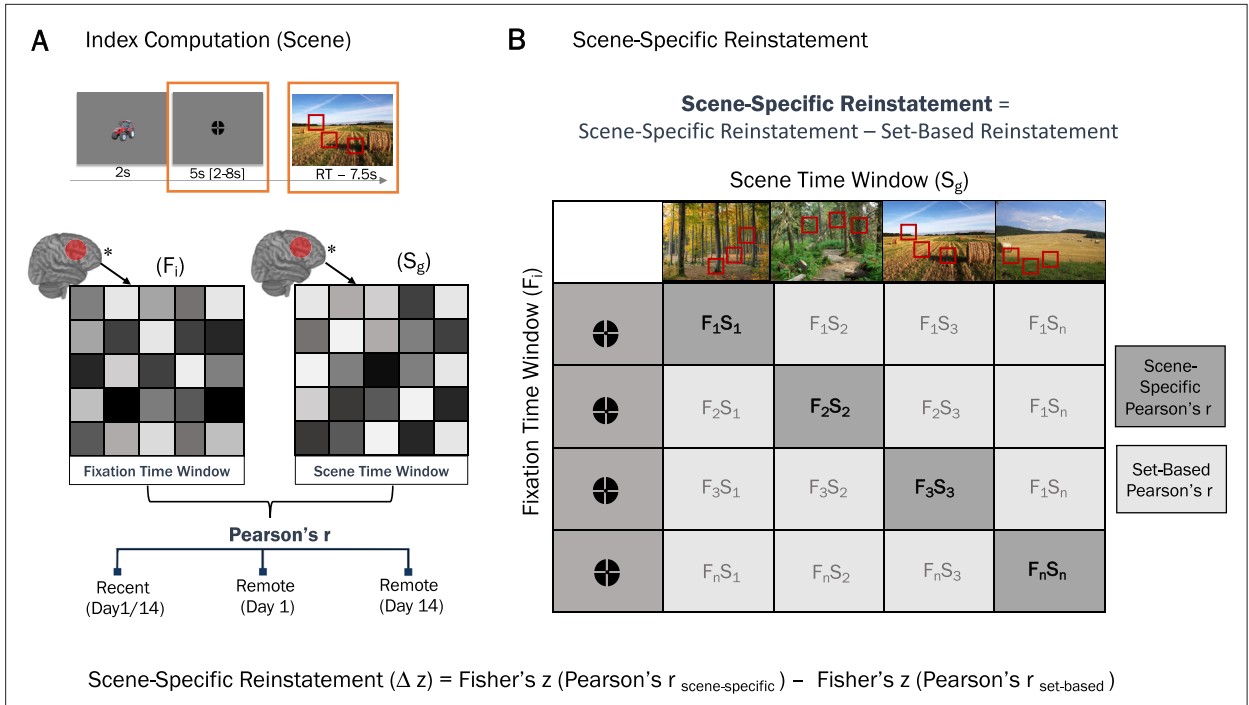

**Figure 6.** Scene-specific reinstatement. (**A**) Index computation (scene). A representational similarity index was computed by calculating the average similarity between activation patterns in the fixation and scene time windows, separately for recent scenes, remote scenes on day 1, and remote scenes on day 14. (**B**) Scene-specific reinstatement. A corrected scene-specific reinstatement index was computed by assessing the average similarity within-trial similarity between the fixation and scene time windows and subtracting the average between-trial (set-based) similarity across all other trials. This controls for baseline similarity unrelated to specific scene content. S – scene time window; F – fixation time window; r – Pearson's correlation index; Δz – difference between two Fisher-transformed r values. * – Activation patterns.

expression of within-participant brain score was associated with higher memory retention rates in short- and long-delay similarly in children and young adults.

## Representational similarity results

In addition to distinct univariate neural upregulation for recent and remote memories, children and adults may exhibit differences in neural representations of these memories. Over time, these representations could also undergo consolidation-related transformations. To address this further, we investigated both more differentiated detailed scene-specific reinstatement and more generic category-based neural representations in children and adults.

## Corrected scene-specific reinstatement

To measure how scene-specific reinstatement at retrieval during fixation time window (after short cue by object presentation; see *Figure 6A*) changes over time as memories decay, we computed a *scene-specific reinstatement index* for each neural representational similarity matrix (RSM). We hypothesized that neural patterns evoked by reinstatement of a specific scene without any visual input during fixation time window would be similar to neural patterns evoked by actual presentation of the scene during the scene time window. Therefore, the scene time window was used as a template against which the fixation period can be compared. Participants were explicitly instructed to recall and visualize the scene and location of the object during fixation time window after being cued by the object. Since the locations were contextually bound to the scene and each object had a unique location in each scene, the location of the object was always embedded in the specific scene context.

To investigate how scene-specific reinstatement changes over time with memory consolidation, all analyses were restricted to correctly remembered items. For each specific scene, the correlation between neural patterns during fixation '*fixation period*' and neural patterns when viewing the scene '*scene period*' was conducted (Fisher-transformed Pearson's r; *Figure 6B*). A *set-based reinstatement*

*index* was calculated as an average distance between '*fixation*' and '*scene*' period for a scene and every other scene within the stimuli set (*Deng et al., 2021*; *Ritchey et al., 2013*; *Wing et al., 2015*). The set-based reinstatement index reflects the baseline level of nonspecific neural activation patterns during reinstatement. We then calculated the *corrected scene-specific reinstatement index* as the difference between scene-specific and set-based Fisher-transformed Pearson's r (*Deng et al., 2021*; *Ritchey et al., 2013*; *Wing et al., 2015*). Given the temporal proximity of the fixation and scene time window, we refrain from interpreting the absolute values of the observed scene-specific reinstatement index. However, given that the retrieval procedure is the same over time and presumably similarly influenced by the temporal autocorrelations, we focus primarily on the changes in reinstatement index for correctly retrieved memories across immediate, short, and long delays. In other words, the focus in the following analysis lies on the time-related change in the scene-specific reinstatement index.

First, scene-specific reinstatement indices for recent items – tested on different days – did not significantly differ, as indicated by nonsignificant main effects of *Session* (all p>0.323) and *Session × ROI* interactions (all p>0.817) in either age group. This indicates that temporal autocorrelation was consistent across scanning sessions. Based on that, we averaged the scene-specific reinstatement indices for recent items across sessions. To investigate time-dependent changes in scene-specific reinstatement in children and young adults in the predefined ROIs, we conducted LME models, with delay (recent, remote short, and remote long delays) and group (children and young adults) for each ROI, controlling for ROI BOLD activation (*Varga et al., 2023*) during corresponding sessions. All main and interaction effects were FDR-adjusted, and all post hoc tests were Sidak-corrected for multiple comparisons.

Generally, in all predefined ROIs, we observed a significant main effect of *Session* (all p<0.001) and a significant effect of *Group* (all p<0.004, *Figure 7A–G*), except for the LOC (p=0.271, *Figure 7J*). The pattern of time-related decline was similar across age groups, as indicated by not significant *Session × Group* interactions in all ROIs (all p>0.159). There was no significant effect of *BOLD activation* (all p>0.136). The full statistical report on the LME model is in *Supplementary file 9*. A more detailed overview of the observed main effects and their Sidak-corrected post hoc tests is summarized in *Table 2*.

To ensure that the observed scene-specific reinstatement effects were not driven by general signal properties or artifacts unrelated to memory retrieval, we conducted several control analyses.

First, we repeated the reinstatement analysis using the '*object period*' instead of the '*scene period*'. The rationale was that the object and the reinstated scene during the fixation period are expected to rely on distinct neural representations. In line with this, we did not expect a delay-related decline in reinstatement. The derived object-specific similarity index, which is also subject to temporal autocorrelation, showed no significant effect of Session or Delay in any ROI (all p>0.059; see *Supplementary files 10 and 11*), supporting the specificity of the original reinstatement effect.

Second, we tested whether the observed group and delay effects might reflect global or nonspecific BOLD signal fluctuations by analyzing three control regions within the corpus callosum (genu, body, and splenium), where no memory-related reinstatement is expected. The LME models revealed no significant Group effects in any of the white matter ROIs (all p>0.426), indicating no difference between children and adults. Although we observed significant main effects of Session (all p<0.001), post hoc comparisons showed that these effects were driven by differences between the recent (day 0) and most remote (day 14) sessions. Crucially, the key contrasts of interest – recent vs. day 1 remote and day 1 remote vs. day 14 remote – were not significant (all p>0.080; see *Supplementary files 12 and 13*), in contrast to the robust decline observed in key ROIs for scene-specific reinstatement.

Finally, we assessed whether the observed reinstatement effects were specific to successful memory retrieval by examining item-based reinstatement for incorrectly remembered trials. This analysis revealed no session-related decline in any ROI, further supporting the interpretation that the reinstatement effects observed in correctly remembered trials are memory-related rather than driven by unspecific signal changes (see *Figure 7—figure supplement 1*).

Taken together, scene-specific reinstatement declined significantly for overnight compared to immediate memories and declined further after a 2-week delay across all ROIs. These results indicate that the main decrease in scene-specific neural reinstatement for successfully consolidated memories occurs already after a short overnight delay and continues with further decline after a longer, fortnight delay.

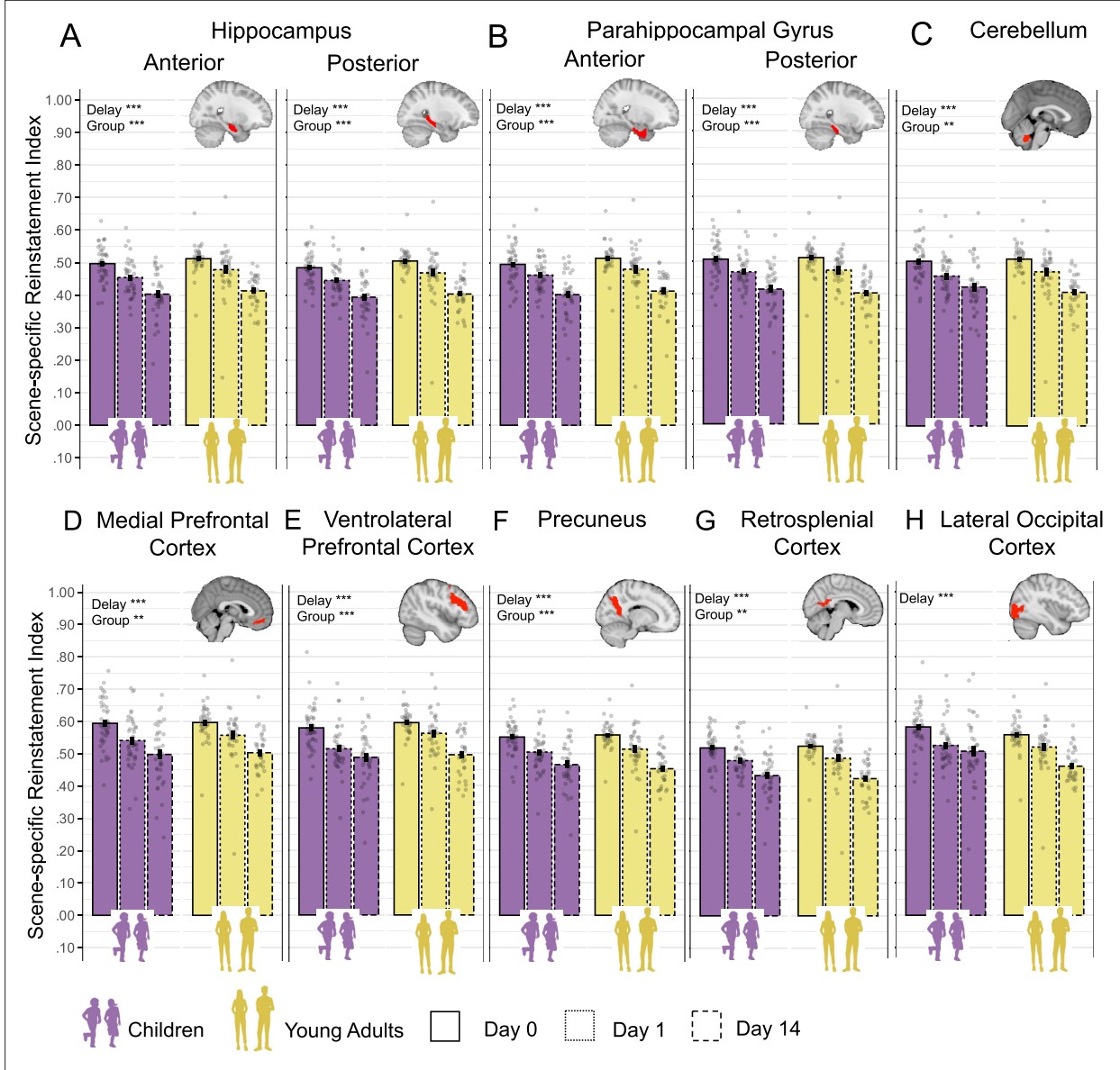

**Figure 7.** Corrected scene-specific neural reinstatement. Scene-specific neural reinstatement is defined as the difference between Fisher-transformed scene-specific and set-specific representational similarity. Scene-specific neural reinstatement index by group (children vs. adults) and session (day 0 – recent, day 1 – remote short delay, day 14 – remote long delay). Bars represent the mean reinstatement index for each session within each group, with error bars indicating standard error of the mean. Transparent dots show individual participant data points, jittered horizontally for visibility. The x-axis is grouped by group and displays (**A**) hippocampus anterior; (**B**) hippocampus posterior; (**C**) parahippocampal gyrus anterior; (**D**) parahippocampal gyrus posterior; (**E**) cerebellum; (**F**) medial prefrontal cortex; (**G**) ventrolateral prefrontal cortex; (**H**) precuneus; (**I**) retrosplenial cortex; (**J**) lateral occipital cortex. *p<0.05; **p<0.01; ***p<0.001 (significant difference). Error bars indicate standard error.

The online version of this article includes the following figure supplement(s) for figure 7:

**Figure supplement 1.** Object-specific neural reinstatement.

## Gist-like neural representations

Another way to evaluate the quality of neural representations during the post-cue fixation time window is to examine potential shifts in the ongoing balance between differentiated detailed ('verbatim') and generalized generic ('gist') memory, as described by the Fuzzy Trace Theory (**Brainerd and Reyna, 2002a**). Although our associative memory paradigm was designed to foster precise, detailed retrieval, it inherently also permits more generic, gist-like retrieval – e.g., some participants may recall 'a field' without its unique details (yielding a generic field representation), whereas others reinstate the full,

**Table 2.** Statistical overview of linear mixed effects (LME) model-based Sidak-corrected post hoc comparisons for scene-specific reinstatement analysis (based on LME model described in Supplementary file 9).

| | Model-based post hoc comparisons* | | | | | | | | |
| | YC >YA | | | Recent >Remote day 1 | | | Remote day 1>day 14 | | |
| ROI | b | $t_{(DF)}$ | p | b | $t_{(DF)}$ | p | b | $t_{(DF)}$ | p |
|---|---|---|---|---|---|---|---|---|---|
| HCa | −.071 | $-5.15_{(89)}$ | <0.001 | 0.040 | $4.35_{(162)}$ | <0.001 | 0.095 | $9.60_{(167)}$ | <0.001 |
| HCp | −0.068 | $-5.14_{(91)}$ | <0.001 | 0.040 | $4.29_{(162)}$ | <0.001 | 0.094 | $9.45_{(168)}$ | <0.001 |
| PHGa | −0.069 | $-4.75_{(90)}$ | <0.001 | 0.039 | $4.05_{(162)}$ | <0.001 | 0.098 | $9.62_{(167)}$ | <0.001 |
| PHGp | −0.055 | $-3.91_{(90)}$ | <0.001 | 0.040 | $3.77_{(178)}$ | <0.001 | 0.096 | $9.07_{(172)}$ | <0.001 |
| mPFC | −0.049 | $-2.94_{(92)}$ | 0.004 | 0.045 | $4.16_{(162)}$ | <0.001 | 0.093 | $7.91_{(169)}$ | <0.001 |
| vlPFC | −0.058 | $-3.84_{(93)}$ | <0.001 | 0.053 | $4.55_{(179)}$ | <0.001 | 0.089 | $7.79_{(169)}$ | <0.001 |
| CE | −0.044 | $-3.05_{(89)}$ | 0.003 | 0.046 | $3.97_{(166)}$ | <0.001 | 0.086 | $7.19_{(170)}$ | <0.001 |
| RSC | −0.041 | $-2.99_{(90)}$ | 0.003 | 0.039 | $3.72_{(162)}$ | <0.001 | 0.094 | $8.56_{(169)}$ | <.001 |
| PC | −0.047 | $-3.33_{(89)}$ | 0.001 | 0.044 | $4.15_{(165)}$ | <0.001 | 0.086 | $7.89_{(168)}$ | <.001 |
| LOC | −0.017 | $-1.09_{(103)}$ | 0.279 | 0.045 | $3.97_{(173)}$ | <0.001 | 0.083 | $7.07_{(174)}$ | <.001 |

*Notes*: Degrees of freedom were adjusted based on Kenward-Roger methods. p-Values were adjusted based on Sidak adjustment. YA – young adults; CH – children; ROI – region of interest; HCa – anterior hippocampus; HCp – posterior hippocampus; PHGa – anterior parahippocampal gyrus; PHGp – posterior parahippocampal gyrus; mPFC – medial prefrontal cortex; vlPFC – ventrolateral prefrontal cortex; CE – cerebellum; RSC – retrosplenial cortex; PC– precuneus; LOC – lateral occipital cortex; b – beta values; t – t-value; DF – degrees of freedom; p – p-value; CI – confidence interval; *p<0.05; **<0.01, ***<0.001 (significant difference).

specific features of the original scene. Accordingly, to quantify gist-like representations of the scenes sharing the same category (e.g. field, forest, etc.) during the fixation time window following the object cueing (see *Figure 1A*, Retrieval; *Figure 8*), we computed a gist-like representation index.

First, a within-category similarity index was computed by correlating the multivoxel patterns during the fixation time window for all correctly remembered scene pairs from the same category (i.e. field, water, housing, forest, infrastructure, indoor, farming), excluding self-correlations. Category exemplars were evenly and randomly distributed across runs, preventing clusters of temporally adjacent trials. By including only correctly recalled trials and building RSMs from both within-run and cross-run scene pairs, we substantially increased the number of independent pairwise comparisons in the RSA – and hence our sensitivity to detect effects. Next, a between-category similarity index was computed in the same way, but for scene pairs drawn from different categories. Finally, a gist-like representation index was defined as the difference between Fischer-transformed within- and between-category correlations (i.e. [within category_recent r – between category_recent r] and [within category_remote r – between category_remote r] for each session, *Figure 8*). Thus, the gist-like representation index reflects the extent to which neural patterns during the fixation window reactivate a generalized category representation (i.e, forest) – over and above any nonspecific similarity to scenes from other categories.

The nonzero values in this index reflect gist-like representation, as the similarity distance would be higher for pairs of trials within the same category, indicating more generic representation (e.g. during representation of scenes belonging to a category 'forest', participants may tend to recall a generic image of some forest without any specific details). In other words, the representation of a more generic, gist-like image of a forest across multiple trials should yield more similar neural activation patterns. Not significant gist-like representation would indicate that even within the same category, representation of specific scenes is sufficiently differential and rich in details, rendering them dissimilar (e.g. participants may tend to recall detailed image of forests: fall forest with yellow trees, dark pine-tree forest, light summer forest with young birch trees, etc.).

First, we aggregated the gist-like representation indices for recent items on day 1 and day 14, as there were no significant differences between sessions in ROIs (all p>0.231). Then we applied a one-sample permutation t-test to test for significance of all gist-like indices against zero in each ROI (for full overview, see *Supplementary file 14*). FDR-corrected values revealed that young adults showed only remote day 1 gist-like representation in LOC (p=0.024), while significant remote gist-like

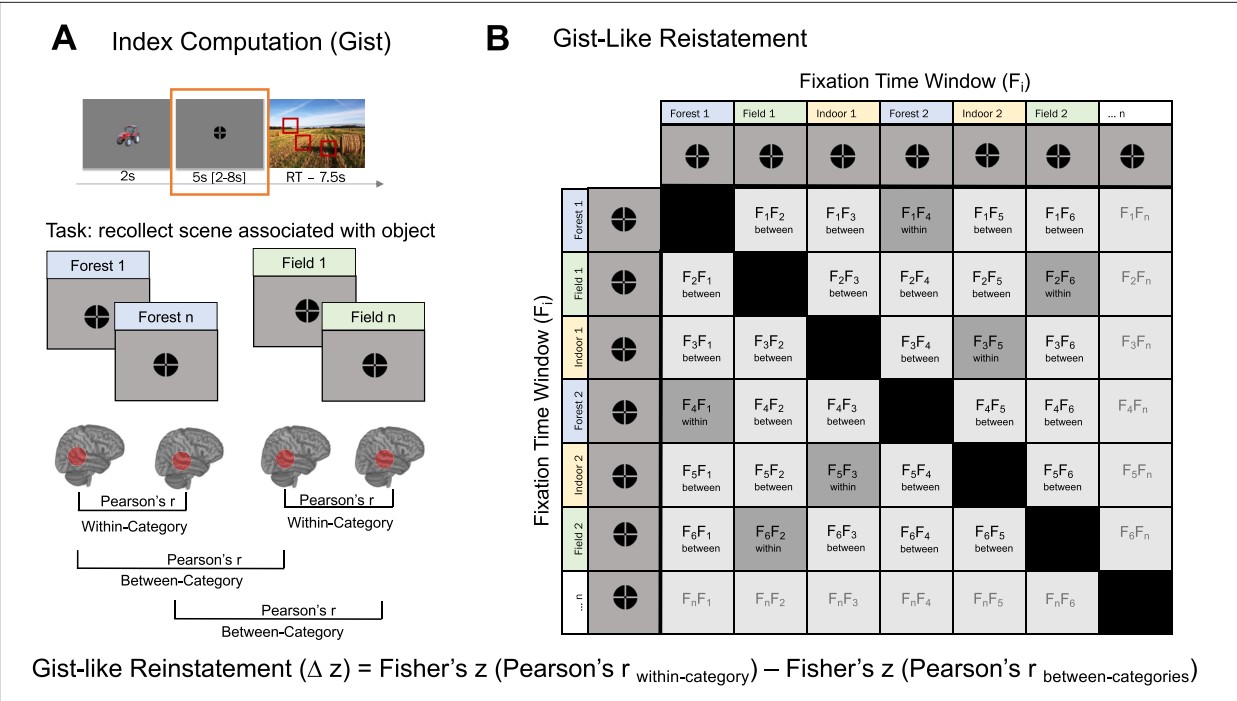

**Figure 8.** Representational similarity analysis. (**A**) Index computation (gist). A representational similarity index was computed by assessing the average similarity for fixation time window for within-category and between-category scenes separately for recent, remote (day 1), and remote (day 14) scenes based on both within-run and cross-run comparisons. The diagonal (similarity of fixation time window with itself) was excluded from the analysis. (**B**) Gist-like representation. A gist-like representation index was computed by assessing the average similarity in fixation time window for the same-category pairs and subtracting from it the any-other-category pairs. S – scene time window; F – fixation time window; r – Pearson's correlation index. Δz – difference between two Fisher-transformed r values.

representation was observed in children on day 1 in precuneus (p=0.044) and LOC (p=0.024), and on day 14 in the mPFC (p=0.013) and vlPFC (p=0.007). Following this, we further analyzed group differences separately for each ROI that showed significant gist-like representation, controlling for the BOLD mean activation in each ROI during corresponding sessions.

Second, we investigated the time-dependent change in gist-like representation in ROIs that showed gist-like representation. For the mPFC (*Figure 9A*), we observed a significant main effect of *Group*, $F_{(1,244)} = 6.55$, p=0.011, $\omega^2 = 0.02$, indicating significantly higher gist-like representation in the mPFC in children compared to young adults, b = 0.011, $t_{(82)} = 2.52$, p=0.013. Additionally, a significant main effect of *Session*, $F_{(1,244)} = 3.89$, p=0.022, $\omega^2 = 0.02$, indicated higher remote day 14 compared to remote day 1 gist-like representation, b = 0.014, $t_{(180)} = 2.64$, p=0.027. For the vlPFC (*Figure 9B*), we observed a significant effect of *Session*, $F_{(1,174)} = 4.45$, p=0.013, $\omega^2 = 0.04$, indicating higher remote day 14 gist-like representation compared to recent one, b = 0.013, $t_{(195)} = 2.91$, p=0.012. A significant *Session × Group* interaction, $F_{(1,167)} = 3.04$, p=0.05, $\omega^2 = 0.02$, highlighting significantly higher remote day 14 gist-like representation in children compared to young adults, b = 0.017, $t_{(249)} = 2.52$, p=0.037. Neither LOC nor precuneus showed any significant main or interaction effects (all p>0.062; *Figure 9C and D*). Taken together, only the child group showed gist-like representation in the medial and ventrolateral prefrontal brain regions that was significantly higher during retrieval of long-delay remote memories, indicating a reorganization of memory representations in children.

## Neural-behavioral correlations

Further, we also explored whether over time, recent, short- and long-delay scene-specific reinstatement and gist-like representations are beneficial or detrimental for memory performance by correlating the indices with memory retention rates. We derived, with a PLSC analysis, latent brain pattern across implicated ROIs that share the most variance with delay-related variations in memory accuracy.

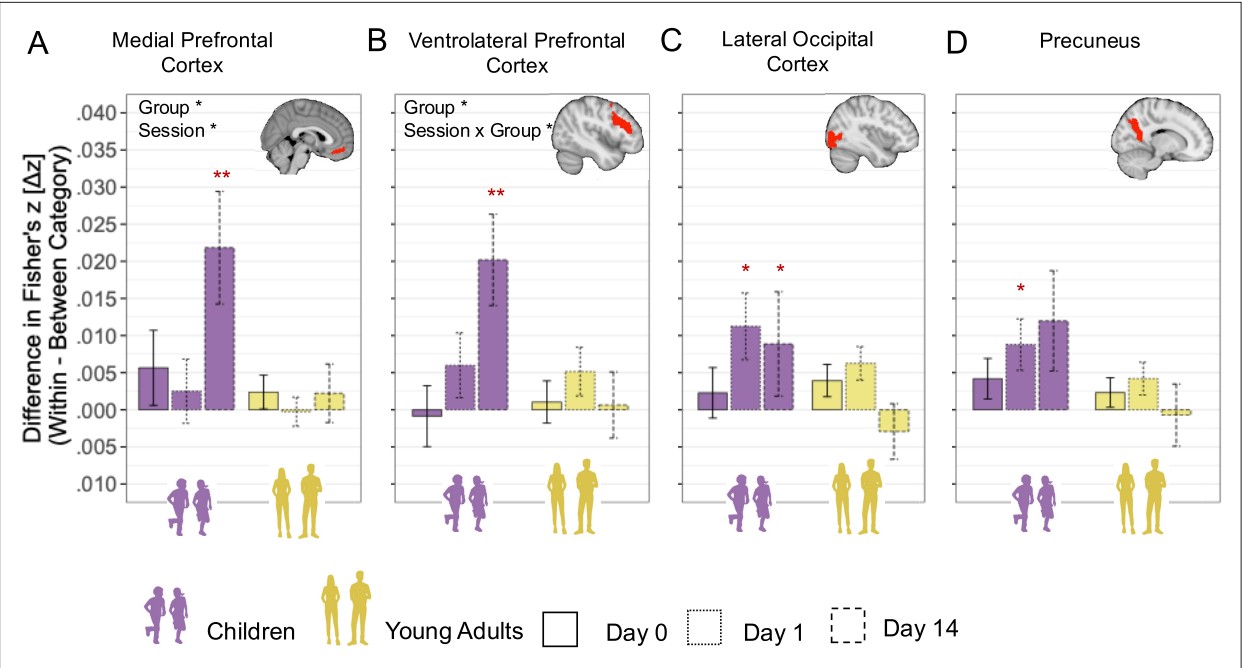

**Figure 9.** Gist-like representations. Bar plots show mean gist-like representation index (difference in Fisher's z-transformed (Δz) similarity: *within-category – between-category*) in each group (children, young adults) and session (day 0, day 1, day 14), computed from combined within- and cross-run comparisons. Error bars indicate ± 1 standard error of the mean. A representation value above zero (denoted by red asterisks) reflects greater neural pattern similarity during fixation time window between items from the same category than across categories. Bar positions are grouped by age group (x-axis). Session-specific estimates (days 0, 1, 14) are differentiated by line of bar border. (**A**) Medial prefrontal cortex; (**B**) ventrolateral prefrontal cortex; (**C**) lateral occipital cortex; (**D**) precuneus; *p<0.05; **p<0.01; ***p<0.001 (significant difference; false discovery rate (FDR)-corrected for multiple comparisons); nonsignificant difference was not specifically highlighted.

## Neural-behavioral correlations (scene-specific reinstatement)

For the scene-specific reinstatement, all predefined ROIs in both age groups were included. With further bootstrapping, we identified Z-score estimates of robustness (larger/smaller than ± 1.96 (a<0.05)) of the components within the multivariate brain profile.

First, for the recent delay (30 min after learning), the permutation test of significance resulted in a single LV that robustly represents the association of scene-specific reinstatement brain profile and memory accuracy across both age groups (*Figure 10B*, r=0.293, p=0.007). Higher recent memory accuracy was robustly associated with greater scene-specific reinstatement in the anterior PHG (Z-score=3.010, r=0.819), posterior PHG (Z-score=2.575, r=0.367), anterior HC (Z-score=2.629, r=0.3713), posterior HC (Z-score=3.009, r=0.417), and precuneus (Z-score=2.206, r=0.318) across age groups.

Second, for short delay, the permutation test of significance resulted in a single LV that robustly represents the association of scene-specific reinstatement brain profile and memory accuracy across both age groups (*Figure 10B*, r=0.339, p=0.0017). Higher memory accuracy was robustly associated with greater scene-specific reinstatement in the anterior PHG (Z-score=2.885, r=0.371), posterior PHG (Z-score=2.597, r=0.342), anterior HC (Z-score=3.126, r=0.399), posterior HC (Z-score=2.844, r=0.375), vlPFC (Z-score=2.434, r=0.317), mPFC (Z-score=2.753, r=0.333), and LOC (Z-score=2.176, r=0.298) across age groups.

Third, for long delay, the permutation test of significance resulted in a single LV that robustly represents the association of scene-specific reinstatement brain profile and memory accuracy across both age groups (*Figure 10C*, r=0.455, p=<0.001). Higher memory accuracy was robustly associated with greater scene-specific reinstatement in the anterior PHG (Z-score=6.213, r=0.414), posterior PHG (Z-score=4.810, r=0.334), anterior HC (Z-score=5.353, r=0.389), posterior HC (Z-score=4.707, r=0.354), precuneus (Z-score=3.404, r=0.281), vlPFC (Z-score=3.291, r=0.266), RSC (Z-score=3.72, r=0.293), LOC (Z-score=3.288, r=0.282), and cerebellum (Z-score=3.842, r=0.308) across age groups.

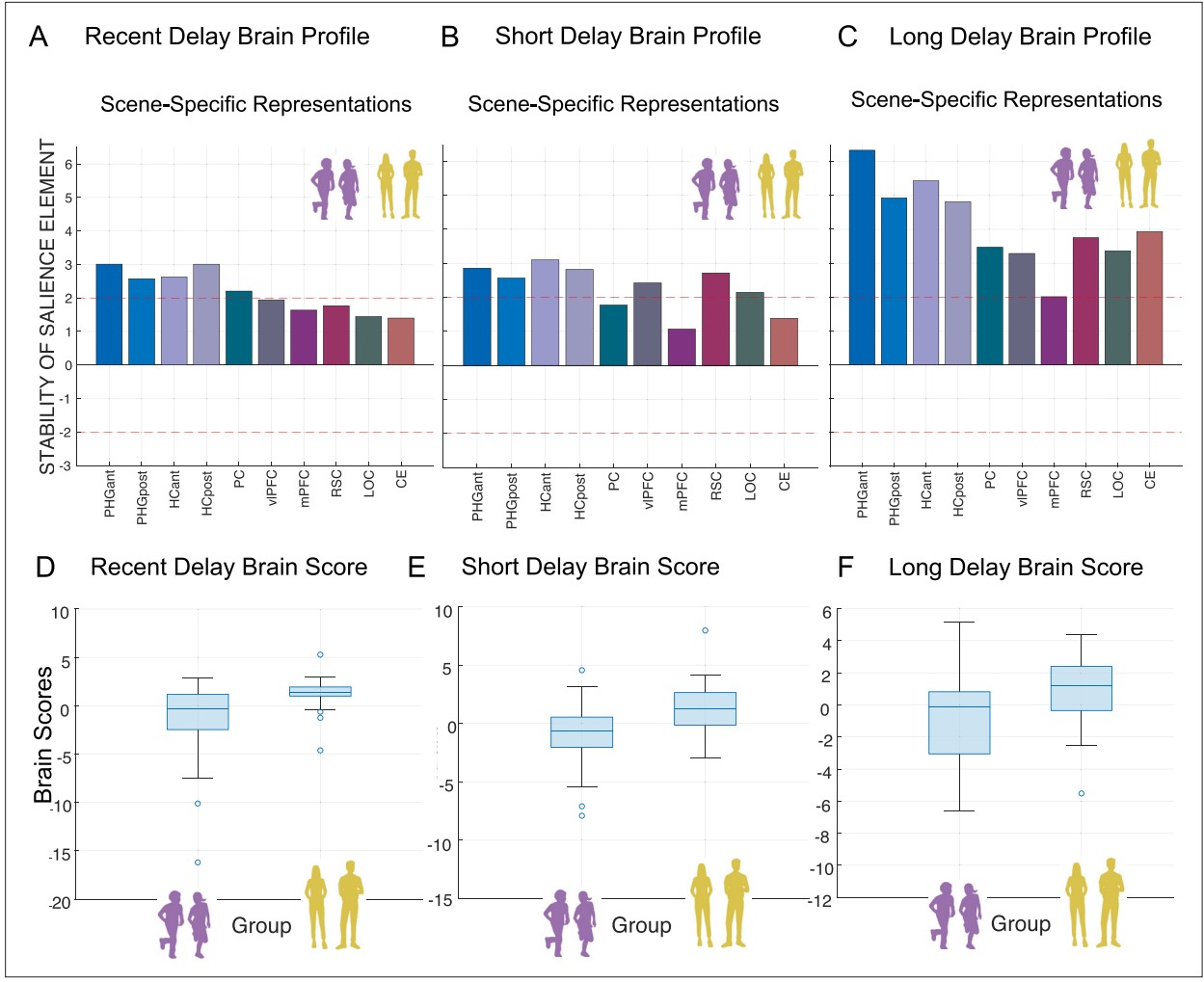

**Figure 10.** Multivariate short- and long-delay brain profiles of scene-specific reinstatement are associated with variations in memory accuracy. (**A**) *Recent delay brain profile*. Latent variable weights or saliences for each region of interest (ROI) build up one latent variable that expresses a composite immediate-delay scene-specific reinstatement brain profile. (**B**) *Short-delay brain profile*. Latent variable weights or saliences for each ROI build up one latent variable that expresses a composite short-delay scene-specific reinstatement brain profile. (**C**) *Long-delay brain profile*. Latent variable weights or saliences for each ROI build up one latent variable that expresses a composite long-delay scene-specific reinstatement brain profile. Stability of salience elements is defined by Z-scores (depicted as red lines: a value larger/smaller than ± 1.96 is treated as reliably robust at (a<0.05)). The bar plot shows the bootstrap ration (BSR) values for the latent variable, reflecting the stability of the relationship between brain scene-specific neural reinstatement and memory performance. (**D**) Recent delay brain scores. (**E**) Short-delay brain scores. (**F**) Long-delay brain scores. Each box represents the distribution of brain scores within a group, with central lines indicating the median and boxes showing the interquartile range. Whiskers represent the full range of non-outlier values. *Note:* PHGa – anterior parahippocampal gyrus; PHGp – posterior parahippocampal gyrus; HCa – anterior hippocampus; HCp – posterior hippocampus; PC – precuneus; vlPFC – ventrolateral prefrontal cortex; mPFC – medial prefrontal cortex; RSC – retrosplenial cortex; LOC – lateral occipital cortex; CE – cerebellum; r – Spearman's rank order correlation index.

Identified brain profiles across groups suggest shared patterns between neural mean signal differences in differential sets of ROIs and memory accuracy are consistent across children and adults. As this approach optimizes for consistent covariance pattern across the full sample, it does not test for group-specific profiles. When conducting the same PLS models within each group, no stable latent profile emerged (all p>0.069). The reduced within-group sample may have affected the bootstrap-based stability. To address this, we explored whether groups differ in their expression of the common LV (i.e. brain scores). This analysis revealed that children showed significantly lower brain scores than adults both in immediate delay, $t_{(85)} = -3.971$, p=0.0001 (*Figure 10C*), in short delay, $t_{(81)} = -2.973$, p=0.004 (*Figure 10C*), and long delay, $t_{(70)} = -2.659$, p=0.01 (*Figure 10D*), suggesting that while the brain-behavior profile was shared, its expression varied by group.

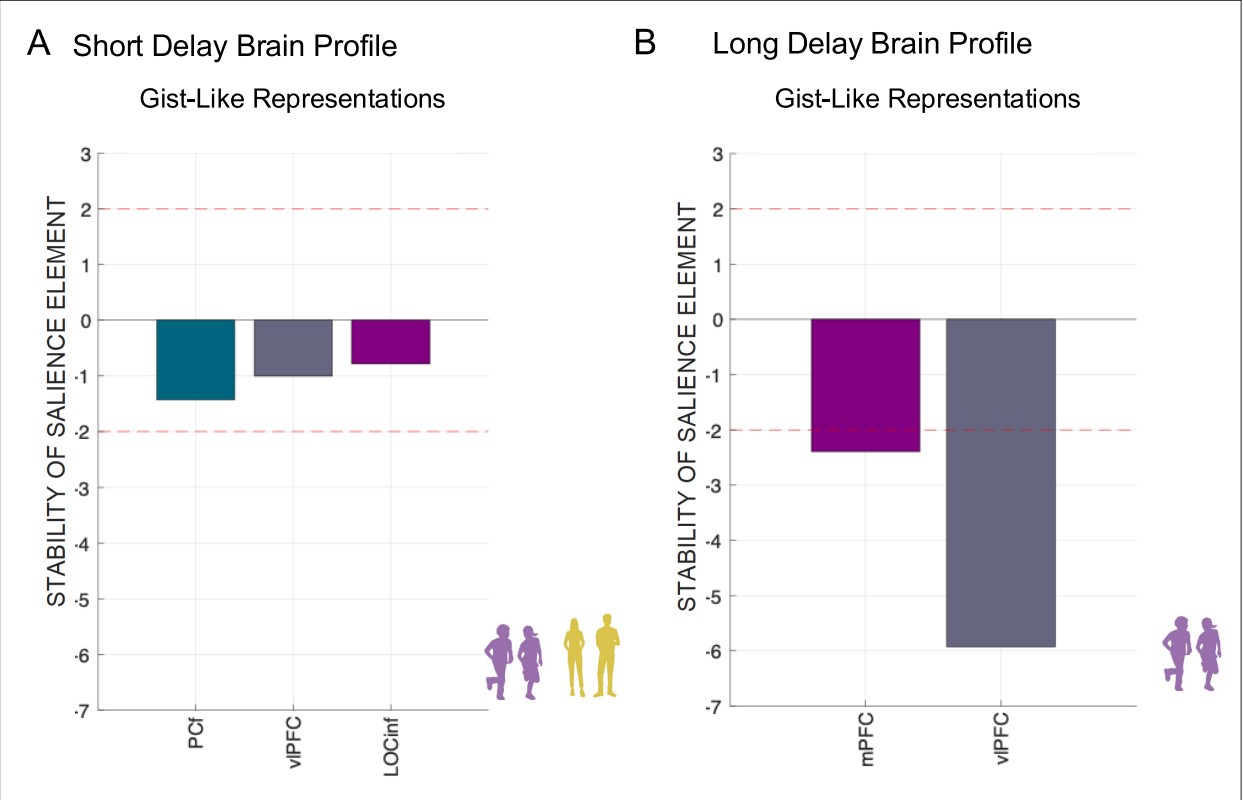

**Figure 11.** Multivariate short- and long-delay brain profiles of gist-like representations are associated with variations in memory accuracy. (**A**) *Short-delay brain profile.* Latent variable weights or saliences for each region of interest (ROI) build up one latent variable that expresses a composite short-delay gist-like representations brain profile across age groups. (**B**) *Long-delay brain profile.* Latent variable weights or saliences for each ROI build up one latent variable that expresses a composite long-delay gist-like representations brain profile in child group. The bar plot shows the bootstrap ration (BSR) values for the latent variable, reflecting the stability of the relationship between brain gist-like neural representations and memory performance. Stability of salience elements is defined by Z-scores (depicted as red lines: a value larger/smaller than ± 1.96 is treated as reliably robust at (a<0.05)). *Note:* vlPFC – ventrolateral prefrontal cortex; mPFC – medial prefrontal cortex; LOC – lateral occipital cortex; r – Spearman's rank order correlation index.

These results underscore the importance of scene-specific reinstatement in positively contributing to memory performance for detailed associative information both in children and in adults.

## Neural-behavioral correlations (gist-like representations)

For gist-like representations, we included only those ROIs that showed significant gist-like representations (i.e. for short delay: LOC and precuneus across both age groups; for long delay: vlPFC and mPFC in child group).

First, for a short delay, the permutation test of significance did not result in a single LV that robustly represents the association between brain profile of gist-like representations and memory accuracy in children (*Figure 11A*, r=0.221, p=0.065). Second, for long delay, the permutation test of significance resulted in a single LV that robustly represented the association of gist-like representations brain profile and memory accuracy in the child group (*Figure 11B*, r=0.516, p=0.0014). The higher long delay day 14 memory accuracy was robustly associated with lower gist-like representations in the mPFC (Z-score=−2.396, r=−0.498), and vlPFC (Z-score=−5.918, r=−0.876) in children.

The significant negative association between long delay gist-like representations in both prefrontal brain regions and memory accuracy observed in children underscores that gist-like representations were detrimental to memory performance for detailed associative information in children in long delay. Taken together, more differentiated detail-rich neural reinstatement was related to better memory retrieval in both children and young adults. On the other hand, more gist-like neural representations, uniquely found in children, were related to worse memory retrieval.

## Discussion

In the present study, we investigated system-level memory consolidation of object-location associations across three delays: immediately after learning, after one night of sleep (short delay), and after 2 weeks (long delay). We tracked changes in neural activation and multivariate activation patterns over time, comparing 5- to 7-year-old children and young adults. Our main findings are as follows: (i) Children showed greater decline in memory retention compared to young adults at both short and long delays. (ii) Regarding neural upregulation – reflected as the mean difference in activation between remote and recent retrieval – the two age groups showed distinct changes over time. Young adults exhibited an increase in neural upregulation over time in the posterior PHG, cerebellum, and LOC, and an overall higher neural upregulation in the vlPFC compared to children. In contrast, only children showed a decrease in neural upregulation over time in the RSC and overall greater neural upregulation in the mPFC than adults. Distinct neural upregulation profiles with specific sets of brain regions were related to immediate, short, and long delay memory accuracy. (iii) Using RSA, we found that differentiated scene-specific reinstatement declined over time in both age groups. Notably, more generic gist-like representations were observed only in children, particularly in medial and ventrolateral prefrontal regions. Importantly, higher scene-specific reinstatement was related to better memory retention in both age groups, whereas greater gist-like representations were related to lower memory retention at the long delay only in children.

Our study extends previous adult-based findings and, for the first time, demonstrates that the memory retrieval after consolidation in children is accompanied by differential patterns of neural activation of some of the core retrieval regions, attenuated neural reinstatement of detailed memories, and the emergence of generic gist-like representations. These findings suggest that adults leverage mature neural memory systems and extensive existing knowledge base to encode and consolidate new complex information with detailed accuracy. In contrast, children – whose neural system is still developing – may rely more on consolidating gist information as a foundational scaffold for their still sketchy knowledge base, possibly at the cost of episodic detailedness. At this developmental stage, focusing on precise detail may not be yet prioritized (*Keresztes et al., 2018*). Each of these findings is discussed in detail in the following sections.

### Less robust short- and long-delay memory retention in children compared to young adults

Our findings indicate that preschool children (5–7 years of age) can encode and retain complex associative and highly contextualized information successfully over extended periods following adaptive learning. However, their overall learning and retrieval performance was lower compared to young adults. Moreover, children exhibited more pronounced declines in memory retention over both short and long delays for correctly learned information, suggesting less robust memory consolidation compared to young adults.

Regarding learning, children needed more cycles to memorize object-scene associations and showed lower learning performance after initial strategic encoding compared to young adults. Although we did not expect children to reach adult-like learning rates given the complex and associative nature of the task (*Pressley et al., 1981*), we aimed to maximize children's learning capacities through adaptive learning procedures. To support this, attention allocation and motivation during encoding and learning were closely monitored through the constant presence of the experimenter and the use of feedback questionnaires. In addition, all participants underwent training in elaborative encoding strategies to support later retrieval.

Overall, our findings on learning suggest that children were less adept than adults at utilizing strategic control over encoding, such as creating and maintaining stories to aid their retrieval. This is consistent with previous literature, showing continuous improvement in children's ability to use elaborative strategies between ages 4 and 8 (*Bjorkund et al., 2009*; *Crowley and Siegler, 1999*; *Pressley, 1982*). Additionally, children at this age may experience difficulties in controlling (*Ruggeri et al., 2019*) and effectively using their learning strategies over time (*Brod, 2021*; *Shing et al., 2010*). Observed lower learning rates may also be attributed to less efficient binding processes in children compared to young adults (*Shing et al., 2010*; *Sluzenski et al., 2006*). Although we included only stimuli from the primary school curriculum to reduce age differences in knowledge availability, ongoing maturation of the memory brain network in 5- to 7-year-old children may have attenuated their benefit from

pre-existing knowledge and memory aid through strategic elaboration (*Ghetti and Bunge, 2012*; *Lenroot and Giedd, 2006*; *Nishimura et al., 2015*; *Ofen, 2012*; *Shing et al., 2008*). Despite these challenges, 5- to 7-year-old children were capable of learning complex associative information to a considerable extent, which aligns with their ability to gradually accumulate world knowledge (*Bauer, 2021*; *Brod and Shing, 2022*; *Wagner, 2010*).

Concerning memory consolidation, our results are in line with previous studies that reported worse memory retention for associative information in school-age children compared to adults (*Østby et al., 2012*; *Schommartz et al., 2023*; *Schommartz et al., 2024*). On the other hand, our results are not in line with sleep-related beneficial effects on mnemonic performance of 7- to 12-year-old children after one night delay (*Peiffer et al., 2020*; *Wang et al., 2018*) that were shown for novel stimuli not related to any prior knowledge (in the sense of arbitrary stimuli). As we opted for well-learned information that should allow for rapid creation of new schemas or integration of new associations into already existing schemas, our findings indicate that the beneficial role of sleep on memory consolidation in children compared to adults may not apply for repeatedly and strategically learned information. Deliberate learning is potentially more advantageous for subsequent memory retention in young adults, as this information may be integrated into pre-existing knowledge structures faster (*van Kesteren et al., 2013*), with higher strategic control of memories upon retrieval and therefore greater accessibility of consolidated memories (*Fandakova et al., 2017*; *Gaudreau et al., 2001*). Taken together, our findings indicate that compared to young adults, 5- to 7-year-old children exhibit less robust memory consolidation for well-learned information, suggesting an overall reduced ability to retain detailed memories in children.

To better understand whether observed age-related differences in memory performance reflect changes or differences in memory strength per se, we conducted exploratory analysis using drift diffusion modeling (DDM; *Lerche and Voss, 2019*; *Palada et al., 2016*; *Ratcliff et al., 2011*; *Ratcliff et al., 2012*; *Ratcliff and McKoon, 2008*; *Zhou et al., 2021*). DDM estimates the underlying cognitive mechanisms driving decision-making by jointly modeling accuracy and reaction time, offering a nuanced index of memory strength via the drift rate parameter. This approach allowed us to quantify trial-wise memory accessibility beyond raw performance measures (see *Supplementary file 15* for detailed overview). The results revealed that children had significantly lower drift rates compared to young adults across all delays, indicating slower and noisier evidence accumulation – possibly due to weaker memory representations. As drift rate closely correlates with memory accuracy (*Ratcliff et al., 2011*), our findings on the memory strength align with those on memory accuracy during retrieval in both age groups. Crucially, drift rate decreased systematically from recent to remote conditions in both groups, but this decline was steeper in adults. This finding suggests that while adults started with stronger memory traces, these detailed, differentiated traces were also more susceptible to decay. In contrast, children's already lower drift rates remained relatively more stable over time. This pattern points to qualitative group differences in how memories are initially encoded and subsequently consolidated. The DDM analysis helped us to dissociate group differences in retrieval dynamics from surface-level accuracy. It revealed how response patterns varied as a function of memory strength across time, supporting the conclusion that developmental differences in memory consolidation cannot be fully explained by initial performance alone. Our neural findings suggest that differences in functional engagement of the retrieval network and the characteristics of memory representations being created and retained may underlie the observed behavioral differences.

## Differential upregulation of remote > recent neural activation over time in children in comparison to young adults

Analyses of neural upregulation (i.e. remote > recent difference in neural activation) over time allowed us to control for the effects of rapid consolidation during repeated learning, while examining changes in short- or long-delay neural activation (*Brodt et al., 2016*; *Brodt et al., 2018*; *Yu et al., 2022*). First, we observed increased upregulation in the vlPFC over time in both age groups, with young adults showing greater vlPFC upregulation overall. Furthermore, we observed stable greater upregulation in the mPFC over time in children. On the one hand, this may indicate a stronger strategic control over retrieval processes in young adults, given the vlPFC's role in strategic remembering and retrieval of stored memories (*Badre and D'Esposito, 2009*; *Kuhl et al., 2012*). Such vlPFC upregulation was beneficial for memory retention. On the other hand, the observed higher mPFC upregulation

in children may reflect less efficient suppression of the default mode network during effortful memory search (*Chai et al., 2014*; *Fair et al., 2008*). Over time, cognitive control during memory retrieval may increase as it requires greater effort to recollect elaborative stories to remember the associated spatial context. Strategic control over memories may be present but less pronounced in children due to the more protracted developmental trajectories of PFC maturation (*Ghetti and Bunge, 2012*; *Gogtay et al., 2004*; *Ofen, 2012*; *Shing et al., 2010*).

In addition, our results indicate that a more pronounced schema-related retrieval may be mediated by mPFC to a greater extent in children than in young adults. This extends previous findings on the involvement of mPFC in structured and schema-related retrieval of long-term memories (*Takashima et al., 2006*; *Yamashita et al., 2009*) to a child developmental cohort. Higher mPFC upregulation in long delay was negatively related to memory performance, suggesting that it is detrimental to the retention of detailed associative memories. In addition, it may suggest consolidation-related transformation of memory traces into less differentiated, more generic, and gist-like memories (*Gilboa and Marlatte, 2017*; *Gilboa and Moscovitch, 2021*).

Second, in other constituents of the recollection network (*Ranganath and Ritchey, 2012*), we observed increased upregulation from short to long delay in the posterior PHG and overall lower upregulation in precuneus (i.e. remote > recent) in young adults. As young adults showed higher memory retention rates for more detail-rich information, this superior memory may be mediated by higher upregulation in the posterior PHG involved in contextual associations and scene memory (*Aminoff et al., 2013*). In children, PHG undergoes prolonged maturation (*Golarai et al., 2007*), and its increased functional maturation is related to long-term scene recollection (*Chai et al., 2010*). In addition, higher mnemonic distinctiveness of more recent memories (i.e. higher retention rates for detailed information) may also be mediated by RSC and precuneus activation profiles, as these regions are involved in mnemonic vividness, spatial, and associative memory as indicated by other findings from immediate delays (*Brodt et al., 2016*; *Hebscher et al., 2019*; *Mitchell et al., 2018*; *Richter et al., 2016*; *Tambini and D'Esposito, 2020*; *Vann et al., 2009*). Moreover, lower precuneus upregulation after a short delay and higher RSC upregulation after a long delay was related to better memory performance. Time-related decrease in activation of the posterior brain regions observed in children aligns with previous findings of *Demaster and Ghetti, 2013*, who reported that the engagement of parietal regions during recollection of correct memories increased with age in 8- to 11-year-old children. Therefore, the continuing maturation of parietal regions in 5- to 7-year-old children (*Sowell et al., 2002*) presumably contributes to the age-related differences in consolidation-related upregulation observed in these regions.

Third, the observed increase in neural upregulation from short to long delay in the LOC and the cerebellum in young adults is also in line with previous findings suggesting that the cerebellum supports rapid cortical storage of memory traces after repeated exposure – even after 24 hr (*Stroukov et al., 2022*) – and shows upregulation of neural activation for long-term episodic memory retrieval (*Andreasen et al., 1999*). Concerning the LOC, prior studies have linked HC-LOC activation to scene-related associative memory consolidation (*Tambini et al., 2010*) and to human object recognition (*Grill-Spector et al., 2001*). Moreover, the network comprising the angular gyrus and LOC has been shown to enhance overnight retention of schema-related memories in young adults (*van der Linden et al., 2017*). Consistent with these findings, we also observed that greater LOC upregulation after a long delay was related to better memory performance. The more pronounced upregulation from short to long delay in these regions in adults suggests that the cerebellum and LOC support long-delay memory retention – one that appears to be functionally immature in middle childhood.

Finally, our findings on age group and delay-invariant activation in the anterior HC and PHG, and posterior HC during the retrieval of detail-rich memories (i.e. the exact location of an object within a scene) are in line with *Nadel and Moscovitch, 1997*, who postulated that the HC formation and related structures remain involved in detail-rich memories upon their retrieval, irrespective of memory age. For example, *Du et al., 2019*, reported stable HC involvement during retrieval of associative memory across delays of 1 day, 1 week, and 1 month in young adults. *Tanrıverdi et al., 2022*, also demonstrated that post-encoding coactivation of HC and cortical brain regions may lead to experience-dependent change in memories, highlighting the importance of HC involvement during consolidation. Furthermore, the absence of age-related differences in HC and anterior PHG involvement is also in line with developmental studies that have reported the relative maturity of the HC in

middle childhood (*Keresztes et al., 2017*; *Lee et al., 2014*; *Shing et al., 2010*), which is concomitant with an improvement in the ability to bind event features together into a coherent representation around the age of 6 years (*Sluzenski et al., 2006*). Specifically, our finding on HC engagement being robust in children and adults extends the Multiple Trace Theory and the Trace Transformation Theory to a child developmental cohort (*Moscovitch and Gilboa, 2022*; *Nadel et al., 2000*). Taken together, the similar engagement of medial-temporal cortex over time in children and adults indicated that the retrieval of well-learned detail-rich memories is mediated by these brain structures already in middle childhood.

To summarize, we provide novel evidence about changes in neural upregulation for successfully consolidated memories over short and long delay, relative to immediately learned memories. While children exhibited adult-like stable neural activation for recent and remote memories in medial-temporal brain regions, young adults relied more on prefrontal, occipital, cerebellar, and parietal brain regions over time, compared to more pronounced reliance on medial prefrontal regions in children. Adults show more mature neocortical consolidation-related engagement, resulting in stronger and more durable detailed memories over time, while in children, immature neocortical engagement may lead to consequent reduction in memory retention of detailed memories.

## Reduced scene-specific reinstatement over time in children and young adults

We found that scene-specific reinstatement decreased over time both in children and in young adults, aligning with delay-related decrease in memory retention. Additionally, higher scene-specific neural reinstatement was related to better memory performance in short and long delays in both age groups.

Our findings contribute to the memory consolidation literature by demonstrating that scene-specific neural reinstatement observed in neocortical, medial temporal, and cerebellar brain regions supports reinstatement of detailed specific contextual memories. This observation is consistent with the Contextual Binding Theory (*Yonelinas et al., 2019*), which posits that stronger reinstatement of contextual details can enhance memory retention. The similar decay of these processes over time in both children and adults suggests that the basic mechanisms of contextual binding are present early in development. Additionally, in line with the Trace Transformation Theory (*Moscovitch and Gilboa, 2022*), our findings suggest that reinstatement patterns continuously transform over time. This transformation, observed across all considered memory-related regions, indicates a consistent and systematic consolidation-related reshaping of the unique scene-specific memory representations over time (*Chen et al., 2017*).

Our findings on scene-specific reinstatement align with and add to the previous literature that shows reliable reinstatement of unique events. For example, our findings align with the effects observed by *Masís-Obando et al., 2022*, for the immediate recall of story details in key memory regions. Consistent with *Oedekoven et al., 2017*, our results show that memory representations for unique events can be reliably detected through scene-specific reinstatement even after extended delays. Furthermore, we build on *Guo and Yang, 2023*, by demonstrating how specific ROI-related profiles of neural reinstatement during retrieval correlate with long-term memory retention. Unlike *Oedekoven et al., 2017*, who reported no time-related differences in reinstatement effects and used the same video clips for immediate and delayed recall – which could have inadvertently reinforced memory through reactivation – our study employed unique stimulus sets for each retrieval session, preventing any reconsolidation of mnemonic representations. This approach revealed a significant attenuation of reinstatement patterns after an overnight delay, which further diminished after 2 weeks, highlighting the importance of intentional reactivation for maintaining the specificity of neural reinstatement.

Our findings indicate similar patterns of scene-specific neural reinstatement between children and young adults. Building on *Fandakova et al., 2019*, who found similar distinctiveness of neural representations during encoding in 8- to 15-year-old children and adults, our results suggest that this similarity extends to younger ages, showing comparable distinctiveness of neural representations for unique memories from middle to late childhood and early adolescence. Additionally, our research supports the presence of delay-related change in scene-specific reinstatement in 5- to 7-year-old children, albeit at a lower level compared to adults, aligning with and extending previous studies (*Benear et al., 2022*; *Cohen et al., 2019*; *Golarai et al., 2015*), which demonstrated reliable mnemonic reinstatement for visual input (i.e. faces, movie clips) in 5- to 11-year-old children. Furthermore, we extend

these findings by showing that successful long-term memory retrieval is associated with more differentiated neural reinstatement in both children and young adults, indicating similar mechanisms of detail-rich memory consolidation present as early as 5 to 7 years.

Our results indicate that higher scene-specific neural reinstatement over time correlated with better memory retention in both children and adults. This is in line with the neural fidelity hypothesis (*Xue, 2018*), suggesting that more similar neural reinstatement reflects less noisy representations of mnemonic information. Convergent evidence showed that higher fidelity of neural representation across study episodes leads to successful memory (*Xue et al., 2010*; *Xue et al., 2013*). Similarly, *Masís-Obando et al., 2022*, reported that more specific neural representations predicted subsequent memory performance in young adults.

Of note, our study design, which resulted in temporal autocorrelation in the BOLD signal between memory retrieval (i.e. fixation time window) and scene observation and response (i.e. scene time window), was consistent across all three delay windows. Since the retrieval procedure remained unchanged over time and was similarly influenced by temporal autocorrelations in addition to several control analyses, we attribute our RSA findings to differences in reinstatement between recent and remote trials. Given that the scene time window for the 3AFC task was constant, the brain signals should exhibit similar perception-based but variably memory-based patterns across all delays.

Furthermore, all items, regardless of retrieval delay, underwent extensive learning and showed successful consolidation, as evidenced by correct recall. This suggests that both the fixation and scene time windows engaged memory-related neural processes. According to *Brodt et al., 2018*; *Brodt et al., 2016*, rapid consolidation-related neural reorganization can occur immediately after learning, indicating that even during recent retrieval, scenes are processed in a memory-oriented manner. Additionally, during the scene time window, participants engaged in retrieval by selecting the correct object location within the scene. Thus, while the scene time window involved perceptual processing, its impact is consistent across all items due to uniform exposure to repeated learning, making them equally familiar to participants. Although our paradigm per se cannot arbitrate between perception-based and memory-based nature of retrieval during scene presentation, our exploratory univariate analysis during the scene presentations time window (see *Figure 4—figure supplement 2* and *Supplementary file 18*) revealed higher neural engagement in the key memory regions with passing time, supporting memory-related processing during the scene time window.

Taken together, our findings provide novel evidence that although children exhibit more attenuated scene-specific reinstatement compared to young adults, the consolidation-related decrease in differentiated reinstatement follows similar patterns as in adults. This highlights that despite less robust memory consolidation and lower memory strength, children's neural transformations of distinct memories over time may share the same mechanisms as adults, with scene-specific reinstatement proving beneficial for memory retention in both groups.

## Unique gist-like representations in children

The results showed that only children demonstrated the emergence of generic gist-like representations in medial and ventrolateral prefrontal brain regions during successful long-delay retrieval. Furthermore, greater long-delay gist-like representations were associated with poorer long-delay memory accuracy in children. With these findings, we provide the first neural empirical evidence to support the Fuzzy Trace Theory (*Brainerd and Reyna, 2002a*; *Reyna and Brainerd, 1995*), showing neural reorganization of memory representations in children.

The Fuzzy Trace Theory aims to characterize the shifts in ongoing balance between precise, detailed 'verbatim' memory and more generalized, simplified 'gist' memory (*Brainerd and Reyna, 2002a*) from a developmental perspective. Our associative object-location task allowed the investigation of these 'dichotomies' as it was aimed to cultivate detailed, precise memories for retrieval. Simultaneously, it enabled generalization by creating more generic representations due to the presence of related category-based information. Adults were able to build upon solid pre-existing knowledge by embellishing them with details and integrating them into these structures. Children, in contrast, with their sparser knowledge, may have focused more on solidifying the structure with overlapping information. Aligning with the Fuzzy Trace Theory, our results suggest that reliance on gist-like memory representations is less effective for long-term retention of complex associative information compared to detailed verbatim memory, which seems to be characteristic of adults.

We also observed short-delay gist-like representations in posterior brain regions: the precuneus in children and the LOC in both age groups. These representations were not directly related to memory accuracy. The LOC is involved in object and scene recognition (*Golarai et al., 2007*; *Grill-Spector et al., 2001*) and has been shown to participate in schema-related consolidation or in durable but less specific memories (*van der Linden et al., 2017*). Its involvement in gist-like representations across age groups suggests occipital areas already engage in some degree of categorical abstraction, especially when stimuli share common visual or contextual features. In contrast, the additional precuneus involvement in children might reflect broader and less differentiated cortical engagement. The precuneus is involved in mental imagery, integration of visuospatial and self-referential information, and episodic simulation (*Hebscher et al., 2019*; *Plachti et al., 2023*). In young children, where functional specialization is still developing, memory representations may be less tightly constrained, leading to more diffuse activation patterns across associative and imagery-related regions (*Plachti et al., 2023*). At short delays, memories may still be relatively strong – as supported by our DDM showing higher drift rates – thus allowing these emerging gist-like signals to coexist with detailed memory traces, perhaps as a by-product of early consolidation.

The emergence of long-delay gist-like neural representations in both the mPFC and vlPFC in children may reflect consolidation-related integration of memory representations into more abstract, generic forms over time. This aligns with the mPFC's known role in integrating across memories (*Schlichting et al., 2015*), the increase in semantically transformed representations for related information over time in adults (*Krenz et al., 2023*), and the integration of new information into schema (*Gilboa and Marlatte, 2017*; *Preston and Eichenbaum, 2013*). While gist-like neural representations may support the generalization of information to bolster the sparse knowledge structures in children, this occurs at the cost of memory precision (*Reyna et al., 2016*). The involvement of the vlPFC in gist-like representations was also stable. The vlPFC has been implicated in controlled semantic retrieval, selection among competing memory traces, and integration of overlapping information (*Badre and Wagner, 2007*; *Simons and Spiers, 2003*). In the context of developing memory systems, children's engagement of the vlPFC may reflect an effort to resolve interference among overlapping scene-object memories by drawing on more abstracted or semantically reduced representations. This aligns with the findings that the vlPFC may support the selection and organization of relevant features, especially under cognitive load or when representations are weak (*Bunge et al., 2004*; *Sanefuji et al., 2011*; *Trelle et al., 2019*). Thus, the coactivation of mPFC and vlPFC in children during long-delay memory retrieval suggests that gist-based retrieval strategies are not only present but possibly compensatory, reflecting an adaptive but less precise means of accessing complex memories.

Importantly, we found that gist-like neural representations in the mPFC and vlPFC at long delay were negatively associated with memory accuracy in children. This suggests that, while gist representations may serve a generalizing function, they are less effective for supporting retrieval of detailed object-location associations. In contrast to our findings, *Masís-Obando et al., 2022*, demonstrated that more schema-based representations in the mPFC were associated with better subsequent memory performance in adults. However, the study utilized stimuli with clearly differentiable schema and details components, whereas our design required the retention of both contextual details and object associations. It is important to note that in our study, gist-like representations were observed only for correctly remembered items, suggesting that children retained some core aspects of the memory trace, even if details were compromised. This aligns with the idea that gist-based representations preserve the overall meaning or category, but not the specific spatial or contextual bindings necessary for high-fidelity retrieval (*Reyna et al., 2016*). Thus, the negative correlation may not reflect an entirely detrimental effect, but rather a trade-off between generalization and detail preservation.

Overall, our results are in line with *Brainerd et al., 2002b*, showing that in middle childhood, precise mnemonic representations (i.e. scene-specific reinstatement) and gist-like mnemonic representations can coexist at the neural level. These findings also extend the adult literature, supporting the notion of qualitative transformations of memory traces, whereby detailed and more schematic, generic memories may simultaneously be present (*Chen et al., 2017*; *St-Laurent and Buchsbaum, 2019*; *Ye et al., 2020*). Building on the postulations from *Keresztes et al., 2018*, and *Ngo et al., 2021*, who showed that 5- to 7-year-old children tend to rely more on generalization, our findings suggest that retaining memories with viewer-specific details may allow for faster integration of overlapping features into emerging knowledge structures (*Bauer, 2021*; *Gilboa and Marlatte, 2017*). In contrast, adults may

form strong, highly detailed memories supported by effective strategic retrieval mechanisms, without the need to form gist-like representations. Although category-level reinstatement has been documented in adults (e.g. *Kuhl and Chun, 2014*; *Tompary et al., 2020*; *Tompary and Davachi, 2017*), the absence of such effects in our adult group may reflect differences in study design, particularly our use of non-repeated, cross-trial comparisons based on fixation events. It may also reflect different consolidation strategies, with adults preserving more differentiated or item-specific representations, while children tend to form more schematic or generalizable representations – a pattern consistent with our interpretation and supported by prior work (*Fandakova et al., 2019*; *Sekeres et al., 2018a*).

Taken together, our findings provide novel evidence that children's memory consolidation is characterized by a shift toward gist-based representations, supported by mPFC and vlPFC engagement. While these representations may aid generalization and schema building, they appear less effective for detailed retrieval, especially over long delays. With this, we provide the first empirical evidence to support Fuzzy Trace Theory at the level of gist-like neural representations in children. Future research may build on this approach to further explore conditions under which schema-based representations enhance memory performance and how these processes differ across development.

## Limitations

Several limitations of the current study should be noted. First, our test for memory was based on a 3AFC procedure, which was intended to reduce the need for strategic search (e.g. in free recall). As reorganization and stabilization in consolidation depend on the psychological nature of mnemonic representations (*Moscovitch and Gilboa, 2022*), future studies may employ more demanding recall-based memories to characterize memory consolidation more comprehensively. Particularly, future studies may differentiate mnemonic accessibility vs. precision (*Murray et al., 2015*; *Richter et al., 2016*), as they may show differential temporal dynamics in the developing brain and involve differential neural mechanisms. Second, as we included only stimuli congruent with prior knowledge, future studies may introduce knowledge-incongruent information to investigate the beneficial effect of prior knowledge on memory consolidation more directly. Prior knowledge may impact learning and consolidation of information over time differentially by development (*McKenzie and Eichenbaum, 2011*; *van Kesteren et al., 2013*; *Wang and Morris, 2010*). Third, we concentrated on a limited age range in middle childhood. To characterize how neural mechanisms of memory consolidation evolve over time, future studies should include other developmental cohorts. Fourth, we acknowledge that our study design leads to temporal autocorrelation in the BOLD signal when calculating RSA between fixation and scene time windows. Although we argue that our results, given the identical procedure over time, are more attributed to the delay-related changes in the neural reinstatement, future studies should tailor the design of the retrieval procedure to warrant cross-run comparisons. This could be achieved by introducing the same items repeatedly across different runs. Fifth, our task may not have been demanding enough for young adults to fully challenge their memory retention and encourage the formation of more gist-like representations. Future studies could explore this further by using more challenging conditions to enhance the formation of more generic memories in adults and avoid bias related to prior knowledge. Sixth, although we focused on ROIs associated with the recollection network and implicated in retrieval of visual information, we did not investigate the connectivity between these brain regions and how it changes as memories age. Future studies should investigate consolidation-related neural connectivity patterns and their temporal dynamics in the developing brain. Finally, children in our sample were positively biased in socio-demographic score and IQ compared to young adults, which may restrict the generalizability of our results.

## Conclusions

In this study, we present novel empirical evidence on the neural mechanisms underlying the less robust memory retention of intentionally learned object-location associations in 5- to 7-year-old children compared to young adults. Our findings reveal that, over time, children show attenuated consolidation-related upregulation in neocortical and cerebellar brain regions during successful retrieval. Furthermore, they appear to form different types of memory representations than young adults: while both groups show delay-related change in detailed scene-specific reinstatement, only children exhibit the emergence of more generic gist-like representations, particularly after a longer delay. Our results suggest that, unlike the mature consolidation systems in young adults, the

developing brains of early school-age children support only partially adult-like neural changes over time. Children show less pronounced neural upregulation in core retrieval regions. At the same time, they appear to rely more on gist-like representations, possibly as a developmental mechanism to scaffold and accumulate schema-relevant knowledge despite weaker detailed memory.

## Materials and methods

### Participants

Sixty-three typically developing children and 46 young adults were recruited to participate in the study through advertisement in newspapers, on the university campus, word-of-mouth, and city registry. All participants had normal vision with or without correction, no history of psychological or neurological disorders or head trauma, average IQ >85, and were term-born (i.e. born after 37 weeks of pregnancy). Fourteen children were excluded due to: (i) incomplete task execution and missing data (n=2); (ii) poor quality of the data (n=7); (iii) technical issues during data acquisition (n=5). Seven young adult participants were excluded due to incomplete task execution and missing data (n=5) or being identified as extreme outliers (n=2) based on interquartile range (IQR; above $Q3_{upper\ quartile(75th\ percentile)}$+3xIQR or below $Q1_{lower\ quartile(25th\ percentile)}$ – 3xIQR; *Hawkins, 1980*) for memory behavioral measures. The excluded participants were comparable in terms of age, sex, and socioeconomic status to the final sample. The final total sample consisted of 49 children (22 female, mean age: 6.34 years, age range: 5.3–7.1 years), and 39 young adults (19 female, mean age: 25.60 years, age range: 21.3–30.8 years; see *Table 1* for more details).

All participants or their legal guardians gave written informed consent prior to participation. The study was approved by the ethics committee of the Goethe University Frankfurt am Main (approval E 145/18). The participants received 100 Euro as compensation for taking part in the study.

### Materials and procedure

#### Object-location associations task

Stimuli for the object-location association task were chosen based on the social studies and science curriculum for German primary school first and second graders (see similar procedure in *Brod and Shing, 2019*). The themes were chosen based on ratings provided by four primary school teachers on the familiarity of first graders with the topics. 60 different themes (e.g. classroom, farm, etc.) were chosen, each belonging to one of seven categories (i.e. field, water, housing, forest, infrastructure, indoor, farming). Four scene stimuli and four thematically congruent object pictures were selected for each theme (see *Figure 1* for an example), resulting in 240 individual scenes and 240 individual objects. The 240 object-scene pairs were assigned to versions A and B, each containing 120 object-scene pairs. Each participant was randomly assigned either version A or version B. There were six possible object locations across all scenes. Around each location, there were three possible object placements. The distribution of locations across scenes was controlled to ensure realistic placement of the objects within the scenes (for more detailed information, see *Supplementary file 16*). The object-location association task consisted of three phases (see *Figure 1*):

1. *Initial encoding phase* (day 0, day 1, day 14). A total of 120 object-location pairs were used to create the trials in this phase, with 60 pairs presented on day 0, 30 pairs on day 1, and 30 pairs on day 14. The initially learned object-scene associations on day 0 were split in two halves based on their categories. Specifically, half of the pairs from the first set and half of the pairs from the second set of 30 object-scene associations were used to create the set of 30 remote pairs for day 1 testing. A similar procedure was repeated for the remaining pairs to create a set of remote object-scene associations for day 14 retrieval. We tried to equally distribute the categories of pairs between the testing sets. During each trial, participants viewed an object in isolation for 2 s, followed by the same object superimposed on a scene at a particular location for 10 s. After this, a blank screen with a fixation cross was presented for 1 s. Participants were instructed to memorize the object-location pairs and to remember the exact location of the object within the scene using elaborative encoding strategies, such as creating a story or making a 'mental photo' of the scene. Such elaborative encoding strategies have been shown to improve memory performance in both children and adults (*Craik and Tulving, 1975*; *Pressley, 1982*; *Pressley et al., 1981*; *Shing et al., 2008*);

2. *Learning phase* (day 0, day 1, day 14). Following the initial encoding phase, participants continued learning the correct location of the object within the scene through adaptively repeated retrieval-encoding cycles. The cycles continued until participants achieved at least 83% correct responses or until the maximum of four cycles had been completed. The number of cycles, therefore, ranged from a minimum of two to a maximum of four. The 83% threshold was established during piloting as a guideline to determine the appropriate number of learning-retrieval cycles, rather than as a strict learning criterion. It served to standardize task continuation, not to serve as a basis for excluding participants post hoc. Children who did not reach the 83% threshold after the fourth cycle were still included in the analysis if their performance exceeded chance level (33%). Excluding them would have biased the sample toward higher-performing children and reduced the ecological validity of our findings. Including them ensures a more representative view of children's performance under extended learning conditions.

3. During each trial, participants were first presented with an isolated object for 2 s, followed by a 1-second blank screen with a fixation cross. They were then shown a scene containing three red-framed rectangles, indicating possible location choices. Participants had to select the correct location by choosing one of the rectangles within 12 s, and the chosen rectangle was highlighted for 0.5 s. After this, feedback in the form of a smiley face was given, with the happy face for a correct answer, a sad face for an incorrect answer, and a sleeping face for a missed answer. Following the feedback, correct object-location associations were displayed for 2 s if the choice was correct and for 3 s if the choice was incorrect or missed.

4. *Retrieval phase* (day 1 and day 14). The retrieval phase was conducted inside the MRI scanner.

Participants were presented with 30 recently learned items and 30 remote items learned on day 0. The remote item sets for day 1 and day 14 based on items from day 0 did not differ in the learning accuracy in either age group (all p>0.06 as based on the analysis of variance [ANOVA]).

Participants were instructed to recollect and visualize ('put in front of their mental eyes') as vividly as possible the location of the object within the scene. In this way, we prompted the recall of the scene and the location of the object within this scene.

Each trial began with a fixation cross-jittered between 3 and 7 s (mean of 5 s). Participants were then presented with an isolated object for 2 s, followed by the presentation of another fixation cross-jittered between 2 and 8 s (mean of 5 s). Following the fixation cross, participants were prompted with the associated scene and were required to recall the location of the object by selecting one of the three red rectangles on the scene within 7.5 s. If participants failed to respond within the deadline, the trial was terminated. No time-outs were recorded for young adults, while 5.4% of time-out trials were recorded for children, and these trials were excluded for analysis. After a choice was made or the response deadline was reached, the scene remained on the screen for an additional 0.5 s. The jitters and the order of presentation of recent and remote items were determined using OptimizeXGUI (*Spunt, 2016*) which followed an exponential distribution (*Dale, 1999*). Ten unique recently learned items (from the same testing day) and ten unique remotely learned items (from day 0) were distributed within each run (in total three runs) in the order as suggested by the software as the most optimal. There were three runs with unique sets of stimuli, each resulting in 30 unique recent and 30 unique remote stimuli overall.

## Assessment of demographic and cognitive covariates

IQ scores were assessed using the German version of the 'Kaufman Assessment Battery for Children – Second Edition' (K-ABC II; *Kaufman and Kaufman, 2015*) in children and the 'Wechsler Adult Intelligence Scale – Fourth Edition' (WAIS -IV; *Wechsler, 2015*) in young adults. General socio-demographic questionnaires to assess socio-demographic characteristics of the participants were administered as well.

## Experimental procedure

The testing was conducted over 3 days (see *Figure 1B*). On day 0, the experiment began with a short training session aimed at familiarizing participants with the object-location associations task and elaborative encoding strategy, using five object-location pairs. The experimental task started with the initial encoding of unique sets of object-location associations. Participants had to learn two unique sets comprised of 30 object-location associations each. After encoding each set, participants engaged in a brief distraction task where they listened to and had to recall a string of numbers. Next,

they underwent a learning phase with retrieval-encoding cycles until they reached a criterion of 83% (or a maximum of four cycles). This was done to minimize variances attributed to encoding, allowing for more accurate comparison of subsequent memory consolidation. Afterward, the children visited a mock scanner to become familiar with the MRI scanning environment. This procedure involved teaching the children the sounds of MRI scanning and training them to stay still during scanning.

On day 1, participants first learned a new set of 30 object-location associations, using the same learning procedure as on day 0. This was followed by retrieval in the MRI scanner, during which they were required to recall 30 object-location associations learned on day 0 (short-delay, remote) and another 30 learned on day 1 (recent). On day 14, the same procedure was followed as on day 1, with a new set of 30 object-location associations. They were again required to recall 30 object-location associations learned on day 0 (long-delay, remote) and another 30 learned on day 14 (recent). In total, participants completed 60 retrieval trials in the MR scanner on day 1 and day 14 each, which took approximately 15–20 min. Besides the primary task, participants also completed other psychometric tests across all testing sessions. Additionally, socio-demographic questionnaires were administered to young adults and legal guardians of children.

## Data acquisition
### Behavioral data acquisition
The task paradigm during all phases was presented using Psychtoolbox (*Kleiner et al., 2007*) software in MATLAB 9.5, R2018b (*MATLAB, 2018*). During the encoding and learning phases, stimuli were presented on a computer screen with the resolution of 1920×1080 pixels. During the retrieval phase, an MR-compatible screen with identical resolution was used, and participants used an MR-compatible button box with three buttons. To minimize head movements, foam cushions were placed inside the head coil, and MR-compatible headsets and ear plugs were used to reduce the scanner noise.

### MRI data acquisition
MR images were acquired on a 3 Tesla SIEMENS PRISMA MRI scanner (Siemens Medical Solutions, Erlangen, Germany) using a 64-channel head coil at Berlin Center for Advanced Neuroimaging, Charité Universitätsmedizin Berlin. Each session started with the acquisition of a localizer and head scout sequences for field-of-view alignment (FoV) based on anatomical landmarks. T1-weighted structural images were obtained with the magnetization prepared rapid gradient echo (MP-RAGE) pulse sequence (TR = 2500 ms, echo time = 2.9 ms, flip angle = 8°, FoV = 256 mm, voxel size = 1×1×1 mm$^3$, 176 slices). Functional images were acquired using echo-planar imaging sequences (TR = 800 ms, echo time = 37 ms, flip angle = 52°, FoV = 208 mm, 72 slices, voxel size = 2×2×2 mm$^3$, maximally 588 volumes). In addition, gradient echo images (field maps) were acquired before each functional run for correction of magnetic field inhomogeneities.

## Behavioral data analysis
### Learning and consolidation
The behavioral analyses were performed with R packages (*R Development Core Team, 2021*) in RStudio 2022.07.0 (RStudio, Inc). Throughout the analyses, statistical significance level was set at <0.05.

All p-values were FDR-adjusted for multiple comparisons due to multiple ROIs. As a measure of baseline memory performance, final learning accuracy was defined as the percentage of correctly learned locations in relation to the total number of items at the end of the learning phase of each day. To examine memory consolidation, we quantified memory retention across delays, focusing on trials that were correctly learned on day 0. From these trials, we calculated the percentage of correct responses, separately for day 1 and day 14. We conducted an LME model for memory measures using the lmer function from the lme4 package in R (*Bates et al., 2015*) and lmerTest (*Kuznetsova et al., 2017*). All LME models were calculated with maximum-likelihood estimation and Subject as the random intercept to account for between-subject variability in retention accuracy.

First, to investigate baseline memory performance, we analyzed whether final learning accuracy in all three sessions differed between groups. For that, we included the within-subject factor of *Session* (day 0, day 1, and day 14) and the between-subject factor of *Group* (children and young adults) in the LME model. Second, for memory retention rates, we included *Session* (day 1, day 14), *Item Type*

(recent, remote), and *Group* (children, young adults) as fixed factors in the LME model. In addition, we added *Subjects* as a random factor, as well as *IQ, Sex,* and *Handedness* (**Kang et al., 2017**; **Willems et al., 2014**) as covariates. Degrees of freedom were adjusted using the Satterthwaite's method (**Kuznetsova et al., 2017**) if the assumptions of homogeneity of variances were violated. Significant effects were followed up with Sidak post hoc multiple comparisons. For further group differences in socio-demographic measures, we performed one-way independent ANOVA or Games-Howell test (**Lee and Lee, 2018**). The effect size estimation was performed using omega squared ($\omega^2$) as a less biased estimate for reporting practical significance of observed effects (**Okada, 2013**). To determine the amount of variance explained by the model, we used the partR2 package (**Stoffel et al., 2021**).

## fMRI data pre-processing

Anatomical and functional MR data was pre-processed using fMRIPrep 22.0.0 (**Esteban et al., 2019**), based on Nipype 1.8.3 (**Gorgolewski et al., 2011**). Detailed description of the anatomical and functional data pre-processing can be found in *Supplementary file 16*.

## fMRI data analysis

fMRI data analysis was conducted with FEAT in FSL (Version 6.0.1, FMRIB's Software Library, **Jenkinson et al., 2012**; **Smith et al., 2004**; **Woolrich et al., 2009**). Prior to that, single runs were excluded if there was (i) root-mean-square realignment estimates (**Jenkinson et al., 2002**) exceeding 1 mm, (ii) framewise displacement (FD)>1, and (iii) less than two correct trials in the entire run. Based on these criteria, 14 single runs and 2 complete sessions in children were excluded from further analysis.

### GLM for mean activation

For each participant's fMRI data, a first-level analysis was performed separately for each run using a generalized linear model (GLM) with eight experimental regressors. The regressors represented the onset and duration of the following events: (i) object recent$_{correct}$, (ii) object remote$_{correct}$, (iii) scene recent$_{correct}$, (iv) scene remote$_{correct}$, (v) object recent$_{incorrect}$, (vi) object remote$_{incorrect}$, (vii) scene recent$_{incorrect}$, (viii) scene remote$_{incorrect}$. The duration of object events was 2 s, while the duration of scene events was dependent on the reaction time (RT). The regressors were convolved with a hemodynamic response function, modeled with a double-gamma function with first and second derivatives. Confounding regressors were also included in the GLM and were calculated with fMRIPrep, namely six rigid body realignment parameters, framewise displacement, and standardized DVARS (D, temporal derivatives over time courses; VARS, variance over voxels). In addition, six anatomic component-based noise correction (CompCor, a combination of cerebrospinal fluid and white matter) regressors, global signal-*Supplementary file 17*, and cosine drift terms were included, based on previous methodological studies (**Ciric et al., 2017**; **Esteban et al., 2020**; **Jones and Astle, 2021**; **Satterthwaite et al., 2013**). (We re-run the entire first-level univariate analysis using the pipeline that excluded the global signal. The resulting activation maps [see *Supplementary file 17*] differed notably from those obtained with the original pipeline. Specifically, group differences in cortical regions such as mPFC, cerebellum, and posterior PHG no longer reached significance, and the overall pattern of results appeared noisier. Additional analyses revealed that: (i) the global signal was not dependent on group or session in our sample; (ii) the global signal reduced inter-subject variability in children, likely reflecting improved signal quality; (iii) the global signal stabilized the signal and attenuated non-neuronal variability.) The functional images were spatially smoothed with SUSAN (Smallest Univalue Segment Assimilating Nucleus, **Smith and Brady, 1997**), applying a Gaussian kernel with a full-width at half-maximum of 6 mm. A high-pass Gaussian filter with a cutoff period of 80 s was applied. Contrasts were defined for each run per subject, and within-subject fixed-effects averaging across runs within each session was conducted per subject. Group-level analysis was performed with FLAME1 (**Woolrich et al., 2004**) within each session, based on the statistical maps obtained from the first-level analysis. The main contrast of interest was *object remote >object recent*, as we were primarily interested in the reinstatement of object-scene association before the scene was shown. Univariate analysis was performed with statistical tests voxel-wise and corrected for multiple comparisons with cluster-based thresholding using a z threshold of z>3.1 and a two-tailed probability of 0.001.

Several a priori ROIs were selected based on anatomical masks: bilateral anterior/posterior HC, bilateral anterior/posterior PHG, and RSC. The masks for the medio-temporal lobe ROIs were taken

from the Harvard-Oxford Cortical and Subcortical Atlases (threshold at 30% probability; *Desikan et al., 2006*), and the mask for RSC was taken from the Talairach Atlas (threshold at 30% probability; *Lancaster et al., 2000*; *Talairach and Tournoux, 1988*). For further ROIs in large cortical regions (namely mPFC, precuneus, LOC, vlPFC, and cerebellum), anatomical masks derived from Harvard-Oxford Cortical and Subcortical Atlases or Juelich Atlas (*Amunts et al., 2020*) were combined with a functional task-related map, based on mean activation across recent and remote objects across all participants and sessions, at voxel-wise threshold of z>3.1 and a two-tailed probability of 0.001. With these masks, the mean percent signal change (from the contrast of *object remote$_{correct}$* >*object recent$_{correct}$*) was extracted using FEAT in FSL for each session of each participant, which were then submitted to statistical analysis in R. An LME model was set up to model percent signal change. The LME model was calculated with maximum-likelihood estimation and *Subject* as random intercept to account for between-subject variability. As fixed factors, we included *Session* (day 1, day 14) and *Group* (children, young adults). We also added *IQ, sex, handedness,* and mean reaction time as covariates to the model.

## RSA for neural reinstatement

For the multivariate analysis, single-event (i.e. for every event on each trial) β (beta) estimates were first computed by modeling BOLD time course with a series of GLMs using the Least Square Separate method (LSS; *Abdulrahman and Henson, 2016*; *Mumford et al., 2012*). (Beta estimates were obtained from a LSS regression model. Each event was modeled with its respective onset and duration and, as such, one beta value was estimated per event [with the lags between events differing from trial to trial]. The jitter was included to enable an estimation of the patterns evoked by the events, and all subsequent RSAs were conducted normally on these estimates without further controls.) Each trial contained three events (i.e. object, fixation, and scene), hence a total of 30 GLMs (i.e. 10 for objects, 10 for fixations, and 10 for scenes) were computed for each run, session, and participant. Each of the GLMs contained four experimental regressors: for instance, one for the single fixation of interest and three more for the rest of the events (i.e. for all other fixations except the fixation of interest, for all objects, and for all scenes). The same setup was followed for the object GLMs and the scene GLMs. The regressors were convolved with the hemodynamic response function, which was modeled with a double-gamma function with first and second derivatives. Additionally, the same confounding regressors as the ones for mean-activation analysis were included.

Next, to assess whether mnemonic reinstatement during the fixation period, during which participants were supposed to recollect the scenes associated with the objects, was more item-specific or gist-like, we used the single-event beta estimates of each trial to compute two types of RSMs (*Kriegeskorte et al., 2008*). Each RSM was computed separately for each previously identified ROI. All subsequent analyses were performed with homebrew scripts available at https://github.com/iryna-1schommartz/memokid_fmri, copy archived at *Schommartz, 2025*.

## Scene-specific reinstatement

To measure the extent of scene reinstatement following object presentation, we computed a *scene-specific reinstatement index* for each neural RSM, separately for correctly remembered recent and correctly remembered remote scenes of each session (see *Figure 5*). For each specific scene, we computed the index as the average distance between the '*fixation*' and '*scene period*' (Fisher-transformed Pearson's r; *Figure 5B*), which was the correlation between neural patterns during fixation and neural patterns when viewing the scene. We averaged the index across all items, all runs within a session, and then within subjects, resulting in a single value per predefined ROIs and sessions. In addition to scene-specific reinstatement, we also calculated a *set-based reinstatement index* as a control analysis, which was calculated as an average distance between '*fixation*' and '*scene period*' for a scene and every other scene within the stimuli set (*Deng et al., 2021*; *Ritchey et al., 2013*; *Wing et al., 2015*). The set-based reinstatement index reflects the baseline level of nonspecific neural activation patterns during reinstatement. We then calculated the *corrected scene-specific reinstatement index* as the difference between set-based and scene-specific Fisher-transformed Pearson's values (*Deng et al., 2021*; *Ritchey et al., 2013*; *Wing et al., 2015*). A higher value in this index denotes more distinct scene reinstatement patterns. Only correctly retrieved items were included for this analysis. We obtained the corrected scene-specific reinstatement indices for recent items on day 1 and

day 14 and tested them for session-related differences. If no differences were observed, the set-corrected scene-specific reinstatement indices for recent scenes on days 1 and 14 were averaged to obtain a single value per ROI and participant. We then conducted a final LME model, separately for each ROI, with *Subject* as the random factor and *Delay* (recent, remote day 1, remote day 14) and *Group* (children, young adults) as fixed factors. In addition, mean neural activation was added as a covariate into the model.

## Gist-like representations

Seven overarching thematic categories were identified during stimuli selection (i.e. field, water, housing, forest, infrastructure, indoor, farming). A within-category similarity index was computed based on fixation time window of correctly remembered items belonging to the same category and excluding the similarity computation for the fixation time windows of correctly remembered items with itself. A between-category similarity index was computed based on fixation time window of correctly remembered items belonging to different categories. These indices were computed for each run and across runs, Z-standardized, and then averaged. A gist-like representations index was computed by subtracting between-categories from within-categories Z-transformed distances ([within category$_{recent}$ – between category$_{recent}$] and [within category$_{remote}$ – between category$_{remote}$] for each session, *Figure 7A and B*). The nonzero values in this corrected index reflect gist-like representations, as the similarity distance would be higher for pairs of trials with the same categories than for pairs with different categories. We applied a one-sample permutation t-test to test for significance in each ROI. Similar to the procedure described above, gist-like representation indices for recent items on day 1 and day 14 were averaged when no difference was found, obtaining a single value per ROI and participant. We then conducted a final LME model, separately for each ROI, with *Subject* as the random factor and *Delay* (recent, remote day 1, remote day 14) and *Group* (children, young adults) as fixed factors and mean neural activation as a covariate, to analyze any delay-related differences in gist-like representations index for successfully retrieved trials. Finally, we also explored whether over time, long-delay item-specific and representations are beneficial or detrimental for memory performance by correlating the index with memory retention rates. We tested whether this correlation within each group differs based on ROI. If no differences were observed, we averaged representations indices across ROIs that showed significant representations in long delay.

## Brain-behavioral relations

To examine the connections between brain function and behavior, we utilized brain metrics generated via the application of a multivariate method known as PLSC (*Abdi and Williams, 2013*; *McIntosh et al., 1996*; *Schommartz et al., 2023*). This approach focuses on multivariate links between specified neural measures in ROIs and fluctuations in memory performance over short and long delays across different age cohorts. We argue that this multivariate strategy offers a more comprehensive understanding of the relationships between brain metrics across various ROIs and memory performance, given their mutual dependence and connectivity (refer to *Genon et al., 2022*, for similar discussions).

Initially, we established a cross-subject correlation matrix that included (i) a matrix (n × 10) comprising short- and long-delay brain indices (encompassing both neural upregulation, scene-specific and gist-like indices) for all specified ROIs, and (ii) a vector (n-sized) that represents a continuous assessment of either short-delay or long-delay memory performance (RR): R=CORR (RR, ROIs). Prior to the correlation, all metrics were standardized. The decomposition of this correlation matrix, R=USV', was performed using singular value decomposition, yielding singular vectors U and V, or saliences. Here, the left singular vector symbolizes the weights for short- or long-delay memory accuracy (U), while the right singular vector represents ROI weights (V), indicating specific neural indices that optimally represent R, with S being a matrix of singular values.

Subsequently, PLSC identifies a singular estimable LV, uncovering pairs of latent vectors with maximal covariance that best describe the association between memory retention rates and ROI neural indices. Therefore, LV delineates distinct patterns of neural indices across ROIs closely linked to either short- or long-delay retention rates. Moreover, we computed a singular value for each participant, termed a within-person 'profile', summarizing the robust expression of the defined LV's pattern. This was achieved by multiplying the model-derived ROI weight vector (V) with the within-person estimates of ROI neural metrics.

To verify the generalizability and significance of the saliences or LVs, we performed 5000 permutation tests to derive a p-value. We also determined the stability of the within-LV weights by bootstrapping with 5000 resamples, calculating a bootstrap ratio (BSRs) by dividing each ROI's salience by its bootstrap standard error. BSRs, analogous to Z-scores, serve as normalized robustness estimates; hence, values exceeding 1.96 (p<0.05) indicate statistically stable saliences. Utilizing PLSC for multivariate statistical analysis in one step eliminates the need for multiple comparisons correction across all ROIs (*McIntosh et al., 1996*).

To avoid multicollinearity and redundancy, which might diminish the power to uncover neural-behavioral links through conventional statistical approaches, we initially derived a single metric per participant – a participant's expression of the latent brain pattern (i.e. brain score) for neural indices that share the most variance with either short-delay or long-delay memory accuracy variations. We further explored how these brain patterns correlate with memory performance.

## Acknowledgements

We thank all the children and parents who participated in the study. This work was conducted at the Berlin Center for Advanced Neuroimaging (BCAN) and was supported by the Deutsche Forschungsgemeinschaft (DFG; German Research Foundation, Project-ID 327654276, SFB 1315, 'Mechanisms and Disturbances in Memory Consolidation: from Synapses to Systems'). The work of YLS was also supported by the European Union (ERC-2018-StG-PIVOTAL-758898) and the Hessisches Ministerium für Wissenschaft und Kunst (HMWK; project 'The Adaptive Mind'). We also thank Henriette Schultz and Nina Wald de Chamorro for their assistance with study management and data collection.

## Additional information

### Funding

| Funder | Grant reference number | Author |
| --- | --- | --- |
| Deutsche Forschungsgemeinschaft | 327654276 | Iryna Schommartz<br>Philip F Lembcke<br>Martin Bauer<br>Angela M Kaindl<br>Yee Lee Shing |
| European Research Council | 758898 | Javier Ortiz-Tudela<br>Yee Lee Shing |
| Hessisches Ministerium für Wissenschaft und Kunst | TAM | Yee Lee Shing |
| Deutsche Forschungsgemeinschaft | SFB 1315 | Iryna Schommartz<br>Philip F Lembcke<br>Martin Bauer<br>Angela M Kaindl<br>Claudia Buss<br>Yee Lee Shing |

The funders had no role in study design, data collection and interpretation, or the decision to submit the work for publication.

### Author contributions

Iryna Schommartz, Data curation, Formal analysis, Investigation, Visualization, Methodology, Writing – original draft, Project administration, Writing – review and editing; Philip F Lembcke, Data curation, Formal analysis; Javier Ortiz-Tudela, Formal analysis, Writing – review and editing; Martin Bauer, Formal analysis; Angela M Kaindl, Conceptualization, Funding acquisition, Writing – review and editing; Claudia Buss, Conceptualization, Supervision, Funding acquisition, Project administration, Writing – review and editing; Yee Lee Shing, Conceptualization, Supervision, Funding acquisition, Methodology, Project administration, Writing – review and editing

### Author ORCIDs

Iryna Schommartz ⓘ https://orcid.org/0000-0001-8655-9259

Angela M Kaindl  https://orcid.org/0000-0001-9454-206X

### Ethics

Human subjects: All participants or their legal guardians gave written informed consent prior to participation. The study was approved by the ethics committee of the Goethe University Frankfurt am Main (approval E 145/18).

Reviewer #2 (Public review): https://doi.org/10.7554/eLife.89908.4.sa1
Author response https://doi.org/10.7554/eLife.89908.4.sa2

## Additional files

### Supplementary files

MDAR checklist

Supplementary file 1. Statistical overview of the linear mixed effects model for memory retention rates for initially correctly learned items (corrected for chance performance) based on participants who needed only two learning cycles (N = 28).

Supplementary file 2. Statistical overview of post hoc analysis of the Item Type x Group Interaction effects for the linear mixed effects model for memory retention rates for initially correctly learned items (corrected for chance performance) based on participants who needed only two learning cycles.

Supplementary file 3. Two-sided permutation t-tests were conducted to assess whether the mean signal difference of the contrast remote > recent significantly differed from zero for each combination of ROI, session, and group.

Supplementary file 4. Test of neural activation during object presentation separately for recent and remote memories for significance (higher than zero).

Supplementary file 5. Full statistical overview of LME model for univariate analysis (includes only ROI that show or below zero upregulation).

Supplementary file 6. Statistical overview of the main and interaction effects of the linear mixed effects models recent and remote univariate results of correctly recognized items.

Supplementary file 7. fMRI univariate analysis.

Supplementary file 8. Full statistical overview of LME model for univariate analysis (sub-sampled to those participants who reached accuracy criteria after 2 encoding loops).

Supplementary file 9. Statistical overview of the main and interaction effects of the linear mixed effects model for scene-specific reinstatement.

Supplementary file 10. Statistical overview of the main and interaction effects of the linear mixed effects model for object-specific reinstatement.

Supplementary file 11. Statistical overview of the main and interaction effects of the linear mixed effects model for remote object-specific reinstatement.

Supplementary file 12. Statistical overview of the main and interaction effects of the linear mixed effects model for scene-specific reinstatement for corpus callosum subregions.

Supplementary file 13. Statistical overview of LME-model based Sidak corrected post hoc comparisons for scene-specific reinstatement analysis for corpus callosum subregions.

Supplementary file 14. Test of gist-like representations index for significance (higher than zero).

Supplementary file 15. Memory Strength across Time.

Supplementary file 16. Assessment of demographic and cognitive covariates and fMRI data pre-processing.

Supplementary file 17. Statistical overview of the main and interaction effects of the linear mixed effects models for remote > recent univariate results for correctly recognized items (based on the pipeline without global signal).

Supplementary file 18. Statistical overview of the main and interaction effects of the linear mixed effects model for scene-based univariate neural analysis.

## Data availability

The code for the analyses presented in this paper is openly accessible at https://github.com/iryna-1schommartz/memokid_fmri (copy archived at *Schommartz, 2025*). All behavioural and neuroimaging data has been deposited at https://osf.io/grku2/ and are publicly available.

The following dataset was generated:

| Author(s) | Year | Dataset title | Dataset URL | Database and Identifier |
|---|---|---|---|---|
| Schommartz I | 2025 | Recent and remote memory consolidation. Date Set. | https://doi.org/10. 17605/OSF.IO/GRKU2 | Open Science Framework, 10.17605/OSF.IO/GRKU2 |

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
