## [Editor Report · eLife Assessment]

This paper provides potentially **valuable** insight into why memory consolidation may differ between children (5-7 years of age) and adults. The work hints at developmental differences in neural engagement during the retrieval of recent and remote memories. However, there are several major concerns with the analyses not alleviated by included controls, and as such the evidence supporting the authors' main claims remains **incomplete**.

---

## [Referee Report · Reviewer #2 (Public review)]

Summary:

Schommartz et al. present a manuscript characterizing neural signatures of reinstatement during cued retrieval of middle-aged children compared to adults. The authors utilize a paradigm where participants learn the spatial location of semantically related item-scene memoranda which they retrieve after short or long delays. The paradigm is especially strong as the authors include novel memoranda at each delayed time point to make comparisons across new and old learning. In brief, the authors find that children show more forgetting than adults, and adults show greater engagement of cortical networks after longer delays as well as stronger item-specific reinstatement. Interestingly, children show more category-based reinstatement, however, evidence supports that this marker may be maladaptive for retrieving episodic details. The question is extremely timely both given the boom in neurocognitive research on the neural development of memory, and the dearth of research on consolidation in this age group. Also, the results provide novel insights into why consolidation processes may be disrupted in children.

Comments on latest version:

I carefully reviewed not only the responses to my own reviews as well as those raised by the other reviewers. While they addressed some of the concerns raised in the process, I think many substantive concerns remain.

While I appreciate the authors sub-sample analysis to control for re-exposure to stimuli in children versus adults, the authors only performed this analysis on memory performance and univariate activation, but they did not run this on the main focus of interest which was the pattern analysis. I think this is critical to run as these measures would be the ones most sensitive to repetition and are the foundation for the major claims of the manuscript.

Also, I still agree that the authors should do an analysis the subsets the number of trials. While they highlight problems with the loss of statistical power and introduced variability, it is these two very same factors that could be potentially driving these differences.

As part of their efforts to resolve some concerns about their analysis pipeline, the authors show that similar effects do not emerge for incorrectly remembered items. While this is helpful, it would be important to do direct comparisons of subsequently remembered and forgotten items.

There is a major concern that the white matter control ROIs are showing session effects, and even the ones that are for the contrasts of interest are marginally significant (p=0.08). This raises significant concerns about the ability to interpret the authors' main signal of interest. While I appreciate many of the other control analyses, this one analysis is quite worrisome.

Similarly, for the item related analysis, the results should look absolutely different, but the authors are showing effects of p-values that are hovering around significance. Indeed, for these analyses to be true controls, perhaps they should directly control across conditions (i.e., use the item reinstatement as a confound control statistically).

The across run comparisons are a nice addition to the revision, and although they are similar to within conditions, I would recommend when combining these signals there is a factor included for within versus across run comparisons, and the authors show that there are no interactions with this feature.

---

## [Author Response]

The following is the authors’ response to the previous reviews

**Reviewer #1 (Public Review):**
Summary:This paper by Schommartz and colleagues investigates the neural basis of memory reinstatement as a function of both how recently the memory was formed (recent, remote) and its development (children, young adults). The core question is whether memory consolidation processes as well as the specificity of memory reinstatement differ with development. A number of brain regions showed a greater activation difference for recent vs. remote memories at the long versus shorter delay specifically in adults (cerebellum, PHG, LOC). A different set showed decreases in the same comparison, but only in children (precuneus, RSC). The authors also used neural pattern similarity analysis to characterize reinstatement, though still in this revised paper I have substantive concerns about how the analyses were performed. While scene-specific reinstatement decreased for remote memories in both children and adults, claims about its presence cannot be made given the analyses. Gist-level reinstatement was observed in children but not adults, but I also have concerns about this analysis. Broadly, the behavioral and univariate findings are consistent with the idea memory consolidation differs between children and adults in important ways, and takes a step towards characterizing how.Strengths:The topic and goals of this paper are very interesting. As the authors note, there is little work on memory consolidation over development, and as such this will be an important data point in helping us begin to understand these important differences. The sample size is great, particularly given this is an onerous, multi-day experiment; the authors are to be commended for that. The task design is also generally well controlled, for example as the authors include new recently learned pairs during each session.Weaknesses:As noted above and in my review of the original submission, the pattern similarity analysis for both item and category-level reinstatement were performed in a way that is not interpretable given concerns about temporal autocorrelation within scanning run.Unfortunately these issues remain of concern in this revision because they were not rectified. Most of my review focuses on this analytic issue, though I also outline additional concerns.(1) The pattern similarity analyses are largely uninterpretable due to how they were performed.(a) First, the scene-specific reinstatement index: The authors have correlated a neural pattern during a fixation cross (delay period) with a neural pattern associated with viewing a scene as their measure of reinstatement. The main issue with this is that these events always occurred back-to-back in time. As such, the two patterns will be similar due simply to the temporal autocorrelation in the BOLD signal. Because of the issues with temporal autocorrelation within scanning run, it is always recommended to perform such correlations only across different runs. In this case, the authors always correlated patterns extracted from the same run, and which moreover have temporal lags that are perfectly confounded with their comparison of interest (i.e., from Fig 4A, the "scene-specific" comparisons will always be back-to-back, having a very short temporal lag; "set-based" comparisons will be dispersed across the run, and therefore have a much higher lag). The authors' within-run correlation approach also yields correlation values that are extremely high - much higher than would be expected if this analysis was done appropriately. The way to fix this would be to restrict the analysis to only cross-run comparisons, which is not possible given the design.To remedy this, in the revision the authors have said they will refrain from making conclusions about the presence of scene-specific reinstatement (i.e., reinstatement above baseline). While this itself is an improvement from the original manuscript, I still have several concerns. First, this was not done thoroughly and at times conclusions/interpretations still seem to imply or assume the presence of scene reinstatement (e.g., line 979-985, "our research supports the presence of scene-specific reinstatement in 5-to-7-year-old children"; line 1138).

We thank the reviewers for pointing out that there are inconsistencies in our writing. We agree that we cannot make any claims about the baseline level of scene-specific reinstatement. To reiterate, our focus is on the changes in reinstatement over time (30 minutes, 24 hours, and two weeks after learning), which showed a robust decrease. Importantly, scenespecific reinstatement indices for recent items — tested on different days — did not significantly differ, as indicated by non-significant main effects of Session (all p > .323) and Session x ROI interactions (all p > .817) in either age group. This supports our claim that temporal autocorrelation is stable and consistent across conditions and that the observed decline in scene-specific reinstatement reflects a time-dependent change in remote retrieval. We have revised the highlighted passages, accordingly, emphasizing the delay-related decrease in scene-specific reinstatement rather than its absolute magnitude.

Second, the authors' logic for the neural-behavioural correlations in the PLSC analysis involved restricting to regions that showed significant reinstatement for the gist analysis, which cannot be done for the analogous scene-specific reinstatement analysis. This makes it challenging to directly compare these two analyses since one was restricted to a small subset of regions and only children (gist), while scene reinstatement included both groups and all ROIs.

We thank the reviewer for pointing this out and want to clarify that it was not our intention to directly compare these analyses. For the neural-behavioral correlations, we included only those regions identified based on gist-like representations baseline, whereas for scene-specific reinstatement, we included all regions due to the absence of such a baseline. The primary aim of the PLSC analysis was to identify a set of regions that, after a stringent permutation and bootstrapping procedure, form a latent variable that explains a significant proportion of variance in behavioral performance across all participants.

Third, it is also unclear whether children and adults' values should be directly comparable given pattern similarity can be influenced by many factors like motion, among other things.

We thank the reviewer for raising this important point. In our multivariate analysis, we included confounding regressors specifically addressing motion-related artefacts. Following recent best practices for mitigating motion-related confounding factors in both adult and pediatric fMRI data (Ciric et al., 2017; Esteban et al., 2020; Jones et al., 2021; Satterthwaite et al., 2013), we implemented the most effective motion correction strategies.

Importantly, our group × session interaction analysis focuses on relative changes in reinstatement over time rather than comparing absolute levels of pattern similarity between children and adults. This approach controls for potential baseline differences and instead examines whether the magnitude of delay-related changes differs across groups. We believe this warrants the comparison and ensures that our conclusions are not driven by group-level differences in baseline similarity or motion artifacts.

My fourth concern with this analysis relates to the lack of regional specificity of the effects. All ROIs tested showed a virtually identical pattern: "Scene-specific reinstatement" decreased across delays, and was greater in children than adults. I believe control analyses are needed to ensure artifacts are not driving these effects. This would greatly strengthen the authors' ability to draw conclusions from the "clean" comparison of day 1 vs. day 14. (A) The authors should present results from a control ROI that should absolutely not show memory reinstatement effects (e.g., white matter?). Results from the control ROI should look very different - should not differ between children and adults, and should not show decreases over time.(C) If the same analysis was performed comparing the object cue and immediately following fixation (rather than the fixation and the immediately following scene), the results should look very different. I would argue that this should not be an index of reinstatement at all since it involves something presented visually rather than something reinstated (i.e., the scene picture is not included in this comparison). If this control analysis were to show the same effects as the primary analysis, this would be further evidence that this analysis is uninterpretable and hopelessly confounded.

We appreciate the reviewer’s suggestion to strengthen the interpretation of our findings by including appropriate control analyses to rule out non-memory-related artifacts. In response, we conducted several control analyses, detailed below, which collectively support the specificity of the observed reinstatement effects. The report of the results is included in the manuscript (line 593-619).

We checked that item reinstatement for incorrectly remembered trial did not show any session-related decline for any ROI. This indicates that the reinstatement for correctly remembered items is memory-related (see Fig. S5 for details).

We conducted additional analyses on three subregions of the corpus callosum (the body, genu, and splenium). The results of the linear mixed-effects models revealed no significant group effect (all p > .426), indicating no differences between children and adults. In contrast, all three ROIs showed a significant main effect of Session (all p < .001). However, post hoc analyses indicated that this effect was driven by differences between the recent and the Day 14 remote condition. The main contrasts of interest – recent vs. Day 1 remote and Day 1 remote vs. Day 14 remote – were not significant (all p > .080; see Table S10.4), suggesting that, unlike in other ROIs, there was no delay-related decrease in scene-specific reinstatement in these white matter regions.

Then we repeated our analysis using the same procedure but replaced the “scene” time window with the “object” time window. The rationale for this control is that comparing the object cue to the immediately following fixation period should not reflect scene reinstatement, as the object and the reinstated scene rely on distinct neural representations. Accordingly, we did not expect a delay-related decrease in the reinstatement index. Consistent with this expectation, the analysis using the object – fixation similarity index – though also influenced by temporal autocorrelation – did not reveal any significant effect of session or delay in any ROI (all p > .059; see Table S9, S9.1).

Together, these control analyses provide converging evidence that our findings are not driven by global or non-specific signal changes. We believe that these control analyses strengthen our interpretation about delay-related decrease in scene-specific reinstatement index.

(B) Do the recent items from day 1 vs. day 14 differ? If so, this could suggest something is different about the later scans (and if not, it would be reassuring).

The recent items tested on day 1 and day14 do not differ (all p. > .323). This effect remains stable across all ROIs.

(b) For the category-based neural reinstatement: (1) This suffers from the same issue of correlations being performed within run. Again, to correct this the authors would need to restrict comparisons to only across runs (i.e., patterns from run 1 correlated with patterns for run 2 and so on). The authors in their response letter have indicated that because the patterns being correlated are not derived from events in close temporal proximity, they should not suffer from the issue of temporal autocorrelation. This is simply not true. For example, see the paper by Prince et al. (eLife 2022; on GLMsingle). This is not the main point of Prince et al.'s paper, but it includes a nice figure that shows that, using standard modelling approaches, the correlation between (same-run) patterns can be artificially elevated for lags as long as ~120 seconds (and can even be artificially reduced after that; Figure 5 from that paper) between events. This would affect many of the comparisons in the present paper. The cleanest way to proceed is to simply drop the within-run comparisons, which I believe the authors can do and yet they have not. Relatedly, in the response letter the authors say they are focusing mainly on the change over time for reinstatement at both levels including the gist-type reinstatement; however, this is not how it is discussed in the paper. They in fact are mainly relying on differences from zero, as children show some "above baseline" reinstatement while adults do not, but I believe there were no significant differences over time (i.e., the findings the authors said they would lean on primarily, as they are arguably the most comparable).

We thank the reviewer for this important comment regarding the potential inflation of similarity values due to within-run comparisons.

To address the reviewer’s concern, we conducted an additional cross-run analysis for all correctly retrieved trials. The approach restricted comparisons to non-overlapping runs (run1run2, run2-run3, run1-run3). This analysis revealed robust gist-like reinstatement in children for remote Day 14 memories in the mPFC (p = .035) and vlPFC (p = .0007), in adults’ vlPFC remote Day 1 memories (p = .029), as well as in children and adults remote Day 1 memories in LOC (p < .02). A significant Session effect in both regions (mPFC: p = .026; vlPFC: p = .002) indicated increased reinstatement for long delay (Day 14) compared to short-delay and recent session (all p < .05). Given that the cross-run results largely replicate and reinforce the effects found previously with within-run, we believe that combining both sources of information is methodologically justified and statistically beneficial. Specifically, both approaches independently identified significant gist-like reinstatement in children’s mPFC and vlPFC (although within-run vlPFC effect (short delay: p = .038; long delay p = .047) did not survive multiple comparisons), particularly for remote memories. Including both withinrun and between-run comparisons increases the number of unique, non-repeated trial pairs, improving statistical power without introducing redundancy. While we acknowledge that same-run comparisons may be influenced by residual autocorrelation (as shown by Prince et al. 2022, eLife), we believe that our design mitigates this risk through consistency between within-run and cross-run results, long inter-trial intervals, and trial-wise estimation of activation. We have adjusted the manuscript, accordingly, reporting the combined analysis. We also report cross-run and within-run analysis separately in supplementary materials Tables S12.1, S12.2, showing that they converge with the cross-run results and thus strengthen rather than dilute the findings.

As suggested, we now explicitly highlight the change over time as the central finding. We observe a clear increase in gist-like reinstatement from recent to remote memories in children, particularly in mPFC and vlPFC. These effects based on combined within- and cross-run comparisons, are now clearly stated in the main results and interpreted in the discussion accordingly.

(2) This analysis uses a different approach of comparing fixations to one another, rather than fixations to scenes. In their response letter and the revised paper, the authors do provide a bit of reasoning as to why this is the most sensible. However, it is still not clear to me whether this is really "reinstatement" which (in my mind) entails the re-evoking of a neural pattern initially engaged during perception. Rather, could this be a shared neural state that is category specific?

We thank the reviewer for raising this important conceptual point about whether our findings reflect reinstatement in the classical sense — namely, the reactivation of perceptual neural patterns — or a shared, category-specific state.

While traditional definitions of reinstatement emphasize item-specific reactivation (e.g., Ritchey et al., 2013; Xiao et al., 2017) it is increasingly recognized that memory retrieval can also involve the reactivation of abstracted, generalized, or gist-like representations, especially as memories consolidate. Our analysis follows this view, aimed to capture how memory representations evolve over time, particularly in development.

Several studies support this broader notion of gist-like reinstatement. For instance, Chen et al. (2017) showed that while event-specific patterns were reinstated across the default mode network and medial temporal lobe, inter-subject recall similarity exceeded encodingretrieval similarity, suggesting transformation and abstraction beyond perceptual reinstatement. Zhuang et al. (2021) further showed that loss of neural distinctiveness in the

MTL over time predicted false memories, linking neural similarity to representational instability. This aligns with our finding that greater gist-like reinstatement is associated with lower memory accuracy.

Ye et al. (2020) discuss how memory representations are reshaped post-encoding — becoming more differentiated, integrated, or weakened depending on task goals and neural resources. While their work focuses on adults, our previous findings (Schommartz et al., 2023) suggest that children’s neural systems (the same sample) are structurally immature, making them more likely to rely on gist-based consolidation (see Fandakova et al., 2019). Adults, by contrast, may retain more item-specific traces.

Relatedly, St-Laurent & Buchsbaum (2019) show that with repeated encoding, neural memory representations become increasingly distinct from perception, suggesting that reinstatement need not mimic perception. We agree that reinstatement does not always reflect reactivation of low-level sensory patterns, particularly over long delays or in developing brains.

Finally, while we did not correlate retrieval patterns directly with perceptual encoding patterns, we assessed neural similarity among retrieved items within vs. between categories, based on non-repeated, independently sampled trials. This approach is intended to capture the structure and delay-related transformation of mnemonic representations, especially in terms of how they become more schematic or gist-like over time. Our findings align conceptually with the results of Kuhl et al. (2012), who used MVPA to show that older and newer visual memories can be simultaneously reactivated during retrieval, with greater reactivation of older memories interfering with retrieval accuracy for newer memories. Their work highlights how overlapping category-level representations in ventral temporal cortex can reflect competition among similar memories, even in the absence of item-specific cues. In our developmental context, we interpret the increased neural similarity among category members in children as possibly reflecting such representational overlap or competition, where generalized traces dominate over item-specific ones. This pattern may reflect a shift toward efficient but less precise retrieval, consistent with developmental constraints on memory specificity and consolidation.

In this context, we view our findings as evidence of memory trace reorganization — from differentiated, item-level representations toward more schematic, gist-like neural patterns (Sekeres et al., 2018), particularly in children. Our cross-run analyses further confirm that this is not an artifact of same-run correlations or low-level confounds. We have clarified this distinction and interpretation throughout the revised manuscript (see lines 144-158; 1163-1170).

In any case, I think additional information should be added to the text to clarify that this definition differs from others in the literature. The authors might also consider using some term other than reinstatement. Again (as I noted in my prior review), the finding of no category-level reinstatement in adults is surprising and confusing given prior work and likely has to do with the operationalization of "reinstatement" here. I was not quite sure about the explanation provided in the response letter, as category-level reinstatement is quite widespread in the brain for adults and is robust to differences in analytic procedures etc.

We agree that our operationalization of "reinstatement" differs from more conventional uses of the term, which typically involve direct comparisons between encoding and retrieval phases, often with item-level specificity. As our analysis is based on similarity among retrieval-phase trials (fixation-based activation patterns) and focuses on within- versus between-category neural similarity, we agree that the term reinstatement may suggest a stronger encoding–retrieval mapping than we are claiming.

To avoid confusion and overstatement, we have revised the terminology throughout the manuscript: we now refer to our measure as “gist-like representations” rather than “gist-like reinstatement.” This change better reflects the nature of our analysis — namely, that we are capturing shared neural patterns among category-consistent memories that may reflect reorganized or abstracted traces, especially after delay and in development.

As the reviewer rightly points out, category-level reinstatement is well documented in adults (e.g., Kuhl & Chun, 2014; Tompary et al., 2020; Tompary & Davachi, 2017). The absence of such effects in our adult group may indeed reflect differences in study design, particularly our use of non-repeated, cross-trial comparisons based on fixation events. It may also reflect different consolidation strategies, with adults preserving more differentiated or item-specific representations, while children form more schematic or generalizable representations — a pattern consistent with our interpretation and supported by prior work (Fandakova et al., 2019; Sekeres et al., 2018)

We have updated the relevant sections of the manuscript (Results, Discussion (particularly lines 1163- 1184), and Figure captions) to clarify this terminology shift and explicitly contrast our approach with more standard definitions of reinstatement. We hope this revision provides the needed conceptual clarity while preserving the integrity of our developmental findings.

(3) Also from a theoretical standpoint-I'm still a bit confused as to why gist-based reinstatement would involve reinstatement of the scene gist, rather than the object's location (on the screen) gist. Were the locations on the screen similar across scene backgrounds from the same category? It seems like a different way to define memory retrieval here would be to compare the neural patterns when cued to retrieve the same vs. similar (at the "gist" level) vs. different locations across object-scene pairs. This is somewhat related to a point from my review of the initial version of this manuscript, about how scene reinstatement is not necessary. The authors state that participants were instructed to reinstate the scene, but that does not mean they were actually doing it. The point that what is being measured via the reinstatement analyses is actually not necessary to perform the task should be discussed in more detail in the paper.

We appreciate the reviewer’s thoughtful theoretical question regarding whether our measure of “gist-like representations” might reflect reinstatement of spatial (object-location) gist, rather than scene-level gist. We would like to clarify several key points about our task design and interpretation:

(1) Object locations were deliberately varied and context dependent.

In our stimulus set, each object was embedded in a rich scene context, and the locations were distributed across six distinct possible areas within each scene, with three possible object placements per location. These placements were manually selected to ensure realistic and context-sensitive positioning of objects within the scenes. Importantly, locations were not fixed across scenes within a given category. For example, objects placed in “forest” scenes could appear in different screen locations across different scene exemplars (e.g., one in the bottom-left side, another floating above). Therefore, the task did not introduce a consistent spatial schema across exemplars from the same scene category that could give rise to a “location gist.”

(2) Scene categories provided consistent high-level contextual information.

By contrast, the scene categories (e.g., farming, forest, indoor, etc.) provided semantically coherent and visually rich contextual backgrounds that participants could draw upon during retrieval. This was emphasized in the instruction phase, where participants were explicitly encouraged to recall the whole scene based on the stories they created during learning (not just the object or its position). While we acknowledge that we cannot directly verify the reinstated content, this instruction aligns with prior studies showing that scene and context reinstatement can occur even without direct task relevance (e.g., Kuhl & Chun, 2014; Ritchey et al., 2013).

(3) Our results are unlikely to reflect location-based reinstatement.

If participants had relied on a “location gist” strategy, we would have expected greater neural similarity across scenes with similar spatial layouts, regardless of category. However, our design avoids this confound by deliberately varying locations across exemplars within categories. Additionally, our categorical neural similarity measure contrasted within-category vs. between-category comparisons — making it sensitive to shared contextual or semantic structure, not simply shared screen positions.

Considering this, we believe that the neural similarity observed in the mPFC and vlPFC in children at long delay reflects the emergence of scene-level, gist-like representations, rather than low-level spatial regularities. Nevertheless, we now clarify this point in the manuscript and explicitly discuss the limitation that reinstatement of scene context was encouraged but not required for successful task performance.

Future studies could dissociate spatial and contextual components of reinstatement more directly by using controlled spatial overlap or explicit location recall conditions. However, given the current task structure, location-based generalization is unlikely to account for the category-level similarity patterns we observe.

(2) Inspired by another reviewer's comment, it is unclear to me the extent to which age group differences can be attributed to differences in age/development versus memory strength. I liked the other reviewer's suggestions about how to identify and control for differences in memory strength, which I don't think the authors actually did in the revision. They instead showed evidence that memory strength does seem to be lower in children, which indicates this is an interpretive confound. For example, I liked the reviewer's suggestion of performing analyses on subsets of participants who were actually matched in initial learning/memory performance would have been very informative. As it is, the authors didn't really control for memory strength adequately in my opinion, and as such their conclusions about children vs. adults could have been reframed as people with weak vs. strong memories. This is obviously a big drawback given what the authors want to conclude. Relatedly, I'm not sure the DDM was incorporated as the reviewer was suggesting; at minimum I think the authors need to do more work in the paper to explain what this means and why it is relevant. (I understand putting it in the supplement rather than the main paper, but I still wanted to know more about what it added from an interpretive perspective.)

We appreciate the reviewer’s thoughtful concerns regarding potential confounding effects of memory strength on the observed age group differences. This is indeed a critical issue when interpreting developmental findings.

While we agree that memory strength differs between children and adults — and our own DDM-based analysis confirms this, mirroring differences observed in accuracy — we would like to emphasize that these differences are not incidental but rather reflect developmental changes in the underlying memory system. Given the known maturation of both structural and functional memory-related brain regions, particularly the hippocampus and prefrontal cortex, we believe it would be theoretically inappropriate to control for memory strength entirely, as doing so would remove variance that is central to the age-related neural effects we aim to understand.

To address the reviewer's concern empirically, we conducted an additional control analysis in which we subsampled children to include only those who reached learning criterion after two cycles (N = 28 out of 49 children, see Table S1.1, S1.2, Figure S1, Table S9.1), thereby selecting a high-performing subgroup. Importantly, this subsample replicated behavioral and neural results to the full group. This further suggests that the observed age group differences are not merely driven by differences in memory strength.

As abovementioned, the results of the DDM support our behavioral findings, showing that children have lower drift rates for evidence accumulation, consistent with weaker or less accessible memory representations. While these results are reported in the Supplementary Materials (section S2.1, Figure S2, Table S2), we agree that their interpretive relevance should be more clearly explained in the main text. We have therefore updated the Discussion section to explicitly state how the DDM results provide converging evidence for our interpretation that developmental differences in memory quality — not merely strategy or task performance — underlie the observed neural differences (see lines 904-926).

In sum, we view memory strength not as a confound to be removed, but as a meaningful and theoretically relevant factor in understanding the emergence of gist-like representations in children. We have clarified this interpretive stance in the revised manuscript and now discuss the role of memory strength more explicitly in the Discussion.

(3) Some of the univariate results reporting is a bit strange, as they are relying upon differences between retrieval of 1- vs. 14-day memories in terms of the recent vs. remote difference, and yet don't report whether the regions are differently active for recent and remote retrieval. For example in Figure 3A, neither anterior nor posterior hippocampus seem to be differentially active for recent vs. remote memories for either age group (i.e., all data is around 0). Precuneus also interestingly seems to show numerically recent>remote (values mostly negative), whereas most other regions show the opposite. This difference from zero (in either direction) or lack thereof seems important to the message. In response to this comment on the original manuscript, the authors seem to have confirmed that hippocampal activity was greater during retrieval than implicit baseline. But this was not really my question - I was asking whether hippocampus is (and other ROIs in this same figure are) differently engaged for recent vs. remote memories.

We thank the reviewer for bringing up this important point. Our previous analysis showed that both anterior and posterior regions of the hippocampus, anterior parahippocampal gyrus and precuneus exhibited significant activation from zero in children and adults for correctly remembered items (see Fig. S2, Table S7 in Supplementary Materials). Based on your suggestion, our additional analysis showed:

(i) The linear mixed-effects model for correctly remembered items showed no significant interaction effects (group x session x memory age (recent, remote)) for the anterior hippocampus (all p > .146; see Table S7.1).

(ii) For the posterior hippocampus, we observed a significant main effect of group (F(1,85), = 5.62, p = .038), showing significantly lower activation in children compared to adults (b = .03, t = -2.34, p = .021). No other main or interaction effects were significant (all p > .08; see Table S7.1).

(iii) For the anterior PHG, that also showed no significant remote > recent difference, the model showed that there was indeed no difference between remote and recent items across age groups and delays (all p > .194; Table S7.1).

Moreover, when comparing recent and remote hippocampal activation directly, there were no significant differences in either group (all FDR-adjusted p > .116; Table S7.2), supporting the conclusion that hippocampal involvement was stable across delays for successfully retrieved items.

In contrast, analysis of unsuccessfully remembered items showed that hippocampal activation was not significantly different from zero in either group (all FDR-adjusted p > .052; Fig. S2.1, Table S7.1), indicating that hippocampal engagement was specific to successful memory retrieval.

To formally test whether hippocampal activation differs between remembered and forgotten items**,** we ran a linear mixed-effects model with Group, Memory Success (remembered vs. forgotten), and ROI (anterior vs. posterior hippocampus) as fixed effects. This model revealed a robust main effect of memory success (F(1,1198) = 128.27, p < .001), showing that hippocampal activity was significantly higher for remembered compared to forgotten items (b = .06, t(1207) = 11.29, p < .001; Table S7.3).

As the reviewer noted, precuneus activation was numerically higher for recent vs. remote items, and this was confirmed in our analysis. While both recent and remote retrieval elicited significantly above-zero activation in the precuneus (Table S7.2), activation for recent items was significantly higher than for remote items, consistent across both age groups.

Taken together, these analyses support the conclusion that hippocampal involvement in successful retrieval is sustained across delays, while other ROIs such as the precuneus may show greater engagement for more recent memories. We have now updated the manuscript text (lines 370-390) and supplementary materials to reflect these findings more clearly, as well as to clarify the distinction between activation relative to baseline and memory-agerelated modulation.

(4) Related to point 3, the claims about hippocampus with respect to multiple trace theory feel very unsupported by the data. I believe the authors want to conclude that children's memory retrieval shows reliance on hippocampus irrespective of delay, presumably because this is a detailed memory task. However the authors have not really shown this; all they have shown is that hippocampal involvement (whatever it is) does not vary by delay. But we do not have compelling evidence that the hippocampus is involved in this task at all. That hippocampus is more active during retrieval than implicit baseline is a very low bar and does not necessarily indicate a role in memory retrieval. If the authors want to make this claim, more data are needed (e.g., showing that hippocampal activity during retrieval is higher when the upcoming memory retrieval is successful vs. unsuccessful). In the absence of this, I think all the claims about multiple trace theory supporting retrieval similarly across delays and that this is operational in children are inappropriate and should be removed.

We thank the reviewer for pointing this out. We agree that additional analysis of hippocampal activity during successful and unsuccessful memory retrieval is warranted. This will provide stronger support for our claim that strong, detailed memories during retrieval rely on the hippocampus in both children and adults. Our previously presented results on the remote > recent univariate signal difference in the hippocampus (p. 14-18; lines 433-376, Fig. 3A) show that this difference does not vary between children and adults, or between Day 1 and Day 14. Our further analysis showed that both anterior and posterior regions of the hippocampus exhibited significant activation from zero in children and adults for correctly remembered items (see Fig. S2, Table S7 in Supplementary Materials). Based on your suggestion, our recent additional analysis showed:

(i) For forgotten items, we did not observe any activation significantly higher than zero in either the anterior or posterior hippocampus for recent and remote memory on Day 1 and Day 14 in either age group (all p > .052 FDR corrected; see Table S7.1, Fig. S2.1).

(ii) After establishing no difference between recent and remote activation across and between sessions (Day 1, Day 14), we conducted another linear mixed-effects model with group x memory success (remembered, forgotten) x region (anterior hippocampus, posterior hippocampus), with subject as a random effect. The model showed no significant effects for the memory success x region interaction (F = 1.12(1,1198), p = .289) and no significant group x memory success x region interaction (F = .017(1,1198), p = .895). However, we observed a significant main effect of memory success (F = 128.27(1,1198), p < .001), indicating significantly higher hippocampal activation for remembered compared to forgotten items (b = .06, t = 11.29, p <.001; see Table S7.3).

(iii) Considering the comparatively low number of incorrect trials for recent items in the adult group, we reran this analysis only for remote items. Similarly, the model showed no significant effects for the memory success x region interaction (F = .72(1,555), p = .398) and no significant group x memory success x region interaction (F = .14(1,555), p = .705). However, we observed a significant main effect of memory success (F = 68.03(1,555), p < .001), indicating significantly higher hippocampal activation for remote remembered compared to forgotten items (b = .07, t = 8.20, p <.001; see Table S7.3).

Taken together, our results indicate that significant hippocampal activation was observed only for correctly remembered items in both children and adults, regardless of memory age and session. For forgotten items, we did not observe any significant hippocampal activation in either group or delay. Moreover, hippocampal activation was significantly higher for remembered compared to forgotten memories. This evidence supports our conclusions regarding the Multiple Trace and Trace Transformation Theories, suggesting that the hippocampus supports retrieval similarly across delays, and provides novel evidence that this process is operational in both children and adults. This aligns also with Contextual Bindings Theory, as well as empirical evidence by Sekeres, Winokur, & Moscovitch (2018), among others. We have added this information to the manuscript.

(5) There are still not enough methodological details in the main paper to make sense of the results. Some of these problems were addressed in the revision but others remain. For example, a couple of things that were unclear: that initially learned locations were split, where half were tested again at day 1 and the other half at day 14; what specific criterion was used to determine to pick the 'well-learned' associations that were used for comparisons at different delay periods (object-scene pairs that participants remembered accurately in the last repetition of learning? Or across all of learning?).

We thank the reviewer for pointing this out. The initially learned object-scene associations on Day 0 were split in two halves based on their categories before the testing. Specifically, half of the pairs from the first set and half of the pairs from the second set of 30 object-scene associations were used to create the set 30 remote pair for Day 1 testing. A similar procedure was repeated for the remaining pairs to create a set of remote object-scene associations for Day 14 retrieval. We tried to equally distribute the categories of pairs between the testing sets. We added this information to the methods section of the manuscript (see p. 47, lines 12371243). In addition, the sets of association for delay test on Day 1 and Day 14 were not based on their learning accuracy. Of note, the analysis of variance revealed that there was no difference in learning accuracy between the two sets created for delay tests in either age group (children: p = .23; adults p = .06). These results indicate that the sets were comprised of items learned with comparable accuracy in both age groups.

(6) In still find the revised Introduction a bit unclear. I appreciated the added descriptions of different theories of consolidation, though the order of presented points is still a bit hard to follow. Some of the predictions I also find a bit confusing as laid out in the introduction. (1) As noted in the paper multiple trace theory predicts that hippocampal involvement will remain high provided memories retained are sufficiently high detail. The authors however also predict that children will rely more on gist (than detailed) memories than adults, which would seem to imply (combined with the MTT idea) that they should show reduced hippocampal involvement over time (while in adults, it should remain high). However, the authors' actual prediction is that hippocampus will show stable involvement over time in both kids and adults. I'm having a hard time reconciling these points. (2) With respect to the extraction of gist in children, I was confused by the link to Fuzzy Trace Theory given the children in the present study are a bit young to be showing the kind of gist extraction shown in the Brainerd & Reyna data. Would 5-7 year olds not be more likely to show reliance on verbatim traces under that framework? Also from a phrasing perspective, I was confused about whether gist-like information was something different from just gist in this sentence: "children may be more inclined to extract gist information at the expense of detailed or gist-like information." (p. 8) - is this a typo?

We thank the reviewer for this thoughtful observation.

Our hypothesis of stable hippocampal engagement over time was primarily based on Contextual Binding Theory (Yonelinas et al., 2019), and the MTT, supported by the evidence provided by Sekeres et al., 2018, which posits that the hippocampus continues to support retrieval when contextual information is preserved, even for older, consolidated memories. Given that our object-location associations were repeatedly encoded and tied to specific scene contexts, we believe that retrieval success for both recent and remote memories likely involved contextual reinstatement, leading to sustained hippocampal activity. Also in accordance with the MTT and related TTT, different memory representations may coexist, including detailed and gist-like memories. Therefore, we suggest that children may not rely on highly detailed item-specific memory, but rather on sufficiently contextualized schematic traces, which still engage the hippocampus. This distinction is now made clearer in the Introduction (see lines 223-236).

We appreciate the reviewer’s point regarding Fuzzy Trace Theory (Brainerd & Reyna, 2002). Indeed, in classic FTT, young children are thought to rely more on verbatim traces due to immature gist extraction mechanisms (primarily from verbal material). However, we use the term “gist-like representations” to refer to schematic or category-level retrieval that emerges through structured, repeated learning (as in our task). This form of abstraction may not require full semantic gist extraction in the FTT sense but may instead reflect consolidation-driven convergence onto shared category-level representations — especially when strategic resources are limited. We now clarify this distinction and revise the ambiguous sentence with typo (“at the expense of detailed or gist-like information”) to better reflect our intended meaning (see p.8).

(7) For the PLSC, if I understand this correctly, the profiles were defined for showing associations with behaviour across age groups. (1) As such, is it not "double dipping" to then show that there is an association between brain profile and behaviour-must this not be true by definition? If I am mistaken, it might be helpful to clarify this in the paper. (2) In addition, I believe for the univariate and scene-specific reinstatement analyses these profiles were defined across both age groups. I assume this doesn't allow for separate definition of profiles across the two group (i.e., a kind of "interaction"). If this is the case, it makes sense that there would not be big age differences... the profiles were defined for showing an association across all subjects. If the authors wanted to identify distinct profiles in children and adults they may need to run another analysis.

We thank the reviewer for this thoughtful comment.

(1) We agree that showing the correlation between the latent variable and behavior may be redundant, as the relationship is already embedded in the PLSC solution and quantified by the explained variance. Our intention was merely to visualize the strength of this relationship. In hindsight, we agree that this could be misinterpreted, and we have removed the additional correlation figure from the manuscript.

We also see the reviewer’s point that, given the shared latent profile across groups, it is expected that the strength of the brain-behavior relationship does not differ between age groups. Instead, to investigate group differences more appropriately, we examined whether children and adults differed in their expression of the shared latent variable (i.e., brain scores). This analysis revealed that children showed significantly lower brain scores than adults both in short delay, t(83) = -4.227, p = .0001, and long delay, t(74) = -5.653, p < .001, suggesting that while the brain-behavior profile is shared, its expression varies by group. We have added this clarification to the Results section (p. 19-20) of the revised manuscript.

(2) Regarding the second point, we agree with the reviewer that defining the PLS profiles across both age groups inherently limits the ability to detect group-specific association, as the resulting latent variables represent shared pattern across the full sample. To address this, we conducted additional PLS analyses separately within each age group to examine whether distinct neural upregulation profiles (remote > recent) emerge for short and long delay conditions.

These within-group analyses, however, were based on smaller subsamples, which reduced statistical power, especially when using bootstrapping to assess the stability of the profiles. For the short delay, although some regions reached significance, the overall latent variables did not reach conventional thresholds for stability (all p > .069), indicating that the profiles were not robust. This suggests that within-group PLS analyses may be underpowered to detect subtle effects, particularly when modelling neural upregulation (remote > recent), which may be inherently small.

Nonetheless, when we exploratively applied PLSC separately within each group using recent and remote activity levels against the implicit baseline (rather than the contrast remote > recent) and its relation to memory performance, we observed significant and stable latent variables in both children and adults. This implies that such contrasts (vs. baseline) may be more sensitive and better suited to detect meaningful brain–behavior relationships within age groups. We have added this clarification to the Results sections of the manuscript to highlight the limitations of within-group contrasts for neural upregulation.

**Author response image 1. sa2fig1:** 

(3) Also, as for differences between short delay brain profile and long delay brain profile for the scene-specific reinstatement - there are 2 regions that become significant at long delay that were not significant at a short delay (PC, and CE). However, given there are ceiling effects in behaviour at the short but not long delay, it's unclear if this is a meaningful difference or just a difference in sensitivity. Is there a way to test whether the profiles are statistically different from one another?

We thank the reviewer for this comment. To better illustrate differential profiles also for high memory accuracy after immediate delay (30 minutes delay), we added the immediate (30 minutes delay) condition as a third reference point, given the availability of scene-specific reinstatement data at this time point. Interestingly, the immediate reinstatement profile revealed a different set of significant regions, with distinct expression patterns compared to both the short and long delay conditions. This supports the view that scene-specific reinstatement is not static but dynamically reorganized over time.

Regarding the ceiling effect at short delay, we acknowledge this as a potential limitation. However, we note that our primary analyses were conducted across both age groups combined, and not solely within high-performing individuals. As such, the grouping may mitigate concerns that ceiling-level performance in a subset of participants unduly influenced the overall reinstatement profile. Moreover, we observed variation in neural reinstatement despite ceiling-level behavior, suggesting that the neural signal retains sensitivity to consolidation-related processes even when behavioral accuracy is near-perfect.

While we agree that formal statistical comparisons of reinstatement profiles across delays (e.g., using representational profile similarity or interaction tests) could be an informative direction, we feel that this goes beyond the scope of the current manuscript.

(4) As I mentioned above, it also was not ideal in my opinion that all regions were included for the scene-specific reinstatement due to the authors' inability to have an appropriate baseline and therefore define above-chance reinstatement. It makes these findings really challenging to compare with the gist reinstatement ones.

We appreciate the reviewer’s comment and agree that the lack of a clearly defined baseline for scene-specific reinstatement limits our ability to determine whether these values reflect above-chance reinstatement. However, we would like to clarify that we do not directly compare the magnitude of scene-specific reinstatement to that of gist-like reinstatement in our analyses or interpretations. These two analyses serve complementary purposes: the scenespecific analysis captures trial-unique similarity (within-item reinstatement), while the gistlike analysis captures category-level representational structure (across items). Because they differ not only in baseline assumptions but also in analytical scope and theoretical interpretation, our goal was not to compare them directly, but rather to explore distinct but co-existing representational formats that may evolve differently across development and delay.

(8) I would encourage the authors to be specific about whether they are measuring/talking about memory representations versus reinstatement, unless they think these are the same thing (in which case some explanation as to why would be helpful). For example, especially under the Fuzzy Trace framework, couldn't someone maintain both verbatim and gist traces of a memory yet rely more on one when making a memory decision?

We thank the reviewer for pointing out the importance of conceptual clarity when referring to memory representations versus reinstatement. We agree that these are distinct but related concepts: in our framework, memory representations refer to the neural content stored as a result of encoding and consolidation, whereas reinstatement refers to the reactivation of those representations during retrieval. Thus, reinstatement serves as a proxy for the underlying memory representation — it is how we measure or infer the nature (e.g., specificity, abstraction) of the stored content.

Under Fuzzy Trace Theory, it is indeed possible for both verbatim and gist representations to coexist. Our interpretation is not that children lack verbatim traces, but rather that they are more likely to rely on schematic or gist-like representations during retrieval, especially after a delay. Our use of neural pattern similarity (reinstatement) reflects which type of representation is being accessed, not necessarily which traces exist in parallel.

To avoid ambiguity, we have revised the manuscript to more explicitly distinguish between reinstatement (neural reactivation) and the representational format (verbatim vs. gist-like), especially in the framing of our hypotheses and interpretation of age group differences.

(9) With respect to the learning criteria - it is misleading to say that "children needed between two to four learning-retrieval cycles to reach the criterion of 83% correct responses" (p. 9). Four was the maximum, and looking at the Figure 1C data it appears as though there were at least a few children who did not meet the 83% minimum. I believe they were included in the analysis anyway? Please clarify. Was there any minimum imposed for inclusion?

We thank the reviewer for pointing this out. As stated in Methods Section (p. 50, lines 13261338) “These cycles ranged from a minimum of two to a maximum of four.<…> The cycles ended when participants provided correct responses to 83% of the trials or after the fourth cycle was reached.” We have corrected the corresponding wording in the Results section (line 286-289) to reflect this more accurately. Indeed, five children did not reach the 83% criterion but achieved final performance between 70 and 80% after the fourth learning cycle. These participants were included in this analysis for two main reasons:

(1) The 83% threshold was established during piloting as a guideline for how many learningretrieval cycles to allow, not a strict learning criterion. It served to standardize task continuation, rather than to exclude participants post hoc.

(2) The performance of these five children was still well above chance level (33%), indicating meaningful learning. Excluding them would have biased the sample toward higherperforming children and reduced the ecological validity of our findings. Including them ensures a more representative view of children’s performance under extended learning conditions.

(10) For the gist-like reinstatement PLSC analysis, results are really similar a short and long delays and yet some of the text seems to implying specificity to the long delay. One is a trend and one is significant (p. 31), but surely these two associations would not be statistically different from one another**?**

We agree with the reviewer that the associations at short and long delays appeared similar. While a formal comparison (e.g., using a Z-test for dependent correlations) would typically be warranted, in the reanalyzed dataset only the long delay profile remains statistically significant, which limits the interpretability of such a comparison.

(11) As a general comment, I had a hard time tying all of the (many) results together. For example adults show more mature neocortical consolidation-related engagement, which the authors say is going to create more durable detailed memories, but under multiple trace theory we would generally think of neocortical representations as providing more schematic information. If the authors could try to make more connections across the different neural analyses, as well as tie the neural findings in more closely with the behaviour & back to the theoretical frameworks, that would be really helpful.

We thank the reviewer for this valuable suggestion. We have revised the discussion section to more clearly link the behavioral and neural findings and to interpret them in light of existing consolidation theories for better clarity.

**Reviewer #2 (Public Review):**
Schommartz et al. present a manuscript characterizing neural signatures of reinstatement during cued retrieval of middle-aged children compared to adults. The authors utilize a paradigm where participants learn the spatial location of semantically related item-scene memoranda which they retrieve after short or long delays. The paradigm is especially strong as the authors include novel memoranda at each delayed time point to make comparisons across new and old learning. In brief, the authors find that children show more forgetting than adults, and adults show greater engagement of cortical networks after longer delays as well as stronger item-specific reinstatement. Interestingly, children show more category-based reinstatement, however, evidence supports that this marker may be maladaptive for retrieving episodic details. The question is extremely timely both given the boom in neurocognitive research on the neural development of memory, and the dearth of research on consolidation in this age group. Also, the results provide novel insights into why consolidation processes may be disrupted in children.We thank the reviewer for the positive evaluation.Comments on the revised version:I carefully reviewed not only the responses to my own reviews as well as those raised by the other reviewers. While they addressed some of the concerns raised in the process, I think many substantive concerns remain.Regarding Reviewer 1:The authors point that the retrieval procedure is the same over time and similarly influenced by temporal autocorrelations, which makes their analysis okay. However, there is a fundamental problem as to whether they are actually measuring reinstatement or they are only measuring differences in temporal autocorrelation (or some non-linear combination of both). The authors further argue that the stimuli are being processed more memory wise rather than perception wise, however, I think there is no evidence for that and that perception-memory processes should be considered on a continuum rather than as discrete processes. Thus, I agree with reviewer 1 that these analyses should be removed.

We thank the reviewer for raising this important question. We would like to clarify a few key points regarding temporal autocorrelation and reinstatement.

During the fixation window, participants were instructed to reinstate the scene and location associated with the cued object from memory. This task was familiar to them, as they had been trained in retrieving locations within scenes. Our analysis aims to compare the neural representations during this retrieval phase with those when participants view the scene, in order to assess how these representations change in similarity over time, as memories become less precise.

We acknowledge that temporal proximity can lead to temporal autocorrelation. However, evidence suggests that temporal autocorrelation is consistent and stable across conditions (Gautama & Van Hulle, 2004; Woolrich et al., 2004). Shinn & Lagalwar (2021)further demonstrated that temporal autocorrelation is highly reliable at both the subject and regional levels. Given that we analyze regions of interest (ROIs) separately, potential spatial variability in temporal autocorrelation is not a major concern.

No difference between item-specific reinstatement for recent items on day 1 and day 14 (which were merged) for further delay-related comparison also suggests that the reinstatement measure was stable for recent items even sampled at two different testing days.

Importantly, we interpret the relative change in the reinstatement index rather than its absolute value.

In addition, when we conducted the same analysis for incorrectly retrieved memories, we did not observe any delay-related decline in reinstatement (see p. 25, lines 623-627). This suggests that the delay-related changes in reinstatement are specific to correctly retrieved memories.

Finally, our control analysis examining reinstatement between object and fixation time points (as suggested by Reviewer 1) revealed no delay-related effects in any ROI (see p.24, lines 605-612), further highlighting the specificity of the observed delay-related change in item reinstatement.

We emphasize that temporal autocorrelation should be similar across all retrieval delays due to the identical task design and structure. Therefore, any observed decrease in reinstatement with increasing delay likely reflects a genuine change in the reinstatement index, rather than differences in temporal autocorrelation. Since our analysis includes only correctly retrieved items, and there is no perceptual input during the fixation window, this process is inherently memory-based, relying on mnemonic retrieval rather than sensory processing.

We respectfully disagree with the reviewer's assertion that retrieval during the fixation period cannot be considered more memory-driven than perception-driven. At this time point, participants had no access to actual images of the scene, making it necessary for them to rely on mnemonic retrieval. The object cue likely triggered pattern completion for the learned object-scene association, forming a unique memory if remembered correctly(Horner & Burgess, 2013). This process is inherently mnemonic, as it is based on reconstructing the original neural representation of the scene (Kuhl et al., 2012; Staresina et al., 2013).

While perception and memory processes can indeed be viewed as a continuum, some cognitive processes are predominantly memory-based, involving reconstruction rather than reproduction of previous experiences (Bartlett, 1932; Ranganath & Ritchey, 2012). In our task, although the retrieved material is based on previously encoded visual information, the process of recalling this information during the fixation period is fundamentally mnemonic, as it does not involve visual input. Our findings indicate that the similarity between memorybased representations and those observed during actual perception decreases over time, suggesting a relative change in the quality of the representations. However, this does not imply that detailed representations disappear; they may still be robust enough to support correct memory recall. Previous studies examining encoding-retrieval similarity have shown similar findings(Pacheco Estefan et al., 2019; Ritchey et al., 2013).

We do not claim that perception and memory processes are entirely discrete, nor do we suggest that only perception is involved when participants see the scene. Viewing the scene indeed involves recognition processes, updating retrieved representations from the fixation period, and potentially completing missing or unclear information. This integrative process demonstrates the interrelation of perception and memory, especially in complex tasks like the one we employed.

In conclusion, our task design and analysis support the interpretation that the fixation period is primarily characterized by mnemonic retrieval, facilitated by cue-triggered pattern completion, rather than perceptual processing. We believe this approach aligns with the current understanding of memory retrieval processes as supported by the existing literature.

The authors seem to have a design that would allow for across run comparisons, however, they did not include these additional analyses.

Thank you for pointing this out. We ran as additional cross-run comparison. This results and further proceeding are reported in the comment for reviewer 1.

To address the reviewer’s concern, we conducted an additional cross-run analysis for all correctly retrieved trials. The approach restricted comparisons to non-overlapping runs (run1run2, run2-run3, run1-run3). This analysis revealed robust gist-like reinstatement in children for remote Day 14 memories in the mPFC (p = .035) and vlPFC (p = .0007), in adults’ vlPFC remote Day 1 memories (p = .029), as well as in children and adults remote Day 1 memories in LOC (p < .02). A significant Session effect in both regions (mPFC: p = .026; vlPFC: p = .002) indicated increased reinstatement for long delay (Day 14) compared to short-delay and recent session (all p < .05). Given that the cross-run results largely replicate and reinforce the effects found previously with within-run, we believe that combining both sources of information is methodologically justified and statistically beneficial. Specifically, both approaches independently identified significant gist-like reinstatement in children’s mPFC and vlPFC (although within-run vlPFC effect (short delay: p = .038; long delay p = .047) did not survive multiple comparisons), particularly for remote memories. Including both withinrun and between-run comparisons increases the number of unique, non-repeated trial pairs, improving statistical power without introducing redundancy. While we acknowledge that same-run comparisons may be influenced by residual autocorrelation(Prince et al., 2022), we believe that our design mitigates this risk through consistency between within-run and crossrun results, long inter-trial intervals, and trial-wise estimation of activation. We have adjusted the manuscript, accordingly, reporting the combined analysis. We also report cross-run and within-run analysis separately in supplementary materials Tables S12.1, S12.2, showing that they converge with the cross-run results and thus strengthen rather than dilute the findings.

As suggested, we now explicitly highlight the change over time as the central finding. We observe a clear increase in gist-like reinstatement from recent to remote memories in children, particularly in mPFC and vlPFC. These effects based on combined within- and cross-run comparisons, are now clearly stated in the main results and interpreted in the discussion accordingly.

(1) The authors did not satisfy my concerns about different amounts of re-exposures to stimuli as a function of age, which introduces a serious confound in the interpretation of the neural data.(2) Regarding Reviewer 1's point about different number of trials being entered into analysis, I think a more formal test of sub-sampling the adult trials is warranted.

(1) We thank the reviewer for pointing this out. Overall, children needed 2 to 4 learning cycles to improve their performance and reach the learning criteria, compared to 2 learning cycles in adults. To address the different amounts of re-exposure to stimuli between the age groups, we subsampled the child group to only those children who reached the learning criteria after 2 learning cycles. For this purpose, we excluded 21 children from the analysis who needed 3 or 4 learning cycles. This resulted in 39 young adults and 28 children being included in the subsequent analysis.

(i) We reran the behavioral analysis with the subsampled dataset (see Supplementary Materials, Table S1.1, Fig. S1, Table S1.2). This analysis replicated the previous findings of less robust memory consolidation in children across all time delays.

(ii) We reran the univariate analysis (see in Supplementary Materials, Table S9.1). This analysis also replicated fully the previous findings. This indicates that the inclusion of child participants with greater material exposure during learning in the analysis of neural retrieval patterns did not affect the group differences in univariate neural results.

These subsampled results demonstrated that the amount of re-exposure to stimuli during encoding does not affect consolidation-related changes in memory retrieval at the behavioral and neural levels in children and adults across all time delays. We have added this information to the manuscript (line 343-348, 420-425).

(2) We appreciate Reviewer 1's suggestion to perform a formal test by sub-sampling the adult trials to match the number of trials in the child group. However, we believe that this approach may not be optimal for the following reasons:

(i) Loss of Statistical Power: Sub-sampling the adult trials would result in a reduced sample size, potentially leading to a significant loss of statistical power and the ability to detect meaningful effects, particularly in a context where the adult group is intended to serve as a robust control or comparison group.

(ii) Introducing sub-sampling could introduce variability that complicates the interpretation of results, particularly if the trial sub-sampling process does not fully capture the variability inherent in the original adult data.

(iii) Robustness of Existing Findings: We have already addressed potential concerns about unequal trial numbers by conducting analyses that control for the number of learning cycles, as detailed in our supplementary materials. These analyses have shown that the observed effects are consistent, suggesting that the differences in trial numbers do not critically influence our findings.

Given these considerations, we hope the reviewer understands our rationale and agrees that the current analysis is robust and appropriate for addressing the research questions.

I also still fundamentally disagree with the use of global signals when comparing children to adults, and think this could very much skew the results.

We thank the reviewer for raising this important issue. To address this concern comprehensively, we have taken the following steps:

(1) Overview of the literature support for global signal regression (GSR). A growing body of methodological and empirical research supports the inclusion of global signal repression as part of best practice denoising pipelines, particularly when analyzing pediatric fMRI data. Studies such as (Ciric et al., 2017; Parkes et al., 2018; J. D. Power et al., 2012, 2014; Power et al., 2012), and (Thompson et al., 2016) show that GSR improves motion-related artifact removal. Critically, pediatric-specific studies (Disselhoff et al., 2025; Graff et al., 2022) conclude that pipelines including GSR are most effective for signal recovery and artifact removal in younger children. Graff et al. (2021) demonstrated that among various pipelines, GSR yielded the best noise reduction in 4–8-year-olds. Additionally, (Li et al., 2019; Qing et al., 2015) emphasized that GSR reduces artifactual variance without distorting the spatial structure of neural signals. (Ofoghi et al., 2021)demonstrated that global signal regression helps mitigate non-neuronal noise sources, including respiration, cardiac activity, motion, vasodilation, and scanner-related artifacts. Based on this and other recent findings, we consider GSR particularly beneficial for denoising paediatric fMRI data in our study.

(2) Empirical comparison of pipelines with and without GSR. We re-run the entire first-level univariate analysis using the pipeline that excluded the global signal regression. The resulting activation maps (see Supplementary Figure S3.2, S4.2, S5.2, S9.2) differed notably from the original pipeline. Specifically, group differences in cortical regions such as mPFC, cerebellum, and posterior PHG no longer reached significance, and the overall pattern of results appeared noisier.

(3) Evaluation of the pipeline differences. To further evaluate the impact of GSR, we conducted the following analyses:

(a) Global signal is stable across groups and sessions. A linear mixed-effects model showed no significant main effects or interactions involving group or session on the global signal (F-values < 2.62, p > .11), suggesting that the global signal was not group- or session-dependent in our sample.

(b) Noise Reduction Assessment via Contrast Variability. We compared the variability (standard deviation and IQR) of contrast estimates across pipelines. Both SD (b = .070, p < .001) and IQR (b = .087, p < .001) were significantly reduced in the GSR pipeline, especially in children (p < .001) compared to adults (p = .048). This suggests that GSR reduces inter-subject variability in children, likely reflecting improved signal quality.

(c) Residual Variability After Regressing Global Signal. We regressed out global signal post hoc from both pipelines and compared the residual variance. Residual standard deviation was significantly lower for the GSR pipeline (F = 199, p < .001), with no interaction with session or group, further indicating that GSR stabilizes the signal and attenuates non-neuronal variability.

Conclusion

In summary, while we understand the reviewer’s concern, we believe the empirical and theoretical support for GSR, especially in pediatric samples, justifies its use in our study. Nonetheless, to ensure full transparency, we provide full results from both pipelines in the Supplementary Materials and have clarified our reasoning in the revised manuscript.

**Reviewer #1 (Recommendations For The Authors):**
(1) Some figures are still missing descriptions of what everything on the graph means; please clarify in captions**.**

We thank the reviewer for pointing this out. We undertook the necessary adjustments in the graph annotations.

(2) The authors conclude they showed evidence of neural reorganization of memory representations in children (p. 41). But the gist is not greater in children than adults, and also does not differ over time-so, I was confused about what this claim was based on?

We thank the reviewer for raising this question. Our results on gist-like reinstatements suggest that gist-like reinstatement was significantly higher in children compared to adults in the mPFC in addition to the child gist-like reinstatement indices being significantly higher than zero (see p.27-28). These results support our claim on neural reorganization of memory represenations in children. We hope this clarifies the issue.

References

Bartlett, F. C. (1932). Remembering: A study in experimental and social psychology. Cambridge University Press.

Brainerd, C. J., & Reyna, V. F. (2002). Fuzzy-Trace Theory: Dual Processes in Memory, Reasoning, and Cognitive Neuroscience (pp. 41–100). https://doi.org/10.1016/S00652407(02)80062-3

Chen, J., Leong, Y. C., Honey, C. J., Yong, C. H., Norman, K. A., & Hasson, U. (2017). Shared memories reveal shared structure in neural activity across individuals. Nature Neuroscience, 20(1), 115–125. https://doi.org/10.1038/nn.4450

Ciric, R., Wolf, D. H., Power, J. D., Roalf, D. R., Baum, G. L., Ruparel, K., Shinohara, R. T., Elliott, M. A., Eickhoff, S. B., Davatzikos, C., Gur, R. C., Gur, R. E., Bassett, D. S., & Satterthwaite, T. D. (2017). Benchmarking of participant-level confound regression strategies for the control of motion artifact in studies of functional connectivity. NeuroImage, 154, 174–187. https://doi.org/10.1016/j.neuroimage.2017.03.020

Disselhoff, V., Jakab, A., Latal, B., Schnider, B., Wehrle, F. M., Hagmann, C. F., Held, U., O’Gorman, R. T., Fauchère, J.-C., & Hüppi, P. (2025). Inhibition abilities and functional brain connectivity in school-aged term-born and preterm-born children. Pediatric Research, 97(1), 315–324. https://doi.org/10.1038/s41390-024-03241-0

Esteban, O., Ciric, R., Finc, K., Blair, R. W., Markiewicz, C. J., Moodie, C. A., Kent, J. D., Goncalves, M., DuPre, E., Gomez, D. E. P., Ye, Z., Salo, T., Valabregue, R., Amlien, I. K., Liem, F., Jacoby, N., Stojić, H., Cieslak, M., Urchs, S., … Gorgolewski, K. J. (2020). Analysis of task-based functional MRI data preprocessed with fMRIPrep. Nature Protocols, 15(7), 2186–2202. https://doi.org/10.1038/s41596-020-0327-3

Fandakova, Y., Leckey, S., Driver, C. C., Bunge, S. A., & Ghetti, S. (2019). Neural specificity of scene representations is related to memory performance in childhood. NeuroImage, 199, 105–113. https://doi.org/10.1016/j.neuroimage.2019.05.050

Gautama, T., & Van Hulle, M. M. (2004). Optimal spatial regularisation of autocorrelation estimates in fMRI analysis. NeuroImage, 23(3), 1203–1216. https://doi.org/10.1016/j.neuroimage.2004.07.048

Graff, K., Tansey, R., Ip, A., Rohr, C., Dimond, D., Dewey, D., & Bray, S. (2022). Benchmarking common preprocessing strategies in early childhood functional connectivity and intersubject correlation fMRI. Developmental Cognitive Neuroscience, 54, 101087. https://doi.org/10.1016/j.dcn.2022.101087

Horner, A. J., & Burgess, N. (2013). The associative structure of memory for multi-element events. Journal of Experimental Psychology: General, 142(4), 1370–1383. https://doi.org/10.1037/a0033626

Jones, J. S., the CALM Team, & Astle, D. E. (2021). A transdiagnostic data-driven study of children’s behaviour and the functional connectome. Developmental Cognitive Neuroscience, 52, 101027. https://doi.org/10.1016/j.dcn.2021.101027

Kuhl, B. A., Bainbridge, W. A., & Chun, M. M. (2012). Neural Reactivation Reveals Mechanisms for Updating Memory. Journal of Neuroscience, 32(10), 3453–3461. https://doi.org/10.1523/JNEUROSCI.5846-11.2012

Kuhl, B. A., & Chun, M. M. (2014). Successful Remembering Elicits Event-Specific Activity Patterns in Lateral Parietal Cortex. Journal of Neuroscience, 34(23), 8051–8060. https://doi.org/10.1523/JNEUROSCI.4328-13.2014

Li, J., Kong, R., Liégeois, R., Orban, C., Tan, Y., Sun, N., Holmes, A. J., Sabuncu, M. R., Ge, T., & Yeo, B. T. T. (2019). Global signal regression strengthens association between resting-state functional connectivity and behavior. NeuroImage, 196, 126–141. https://doi.org/10.1016/j.neuroimage.2019.04.016

Ofoghi, B., Chenaghlou, M., Mooney, M., Dwyer, D. B., & Bruce, L. (2021). Team technical performance characteristics and their association with match outcome in elite netball. International Journal of Performance Analysis in Sport, 21(5), 700–712. https://doi.org/10.1080/24748668.2021.1938424

Pacheco Estefan, D., Sánchez-Fibla, M., Duff, A., Principe, A., Rocamora, R., Zhang, H., Axmacher, N., & Verschure, P. F. M. J. (2019). Coordinated representational reinstatement in the human hippocampus and lateral temporal cortex during episodic memory retrieval. Nature Communications, 10(1), 2255. https://doi.org/10.1038/s41467019-09569-0

Parkes, L., Fulcher, B., Yücel, M., & Fornito, A. (2018). An evaluation of the efficacy, reliability, and sensitivity of motion correction strategies for resting-state functional MRI. NeuroImage, 171, 415–436. https://doi.org/10.1016/j.neuroimage.2017.12.073

Power, J. D., Barnes, K. A., Snyder, A. Z., Schlaggar, B. L., & Petersen, S. E. (2012). Spurious but systematic correlations in functional connectivity MRI networks arise from subject motion. NeuroImage, 59(3), 2142–2154. https://doi.org/10.1016/j.neuroimage.2011.10.018

Power, J. D., Mitra, A., Laumann, T. O., Snyder, A. Z., Schlaggar, B. L., & Petersen, S. E. (2014). Methods to detect, characterize, and remove motion artifact in resting state fMRI. NeuroImage, 84, 320–341. https://doi.org/10.1016/j.neuroimage.2013.08.048

Power, S. D., Kushki, A., & Chau, T. (2012). Intersession Consistency of Single-Trial Classification of the Prefrontal Response to Mental Arithmetic and the No-Control State by NIRS. PLoS ONE, 7(7), e37791. https://doi.org/10.1371/journal.pone.0037791

Prince, J. S., Charest, I., Kurzawski, J. W., Pyles, J. A., Tarr, M. J., & Kay, K. N. (2022). Improving the accuracy of single-trial fMRI response estimates using GLMsingle. ELife, 11. https://doi.org/10.7554/eLife.77599

Qing, Z., Dong, Z., Li, S., Zang, Y., & Liu, D. (2015). Global signal regression has complex effects on regional homogeneity of resting state fMRI signal. Magnetic Resonance Imaging, 33(10), 1306–1313. https://doi.org/10.1016/j.mri.2015.07.011

Ranganath, C., & Ritchey, M. (2012). Two cortical systems for memory-guided behaviour. Nature Reviews Neuroscience, 13(10), 713–726. https://doi.org/10.1038/nrn3338

Ritchey, M., Wing, E. A., LaBar, K. S., & Cabeza, R. (2013). Neural Similarity Between Encoding and Retrieval is Related to Memory Via Hippocampal Interactions. Cerebral Cortex, 23(12), 2818–2828. https://doi.org/10.1093/cercor/bhs258

Satterthwaite, T. D., Elliott, M. A., Gerraty, R. T., Ruparel, K., Loughead, J., Calkins, M. E., Eickhoff, S. B., Hakonarson, H., Gur, R. C., Gur, R. E., & Wolf, D. H. (2013). An improved framework for confound regression and filtering for control of motion artifact in the preprocessing of resting-state functional connectivity data. NeuroImage, 64, 240–256. https://doi.org/10.1016/j.neuroimage.2012.08.052

Schommartz, I., Lembcke, P. F., Pupillo, F., Schuetz, H., de Chamorro, N. W., Bauer, M., Kaindl, A. M., Buss, C., & Shing, Y. L. (2023). Distinct multivariate structural brain profiles are related to variations in short- and long-delay memory consolidation across children and young adults. Developmental Cognitive Neuroscience, 59. https://doi.org/10.1016/J.DCN.2022.101192

Sekeres, M. J., Winocur, G., & Moscovitch, M. (2018). The hippocampus and related neocortical structures in memory transformation. Neuroscience Letters, 680, 39–53. https://doi.org/10.1016/j.neulet.2018.05.006

Shinn, L. J., & Lagalwar, S. (2021). Treating Neurodegenerative Disease with Antioxidants: Efficacy of the Bioactive Phenol Resveratrol and Mitochondrial-Targeted MitoQ and SkQ. Antioxidants, 10(4), 573. https://doi.org/10.3390/antiox10040573

Staresina, B. P., Alink, A., Kriegeskorte, N., & Henson, R. N. (2013). Awake reactivation predicts memory in humans. Proceedings of the National Academy of Sciences, 110(52), 21159–21164. https://doi.org/10.1073/pnas.1311989110

St-Laurent, M., & Buchsbaum, B. R. (2019). How Multiple Retrievals Affect Neural Reactivation in Young and Older Adults. The Journals of Gerontology: Series B, 74(7), 1086–1100. https://doi.org/10.1093/geronb/gbz075

Thompson, G. J., Riedl, V., Grimmer, T., Drzezga, A., Herman, P., & Hyder, F. (2016). The Whole-Brain “Global” Signal from Resting State fMRI as a Potential Biomarker of Quantitative State Changes in Glucose Metabolism. Brain Connectivity, 6(6), 435–447. https://doi.org/10.1089/brain.2015.0394

Tompary, A., & Davachi, L. (2017). Consolidation Promotes the Emergence of Representational Overlap in the Hippocampus and Medial Prefrontal Cortex. Neuron, 96(1), 228-241.e5. https://doi.org/10.1016/j.neuron.2017.09.005

Tompary, A., Zhou, W., & Davachi, L. (2020). Schematic memories develop quickly, but are not expressed unless necessary. PsyArXiv.

Woolrich, M. W., Behrens, T. E. J., Beckmann, C. F., Jenkinson, M., & Smith, S. M. (2004). Multilevel linear modelling for FMRI group analysis using Bayesian inference. NeuroImage, 21(4), 1732–1747. https://doi.org/10.1016/j.neuroimage.2003.12.023

Xiao, X., Dong, Q., Gao, J., Men, W., Poldrack, R. A., & Xue, G. (2017). Transformed Neural Pattern Reinstatement during Episodic Memory Retrieval. The Journal of Neuroscience, 37(11), 2986–2998. https://doi.org/10.1523/JNEUROSCI.2324-16.2017

Ye, Z., Shi, L., Li, A., Chen, C., & Xue, G. (2020). Retrieval practice facilitates memory updating by enhancing and differentiating medial prefrontal cortex representations. ELife, 9, 1–51. https://doi.org/10.7554/ELIFE.57023

Yonelinas, A. P., Ranganath, C., Ekstrom, A. D., & Wiltgen, B. J. (2019). A contextual binding theory of episodic memory: systems consolidation reconsidered. Nature Reviews. Neuroscience, 20(6), 364–375. https://doi.org/10.1038/S41583-019-01504

Zhuang, L., Wang, J., Xiong, B., Bian, C., Hao, L., Bayley, P. J., & Qin, S. (2021). Rapid neural reorganization during retrieval practice predicts subsequent long-term retention and false memory. Nature Human Behaviour, 6(1), 134–145.

https://doi.org/10.1038/s41562-021-01188-4